# Robust High-Dimensional Mean Estimation With Low Data Size, an Empirical Study

**Cullen Anderson**                                             *cyanderson@umass.edu*
*University of Massachusetts Amherst*

**Jeff M. Phillips**                                                 *jeffp@cs.utah.edu*
*University of Utah*

**Reviewed on OpenReview:** *https://openreview.net/forum?id=1QeI99nH9k*

## Abstract

Robust statistics aims to compute quantities to represent data where a fraction of it may be arbitrarily corrupted. The most essential statistic is the mean, and in recent years, there has been a flurry of theoretical advancement for efficiently estimating the mean in high dimensions on corrupted data. While several algorithms have been proposed that achieve near-optimal error, they all rely on large data size requirements as a function of dimension. In this paper, we perform an extensive experimentation over various mean estimation techniques where data size might not meet this requirement due to the high-dimensional setting.

For data with inliers generated from a Gaussian with known covariance, we find experimentally that several robust mean estimation techniques can practically improve upon the sample mean, with the *quantum entropy scaling* approach from Dong *et.al.* (NeurIPS 2019) performing consistently the best. However, this consistent improvement is conditioned on a couple of simple modifications to how the steps to prune outliers work in the high-dimension low-data setting, and when the inliers deviate significantly from Gaussianity. In fact, with these modifications, they are typically able to achieve roughly the same error as taking the sample mean of the uncorrupted inlier data, even with very low data size. In addition to controlled experiments on synthetic data, we also explore these methods on large language models, deep pretrained image models, and non-contextual word embedding models that do not necessarily have an inherent Gaussian distribution. Yet, in these settings, a mean point of a set of embedded objects is a desirable quantity to learn, and the data exhibits the high-dimension low-data setting studied in this paper. We show both the challenges of achieving this goal, and that our updated robust mean estimation methods can provide significant improvement over using just the sample mean. We additionally publish a library of Python implementations of robust mean estimation algorithms, allowing practitioners and researchers to apply these techniques and to perform further experimentation.

## 1 Introduction

Given samples from an unknown distribution, mean estimation is perhaps the most-fundamental and oldest problems in data analysis. And it is even more relevant in modern analysis for learning and AI tasks where data is very high dimensional, and there is little else one can reliably compute – at least not without first grappling with the mean.

In the past several years, there has been a flurry of theoretical advancement on this topic, including improved asymptotic bounds (Lee & Valiant, 2022; Gupta et al., 2023; Catoni, 2011; Gupta et al., 2024; Lugosi & Mendelson, 2017), and the development of more robust methods for dealing with adversarially corrupted

data distributions (Lai et al., 2016; Diakonikolas et al., 2017a; 2019a; Cheng et al., 2019a; Dong et al., 2019; Deshmukh et al., 2022). This paper supports this development in two key ways:

1. We provide a large experimental study of many new methods, which had not been thoroughly compared. In the non-corrupted case, with moderate data size we do not see substantial improvement over the classic sample mean approach. However, in the corrupted setting, we find that some methods can significantly improve upon the sample mean. In some cases consistent improvement on the sample mean requires adjustments that we develop. As a summary, the quantum entropy scaling approach of Dong et al. (2019) (using an adjustment we describe) consistently performs the best as long as inliers are reasonable similar to Gaussian, and often basically matches the mean of the (unknown) inlier data.

2. We bring to the fore the $d > n$ setting, where there are more dimensions $d$ than data points $n$, or at least we do not have $n$ as substantially larger than $d$. This setting is becoming more common as dimensionality grows, but has not typically been considered because the theoretical advancements did not provide exciting new bounds here. In this setting, we revise a common and critical outlier pruning step, and the key adjustment is ultimately simple: a $\sqrt{d/n}$ term, which vanishes when $n \gg d$, needs to be included in a key threshold. This is detailed in Section 3.1.

Our experimental study considers mean estimation in a variety of settings, focusing on when $n < d$ or $n$ is not much larger than $d$. While other experimental studies have been done, many like in Diakonikolas et al. (2017a) provided a comparison in the $n \gg d$ case. And while Deshmukh et al. (2022) has some experiments with $n$ not much larger than $d$, these are not nearly as comprehensive as our study. First, we consider standard Gaussian data with known covariance, and no corruption. Then we extend this to the setting with various types of adversarial corruption. We also consider some limited cases with unknown covariance. However, because straight-forward adaptations of mean-estimation approaches towards estimating covariance (mapping to a $\binom{d}{2}$-dimensional problem) further stresses the need for data size $n$ as a function of $d$, we defer a thorough exploration of this challenge to future work. Finally, we consider real world data scenarios where data is generated via embeddings resulting from large language models, deep pretrained image models, and word embedding models; here we do not have direct enforcement of Gaussianity of the data, but desire a high-dimensional mean nonetheless. In all cases, we consider a wide variety of efficient mean estimation approaches, including both classical ones and modern ones with stronger guarantees in the large $n$ setting. We provide an anonymous link to our code for easy reproducability here: `https://github.com/cullena20/RobustMeanEstimation`.

## 2 Background

We consider as input a set $X \subset \mathbb{R}^d$ of $n$ samples from an unknown distribution, and the goal is to estimate the mean of that distribution. Consider first the case where the distribution is the Gaussian $\mathcal{N}_d(0, I)$ where $I$ is the identity matrix representing an isotropic covariance. For $x \sim \mathcal{N}_d(0, I)$ we have $\mathbb{E}[\|x\|^2] = d$. For the sample mean $\bar{x} \in \mathbb{R}^d$ from $n$ points drawn iid from $\mathcal{N}_d(0, I)$ we have $\mathbb{E}[\|\bar{x}\|^2] = d/n$ and more importantly it strongly concentrates as $\Pr[|\|\bar{x}\|^2 - d/n| > t] \leq 2\exp(-Ct^2)$ for a constant $C$ (Vershynin, 2011). This implies for $n = \Omega(d/\varepsilon^2)$ we have $\|\bar{x}\| < \varepsilon$ with high probability; but for $d > n$ we do not

| key notation | |
|---|---|
| $n$ | # data samples |
| $d$ | # dimensions |
| $\varepsilon$ | error bound |
| $\eta$ | true corruption |
| $\tau$ | expected corruption |

get useful concentration results. The Gaussian is the most studied and used distribution for many reasons including that it has Normal marginals for any dimension, is easy to sample from, models an $\ell_2$ loss, and is the limiting distribution of the central limit theorem. As such, it is our main object of study. However, we note that other distributions have distinct behavior for the large $d$ setting. For instance, for $n$ samples from a distribution with mean $\mu$ and covariance $\Sigma$, the expected squared deviation from the mean in $d$ dimensions can be bounded by $\text{Tr}(\Sigma)/n$ (c.f., (Lee & Valiant, 2022)). This implies for instance if $X$ is drawn uniformly from a unit ball (so $\text{Tr}(\Sigma) = 1$) or other distributions with bounded $\text{Tr}(\Sigma)$, then the behavior for $d > n$ can

still be well-concentrated. On the other hand, other unbounded and heavy-tailed distributions where, like Gaussians, $\text{Tr}(\Sigma) = \Theta(d)$ [1], present similar challenges in the $d > n$ setting.

**Corrupted data models.** Another setting considers some fraction $\eta \in (0, \frac{1}{2})$ of the data to be adversarially corrupted from $X$ (Huber, 1964; Diakonikolas & Kane, 2023). Under the *Huber model*, we draw data $X \sim (1-\eta)P + \eta Q$ where $P$ is the set of inliers with mean $\mu$ (we consider $P = \mathcal{N}_d(\mu, I)$ as identity covariance Gaussian data), and $Q$ is any adversarial outlier distribution. The stronger *total variation* corruption model first draws $X' \sim P$ (with mean $\mu$), and then creates $X$ by adversarially changing any $\eta$-fraction of $X'$ to a new location. That is, it can also adversarially subtract data from the inlier data in addition to adding outliers. How accurately can we recover the mean $\mu$ under these settings? We mostly focus on the Huber model, and observe that subtractive corruption (a component of the stronger total variation model) can induce a consistent and hard to avoid error, and does not seem to expose significant differences between approaches.

As the mean minimizes the sum of squared deviations, the sample mean is very susceptible to outliers. A single point of corruption can arbitrarily affect the sample mean. On the other hand, such corruption can be easily detected by filtering out the furthest points from the sample mean, and recomputing the sample mean on the remainder of the data. A more challenging setting relocates points to roughly $\sqrt{d}$ from the mean, where the inliers are, but all in a tight cluster; then no individual points can be so easily filtered, but the sample mean can be given a non-trivial bias of as much as $\Omega(\eta\sqrt{d})$. We will empirically consider a variety of challenging $\eta$-corruption situations.

For many years, when dealing with high dimensions, practitioners were faced with either potentially large error (e.g., on order of $\eta\sqrt{d}$) in using the sample mean or other generalizations of the median (Small, 1990), or one could spend time exponential in $d$ and return an estimator that is guaranteed to be close to the true mean (Tukey, 1975) (or c.f., (Chen et al., 2015; Zhu et al., 2020a)). Around 2016, two papers broke this barrier (Lai et al., 2016) and (Diakonikolas et al., 2019a). They considered $X \sim \mathcal{N}_d(\mu, I)$, and allowed an $\eta$ fraction of the data to be corrupted and return an estimate of the mean $\hat\mu$ so that $\|\mu - \hat\mu\| \leq O(\eta\sqrt{\log 1/\eta})$ or $\leq O(\eta\sqrt{\log d})$. These works however assume $n = \Omega(d/\eta^2)$; otherwise one runs into the roadblock that even the sample mean of the inliers (the uncorrupted points) has more than $\eta$ error. Since then, much follow-up work has furthered our understanding. Some work (Dong et al., 2019; Cheng et al., 2019a; Depersin & Lecué, 2019) improved the time complexity of robust mean estimation algorithms, and our understanding of the problem's hardness (Diakonikolas et al., 2017c; Hopkins & Li, 2019). Others provide formulations where gradient descent can be used despite non-convexity (Cheng et al., 2020; Zhu et al., 2020b). There has also been effort to improve other robust statistics tasks such as covariance estimation (Chen et al., 2015; 2017; Cheng et al., 2019b), sparse estimation (Balakrishnan et al., 2017; Diakonikolas et al., 2019c; Cheng et al., 2022; Diakonikolas et al., 2022; 2024), list decodable learning (Charikar et al., 2017; Diakonikolas et al., 2017b), robustly learning mixtures of Gaussians (Bakshi et al., 2022), robust optimization (Diakonikolas et al., 2019b; Prasad et al., 2018), robust regression (Diakonikolas et al., 2018; Klivans et al., 2020), or in the context of adversarial machine learning (Tran et al., 2018). Importantly, robust statistics are more amenable to differential privacy, in particular to privacy through noise addition, and privacy mechanisms are naturally robust (Dwork & Lei, 2009; Liu et al., 2021a; Hopkins et al., 2023; Asi et al., 2023). Recent work has also expanded methods for different corruptions models (Liu et al., 2021b; Zhu et al., 2020c). For a more thorough review see the recent textbook by Diakonikolas & Kane (2023).

There has also been significant complementary work in mean estimation under heavy-tailed distributions (Lugosi & Mendelson, 2021; Lugosi, 2022; Gupta et al., 2024; Catoni, 2011; Lugosi & Mendelson, 2017; Devroye et al., 2015; Lee & Valiant, 2022); see the recent survey by Lugosi & Mendelson (2019). Recent work has also developed connections between optimality under heavy-tailed distributions, and optimality in the Huber corruption setting (Prasad et al., 2019).

---

[1]We use standard asymptotic notation so for some constants $C_1, C_2, C_3$ and functions $f, g$ then $g(x) = O(f(x))$ implies $\forall x > C_3$ then $g(x) \leq C_1 f(x) + C_2$; $g(x) = \Omega(f(x))$ implies $\forall x > C_3$ then $g(x) \geq C_1 f(x) + C_2$, with possibly different constants; and $g(x) = \Theta(f(x))$ implies $g(x) = O(f)$ and $g(x) = \Omega(f(x))$.

## 3 Mean Estimation Algorithms

Here we will document the mean estimation algorithms considered in this paper. Some are classic, and we also include several ones from the recent literature designed to be potentially practical and algorithmically efficient. Some include asymptotic theoretical bounds which use astronomical constants; we make a best effort to replace them with reasonable values so they remain practical. Some use an expected corruption parameter $\tau$, meant to be an upper bound true corruption, $\eta$. The ones we consider are as follows:

**sample_mean**: The *sample mean* simply returns $\hat{\mu} = \frac{1}{|X|} \sum_{x \in X} x$.

**coord_median**: The *coordinate-wise median* computes the median of each coordinate individually so $\hat{\mu}_j = \mathsf{median}(\{x_{i,j} \mid x_i \in X\})$.

**coord_trimmed_mean**: First compute a *trimmed mean estimator* for each coordinate individually, parameterized by a value $\tau \in (0,1)$. That is, in one dimension, it sorts the data, and removes $\tau|X|$ points which have the smallest values, and also removes $\tau|X|$ with largest values. Then it computes the mean of the remaining $(1-2\tau)|X|$ points. The *coordinate-wise trimmed mean* applies this estimator separately for each coordinate; which points are removed in coordinate $j$ have no bearing on which points are removed from coordinate $j'$ (Lugosi & Mendelson, 2021).

**median_of_means**: Split the data into $k$ chunks, find the mean of each chunk, take the coordinate wise median of these $k$ means (Lugosi & Mendelson, 2019; Minsker, 2023b;a). As a default, we set $k = 10$; this hyperparameter is explored in Appendix A.5.

**geometric_median**: The *geometric median* is the point which minimizes the sum of distances to all sample points. This is iteratively approximated using the Weiszfeld algorithm (Small, 1990; Vardi & Zhang, 2001).

**lee_valiant**: (Lee & Valiant (2022)) The Lee and Valiant algorithm first estimates the mean $\mu'$ on a $\gamma$ percentage of data points $X_\gamma$ using a mean estimator. It then centers all points to $X' = \{x' = x - \mu' \mid x \in X\}$. Let $X_t$ be the $t$ points in $X$ so their corresponding $x'$ have the largest norm. Let $X'_*$ be the subset consisting of $x' \in X'$ with their corresponding points *not* in $X_\gamma$ or in $X_t$. Then return $\mu' + \frac{1}{|X|} \sum_{x' \in X'_*} x'$. Rather than the extremely large constants in the original paper, we set $\gamma = 0.5$ and $t = \tau|X|$. As default, we use $\mathsf{median\_of\_means}_k$ estimator with $k = 10$ to obtain the initial mean estimator $\mu'$.

**LRV**: (Lai et al. (2016)) The LRV method recursively reduces the dimension by half, until 1 or 2 dimensions remain. Following the original author's code[2], in the ($\leq 2$)-dimensional base case, it returns coord_median. The recursive step has three components. First, it calculates a weight $w_i$ for each point $x_i$ as $w_i = \exp(-\|x_i - a\|^2/(Cs^2))$ where $s^2$ is a robust sample estimate of the trace of the true covariance matrix, $a$ is a rough estimator of the mean chosen as coord_median, and $C$ is a hyperparameter. We use $C = 1$; this hyper parameter is explored in Appendix A.5. Second, it computes $\mu_w = \frac{1}{|X|} \sum_{x_i \in X} w_i x_i$, which is the weighted mean of the input, and $\Sigma_w = \frac{1}{|X|} \sum_{x_i \in X} w_i(x_i - \mu_w)(x_i - \mu_w)^T$, which is the weighted covariance of the input. Let $V$ by the span of the top $\lfloor d/2 \rfloor$ singular vectors of $\Sigma_w$; let $V_\perp$ be the span of the bottom $\lceil d/2 \rceil$ singular vectors of $\Sigma_w$. Third, recurse on data projected onto $V$, and return an estimate $\mu_1$. We also build an estimator $\mu_2$ of the data projected onto the $\lceil d/2 \rceil$-dimensional remainder space $V_\perp$ using the weighted sample mean projected onto $V_\perp$: that is $\mu_2 = \frac{1}{|X_\perp|} \sum_{x_i^\perp \in X_\perp} w_i x_i^\perp$ where $X_\perp$ is the data projected onto $V_\perp$. Finally return $\mu_1 + \mu_2$.

**ev_filtering**: (Diakonikolas et al. (2019a;b)) This method observes that when inliers are from a standard Gaussian, then a set of corrupted data which substantially affects the mean estimate must result in a sufficiently large top eigenvalue after centering (i.e., of the sample covariance matrix), and this can be remedied by pruning points which are far along the top eigenvector. In this method, if after centering by the sample mean $\hat{\mu}$, the top eigenvalue exceeds $O(\tau \log 1/\tau)$ (Diakonikolas et al. (2017a)[3] implements this as $1 + 3\tau \log(1/\tau)$), then this data is considered additively corrupted along the direction of the top eigenvector. We call this the corruption detection step. Then they consider all points projected onto the associated top eigenvector and sorted $P = \langle p_1, \ldots, p_n \rangle$; and then a set of points furthest from the median $\mathsf{med}(P)$ are

---

[2] https://github.com/kevinalai/AgnosticMeanAndCovarianceCode
[3] https://github.com/hoonose/robust-filter

pruned. We call this the pruning step. The determination of which points to prune is based on those which exceed a Gaussian concentration inequality. Specifically, it finds the smallest index $i$ so $T_i = p_i - \mathsf{med}(P) - 2\tau$ satisfies $\frac{n-i}{n} > \gamma(\mathsf{erfc}(T_i/\sqrt{2})/2 + \tau/(d\log(d\tau/0.1)))$, where $\mathsf{erfc}$ is the complementary error function $(1-$ the cdf of the Normal) and prunes all points $i$ or larger. Intuitively, the centered projected data is expected to be a standard Normal distribution, and this bound compares the true percentage of points that exceed a threshold, $T_i$, with the probability that points will exceed that threshold, given by $\mathsf{erfc}$ with some slack terms added. Then the algorithm is recursively called with all points not-yet pruned until the top eigenvalue threshold is not violated. This algorithm critically assumes identity covariance and $n = \Omega(d/\tau^2) \gg d$.

**QUE**: (Dong et al. (2019)) Quantum Entropy Scoring, $\mathsf{QUE}$ for short, scores outliers based on quantum entropy regularization, and returns a mean using the same structure as $\mathsf{ev\_filtering}$, but with a modified pruning procedure. Rather than pruning points based on their projection onto the top eigenvalue, points are given outlier scores relevant to all directions. First, calculate the normalized matrix exponential $U = \exp(\alpha\Sigma)/\mathsf{tr}(\exp(\alpha\Sigma))$ where $\alpha \geq 0$ is a hyperparamater and $\Sigma$ is the sample covariance. Then, calculate a vector of quantum entropy scores, $w$, with $w_i = (x_i - \mu')^T U(x_i - \mu')$, where $x_i$ is the $i$th data point and $\mu'$ is the sample mean. This is implemented efficiently using a Chebyshev expansion of the matrix exponential and Johnson-Lindenstrauss approximations. Points with the largest scores are pruned, and the algorithm continues recursively with the remaining points until the top eigenvalue threshold is not violated. Following the original author's code (Dong et al., 2019)[4], we prune $\tau/2$ percentage of points during every iteration. Additionally, while the author's provide a theoretical threshold on the top eigenvalue, the constants are not given. Rather than tuning this threshold, we implement it using the same threshold as $\mathsf{ev\_filtering}$; that is $1 + 3\tau \log 1/\tau$. Because of this threshold, the algorithm critically assumes identity covariance and $n = \Omega(d/\tau^2) \gg d$. We set $\alpha = 4$ as in the author code; simple experiments show little variation with $\alpha$ between 0.5 and 200.

**PGD**: (Cheng et al. (2020)) Projected Gradient Descent, $\mathsf{PGD}$ for short, frames robust mean estimation as a non-convex optimization problem, and despite non-convexity, directly solves this using gradient descent. $\mathsf{PGD}$ finds a vector, $w$, of outlier scores, which can then be used to return a mean estimate $\mu' = \frac{1}{|X|}\sum_{x_i \in X} w_i x_i$. $w$ is found to minimize the spectral norm of the standard weighted covariance matrix, $\Sigma_w$, subject to the constraint that the weights represent at least a $(1-\tau)$-density fractional subset of the dataset. The vector $w$ is found as an approximate stationary point to this objective by first performing gradient descent on the spectral norm of the weighted covariance matrix, and then projecting onto the simplex of feasible weight vectors. First, define a function $F(u, w) = u^T \Sigma_w u$. Then, repeat the following for $\gamma$ iterations, where $\gamma$, following the conventions of a code implementation by the same author as the original paper (Cheng & Lin, 2021)[5], is a hyperparameter. Calculate the top eigenvector, $u_t$, of $\Sigma_w$, which corresponds to finding the unit vector $u_t$ such that $F(w, u_t) \geq (1-\tau)\mathsf{max}_u F(w, u)$. Then, update $w$ as $w = P(w - \alpha\nabla_w F(w, u_t))$ where $P$ projects onto $\Delta_{n,2\tau} = \{w \in \mathbb{R}^n : \|w\|_1 = 1 \text{ and } 0 \leq w_i \leq \frac{1}{(1-2\tau)n}\}$, and $\nabla_w F(w, u_t)) = Xu_t \odot Xu_t - 2(w^T Xu_t)Xu_t$ where $\odot$ indicates element-wise multiplication, and $\alpha$ is the learning rate, initialized as $1/n$ and updated dynamically through learning. We set the number of iterations $\gamma = 15$; this hyperparameter is explored in Appendix A.5.

$\ell_p\_\mathbf{min}$: (Deshmukh et al. (2022)) This method frames robust mean estimation as a semi-definite program (SDP). Similar to $\mathsf{PGD}$, a vector, $w$, of outlier scores is found, and the weighted mean $\mu' = \frac{1}{|X|}\sum_{x_i \in X} w_i x_i$ is returned. The $\ell_p$ norm for hyperparameter $0 \leq p \leq 1$ is maximized with respect to $w$, under the constraint that the top eigenvalue of the weighted covariance matrix is less than a constant. The weight vector $w$ is iteratively updated by solving a SDP until the number of iterations is less than a bound determined by $\tau$, in which case $\hat{\mu}$ defined above is returned. Update $w$ by approximately solving an SDP to maximize $w$ in $\|w\|_1$ over $\Delta_{n,\tau}$. Each step of the optimization problem is convex and can be solved as the following packing SDP:

$$\mathsf{max}_w \quad \text{s.t.} \quad w_i \geq 0 \; \forall i, \quad \sum_{i=1}^{n} w_i \begin{bmatrix} e_i e_i^T & \\ & (x_i - \mu_w)(x_i - \mu_w)^T \end{bmatrix} \preceq \begin{bmatrix} I_{n \times n} & \\ & c_\tau n I_{d \times d} \end{bmatrix},$$

where, $c_\tau$ is a function of $\tau$. This analysis of this algorithm critically assumes identity covariance and $n = \Omega(d/\tau^2) \gg d$.

---

[4] https://github.com/twistedcubic/que-outlier-detection
[5] https://github.com/chycharlie/robust-bn-faster

### 3.1 New Algorithms and Variants

We also consider a few new methods, with subtle but important extensions of these existing ones.

The primary insight needed to adapt methods to the $d \geq n$ case is found by revisiting how we identify outliers with respect to a $d$-dimensional Gaussian distribution. The bounds used in the $n \gg d$ case have enough data in each direction $d$ to concentrate, whereas in the $d \geq n$ case we need to account for this additional variance. The key result leverages a theorem of Vershynin (2011) to understand the concentration of the top eigenvalue of the sample covariance matrix.

**Theorem 1.** *Let $X$ be a $n \times d$ matrix whose entries are independently drawn from $\mathcal{N}(\mu, I)$. Let $\Sigma = \frac{1}{n}(X - \bar{\mu})^T(X - \bar{\mu})$ be the sample covariance matrix of $X$, where $\bar{\mu} = \frac{1}{n}\sum_i X_i$ and $X_i$ is the ith row of $X$. Then for every $t > 0$, with probability of at least $1 - 3\exp(-t^2/2)$, one has*

$$\|\Sigma\|_2 \leq \left(1 + \sqrt{d/n} + t/\sqrt{n} + \frac{\sqrt{d + \sqrt{2d}t + t^2}}{n}\right)^2.$$

The proof is deferred to Appendix A.1. A more convenient form shows that the fourth term is lower-order and can be absorbed into the probability of failure.

**Corollary 1.1.** *Under the same setting as Theorem 1, if one assumes $d/n \leq 16, n \geq 16, t \geq 5$, then with probability of at least $1 - 3\exp(-t^2/8)$, one has*

$$\|\Sigma\|_2 \leq \left(1 + \sqrt{d/n} + t/\sqrt{n}\right)^2.$$

**ev_filtering_low_n**: The ev_filtering algorithm assumes the sample size is $n = \Omega(d/\tau^2)$. This assumption is used in several parts of the analysis, and it allows the filtering bound to be simplified to $1 + \tau \log(1/\tau)$; however, when $n = o(d/\tau^2)$, this simplification does not hold, and the filtering bound needs to depend on $d$. We instead filter points if the top eigenvalue $\lambda_{\max} > (1 + \sqrt{d/n} + t/\sqrt{n})^2$ using Corollary 1.1. We set $t = 10$ to achieve almost $100\%$ ($\approx 0.999$) success. All other steps of the algorithm remain the same.

**QUE_low_n**: The QUE_low_n algorithm extends the same filtering bound as ev_filtering. Although their paper mentions a $O(\sqrt{d/n})$ factor in the error, the code seems to assume $n = \Omega(d/\varepsilon^2)$ and is implemented very similar to ev_filtering. As this approach does not work under low data size, in our newly proposed variant, we instead filter points if the top eigenvalue $\lambda_{\max} > (1 + \sqrt{d/n} + t/\sqrt{n})^2$ using Corollary 1.1.

**$\ell_p$_min_low_n**: The $\ell_p$_min algorithm uses the condition that the top eigenvalue of the weighted covariance matrix is bounded by $c_\tau n$ where $c_\tau$ is a hyperparameter suggested to be set at $1 + \tau \log(1/\tau)$. As previously observed, this threshold does not hold when $n = o(d/\tau^2)$ and to account for this, in our newly proposed variant we set $c_\tau = (1 + \sqrt{d/n} + t/\sqrt{n})^2$, using Corollary 1.1.

**lee_valiant_simple**: We use a simplified version of the Lee and Valiant algorithm (Lee & Valiant, 2022), which aligns with an informal description in their abstract. It completely removes the $\tau$ percentage of points classified as outliers rather than simply downweighting them. That is, it returns $\hat{\mu} = \frac{1}{|X'_*|}\sum_{x' \in X'_*} x$; the average of all points $X'_*$ which were not in the original estimate, nor from the pruned set furthest from $\mu'$.

## 4 Experiments

We generally evaluate the performance of these mean estimation algorithms as data size $n$, dimension $d$, and corruption $\eta$ are varied. Error is measured as the Euclidean distance $\|\mu - \hat{\mu}\|$ between the true mean $\mu$ and the estimate $\hat{\mu}$ returned by a mean estimation algorithm. We set the default values as $n = 500$, $d = 500$, and $\eta = 0.1$. We examine the performance as we fix one of these variables and vary the others under various distributions for both the uncorrupted and corrupted data. We first examine uncorrupted standard normal Gaussian data, demonstrating that nothing really improves upon sample_mean, and observing the robustness of mean estimation techniques when applied to uncorrupted data. We then examine corrupted Gaussian data

over various covariances and noise distributions. The example distributions are chosen among challenging examples in the literature meant to distinguish various models. Experiments were run on a 2022 Macbook Air with Apple M2 Chip, 16GB memory, running MacOS 12.

At one point in our experiment, the values of $n$, $d$, and $\eta$ are used to generate data according to a supplied data generation function and noise scheme (both of which will vary depending on the experiment). A mean estimate is made on this data using each of the mean estimators being tested. For each mean estimator, error is then stored as the Euclidean distance between the true mean of the data and the returned mean estimate. These errors are accumulated over 5 runs and averaged. We additionally plot the error incurred by the sample mean of the original uncorrupted data, which we call the good_sample_mean error. This serves as a valuable baseline for comparison. In practice, we can not expect to achieve error better than the sample mean of the inliers. Therefore, a reasonable goal for a robust estimator is to closely match the performance of good_sample_mean, thereby removing the effects of corrupted data points. Could a robust mean estimator somehow improve upon this? We do not observe this; but we will observe methods that basically match good_sample_mean, even without $n = \Omega(d/\varepsilon^2)$.

**Fraction of corrupted data.** Some algorithms are designed for data where a $\eta$-fraction of the data has been corrupted. And in some cases, this fraction is taken as a parameter $\tau$ used within the algorithms (coord_trimmed_mean, lee_valiant_simple, lee_valiant, ev_filtering, ev_filtering_low_n, QUE, QUE_low_n, PGD, $\ell_p$_min, $\ell_p$_min_low_n).

In theory, these algorithms work best using their parameter $\tau$ set to the true fraction of corrupted data $\eta$, and may even result in arbitrary error if the parameter $\tau$ is not set to at least an upper bound for the true fraction. However, increasing the value of $\tau$ in the algorithms also theoretically increases the error incurred by algorithms. Recent work by Jain et al. (2022) showed a meta algorithm that allows robust estimation algorithms to perform asymptotically optimal without knowing true corruption $\eta$. We investigate robustness to expected corruption, $\tau$, empirically, in Appendix A.6, as we fix $\tau$ and vary true corruption $\eta$. We observe that the best algorithms do not show a strong dependence on this relationship, so long as $\tau$ is an upper bound on $\eta$. Hence, for all other experiments, we simply set the parameter $\tau$ according to the true corrupted fraction $\eta$ or to $\tau = 0.1$ if the data is not corrupted.

**Selecting algorithmic variants.** There are many algorithms to be considered, and plots can become cluttered. To reduce this, we perform some comparison among variants. We summarize key findings here, with further details deferred to Section 7.

| Algorithm | $n = 500$, $d = 500$ | | $n = 200$, $d = 500$ | |
| --- | --- | --- | --- | --- |
| | **Error** | **Time (s)** | **Error** | **Time (s)** |
| sample_mean | $2.47 \pm 0.04$ | $0.00019 \pm 0.000002$ | $2.74 \pm 0.05$ | $0.00019 \pm 0.000001$ |
| LRV | $1.14 \pm 0.04$ | $0.81 \pm 0.12$ | $1.76 \pm 0.10$ | $0.64 \pm 0.03$ |
| PGD | $1.08 \pm 0.02$ | $82.4 \pm 8.8$ | $1.68 \pm 0.05$ | $72.5 \pm 3.2$ |
| ev_filtering_low_n | $1.07 \pm 0.02$ | $0.20 \pm 0.02$ | $1.69 \pm 0.04$ | $0.08 \pm 0.03$ |
| ev_filtering | $13.49 \pm 3.56$ | $0.48 \pm 0.15$ | $17.06 \pm 5.92$ | $0.05 \pm 0.02$ |
| QUE_low_n | $1.04 \pm 0.03$ | $0.71 \pm 0.05$ | $1.70 \pm 0.049$ | $0.35 \pm 0.03$ |
| QUE | $20.81 \pm 0.40$ | $2.70 \pm 0.08$ | $20.88 \pm 0.38$ | $1.99 \pm 0.04$ |
| $\ell_p$_min_low_n | $1.17 \pm 0.04$ | $1182.6 \pm 35.3$ | $1.67 \pm 0.03$ | $265.9 \pm 15.2$ |
| $\ell_p$_min | $1.62 \pm 0.04$ | $1076.8 \pm 43.9$ | $5.62 \pm 0.40$ | $250.07 \pm 17.2$ |

Table 1: Error and Runtime Across Simple Corrupted Identity Covariance Gaussian

First, we do not consider $\ell_p$_min and $\ell_p$_min_low_n in our plots due their exceptionally large runtimes. We report runtimes and errors (defined as the Euclidean distance from the estimated mean to the true mean) of selected algorithms under $n = 500$ and $d = 500$ and under $n = 200$ and $d = 500$ over a simple corrupted Gaussian distribution in Table 1. We report results as the mean $\pm$ the standard deviation, averaged over 5 runs. With the notable exceptions of $\ell_p$_min_low_n, $\ell_p$_min, and PGD, most estimators are efficient and took under 3 seconds to run with $n = 500$ and $d = 500$ for a simple corrupted Gaussian distribution. $\ell_p$_min

and $\ell_p$_min_low_n rely on an SDP solver, which we implement with the cvxpy (Diamond & Boyd, 2016; Agrawal et al., 2018) package and the mosek solver. Although this is theoretically efficient, it is slow in practice for the data scale and dimesionality we consider in this paper. For $n = 500$ and $d = 500$, both algorithms took about 1100 seconds, or about 18 minutes, to return a mean estimate over a simple corrupted data scheme. For that reason, and since we run many trials of each input size and error level, we do not consider these algorithms in our plots. However, we note that employing Corollary 1.1 for $\ell_p$_min_low_n achieves a noticeable performance increase over $\ell_p$_min; showing gains from 1.62 error to 1.17 error in the $n = 500, d = 500$ case and from 5.62 to 1.67 error in the $n = 200, d = 500$ case. PGD is also much slower than other robust estimators, taking approximately 80 seconds to run with $n = 500$ and $d = 500$. While this significant slow down is relevant when considering a practical algorithm, it is not as prohibitive as $\ell_p$_min. As a result, we include it in all of our plots.

Second, we observe that when the data does not satisfy that $n \gg d$, then both ev_filtering and QUE can have catastrophic failure. Our variants ev_filtering_low_n and QUE_low_n avoid this issue in the $d > n$ and $d \approx n$ settings, while basically matching the effectiveness of their original versions when they do not have catastrophic failure. This result is highlighted in Table 1, where we observe that both QUE and ev_filtering achieve significantly worse error than QUE_low_n and ev_filtering_low_n respectively. As a result, we use ev_filtering_low_n and QUE_low_n in all comparisons.

Thirdly, we find that lee_valiant_simple performs slightly better than the original lee_valiant; however the difference is fairly small. We also do not notice any meaningful advantages from using lee_valiant_simple or lee_valiant with different choices of initial mean estimators. As such, we only use lee_valiant_simple in all comparisons.

## 4.1 Uncorrupted Gaussian Data with Identity Covariance

We first evaluate the performance of mean estimation algorithms over uncorrupted Gaussian data with identity covariance. In particular, we draw uncorrupted data $X \sim \mathcal{N}_d(\mu, I)$, where $\mu$ is an arbitrary mean and $I$ is identity covariance. For these experiments, we set $\mu$ to be the all-fives vector, but did not find performance to depend on $\mu$. For algorithms that utilize $\tau$, expected corruption, as input, we use the default value of $\tau = 0.1$.

We provide our first experimental plots in Figure 1; most further experiments will follow this same set-up, consisting of a set of 4 charts, each measuring the Error $\|\mu - \hat{\mu}\|$ on the $y$-axis. The top two charts vary the data size $n$ along the $x$-axis, but on different scales. The top left shows a large scale from $n = 20$ to $n = 5020$, focusing on the $n > d = 500$ paradigm. The top right shows $n = 20$ to $n = 520$, focusing on the $n < d$ paradigm. The bottom left plot show the effect of varying the dimension from $d = 20$ to $d = 1020$ while fixing $n = 500$. The bottom right shows varying the algorithm's parameter, $\tau$, for the expected noise from 0 to 0.45 with fixed $n = 500$, $d = 500$. Each algorithm is shown as a curve, with the average error of 5 independent data generations at regular intervals on the $x$-axis. A shaded area is shown at a radius of 1 standard deviation from that average error value.

The plots are a bit cluttered because most algorithms perform about the same, including sample_mean. No algorithm can be seen to noticeably outperform sample_mean, which, as the MLE for this data, and by the Gauss-Markov theorem, is not surprising. Methods LRV, ev_filtering_low_n, QUE_low_n, PGD, coord_trimmed_mean, geometric_median, and lee_valiant_simple have about the same error in most cases. However, median_of_means, and coord_median perform slightly worse, with the gap becoming more apparent in high dimensions. Moreover, lee_valiant_simple and coord_trimmed_mean do significantly worse with a higher expected corruption parameter $\tau$. This is a result of expected corruption, $\tau$, being a hyperparameter that directly controls the percentage of points to prune. Finally, as predicted by basic theory, with $n$ fixed as the dimension $d$ increases, the measured error increases at a rate roughly $\sqrt{d}$.

## 4.2 Corrupted Gaussian Data with Identity Covariance

We evaluate corrupting noise added to Gaussian data with identity covariance. In particular, we draw $X \sim (1 - \eta)P + \eta Q$ where $P = \mathcal{N}_d(\mu, I)$ and $Q$ describes the corrupted data distribution. This is equivalent

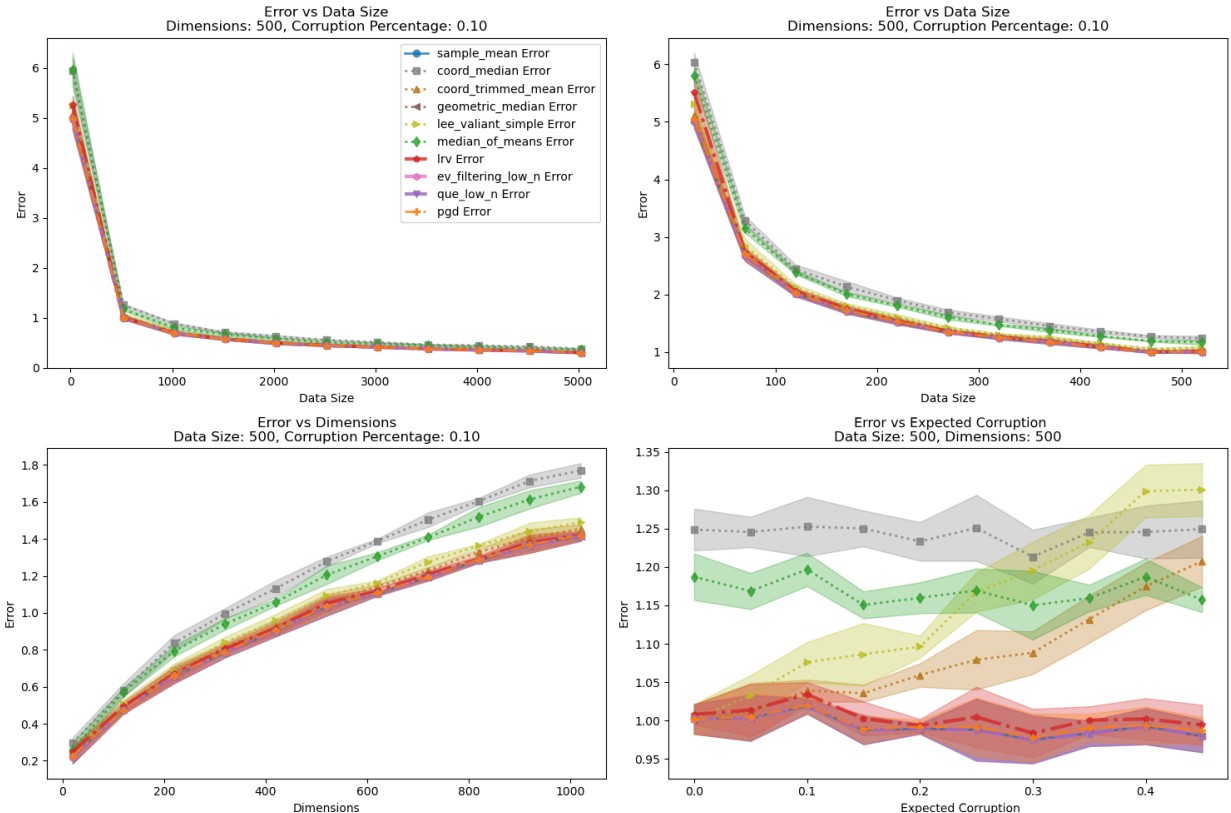

Figure 1: Uncorrupted Gaussian Identity Covariance

to the more general case where any covariance $\Sigma$ is known, as we could simply scale the data to have identity covariance, apply these methods, and scale the mean estimate back. We provide a wrapper in our implementation to perform this operation.

**Gaussian noise shifted to variance shell.** We first consider corrupted data distribution $Q = \mathcal{N}_d(\mu', \frac{1}{10}I)$ so $\|\mu - \mu'\| = \sqrt{d}$. Since $\mathbb{E}_{x \sim P}[\|x - \mu\|^2] = d$, corrupted data from $Q$ is not easily identified. The location of this cluster is determined by a random rotation at every generation to ensure that no coordinate-axis specific bias is introduced. This is shown in Figure 2 in the same 4 experiments as with uncorrupted data, except now the bottom right figure varies $\eta$, the fraction of corrupted data from $Q$, along the $x$-axis. We set the expected corruption hyperparameter equal to true corruption, that is $\tau = \eta$. In Appendix A.6 we explore the relation between expected corruption $\tau$ versus actual corruption $\eta$; for the most part as long as $\tau > \eta$.

There is now more clear separation between the algorithms designed for adversarial corruption, and those not. Here ev_filtering_low_n, QUE_low_n, and PGD do the best among all settings, with LRV, perhaps doing the best, even appearing better than good_sample_mean for large dimensions, although within 1 standard deviation error margin. Due to the high dimensionality, $d$, good_sample_mean, the sample mean of points from the uncorrupted part of the distribution $P$, does not have error approaching 0 until $n$ is very large. ev_filtering_low_n, QUE_low_n, and PGD work so well that they are nearly overlapping this best possible standard. Also, perhaps surprisingly, median_of_means also does nearly as well, especially under larger $n$, though it degrades much worse with larger $\eta$.

In contrast, coord_median, sample_mean, coord_trimmed_mean, geometric_median, and lee_valiant_simple all do considerably worse, even with large data size. With large corruption levels, these all even seem to do worse than just sample_mean, indicating that the algorithms prune the wrong data points or face some other similar issue.

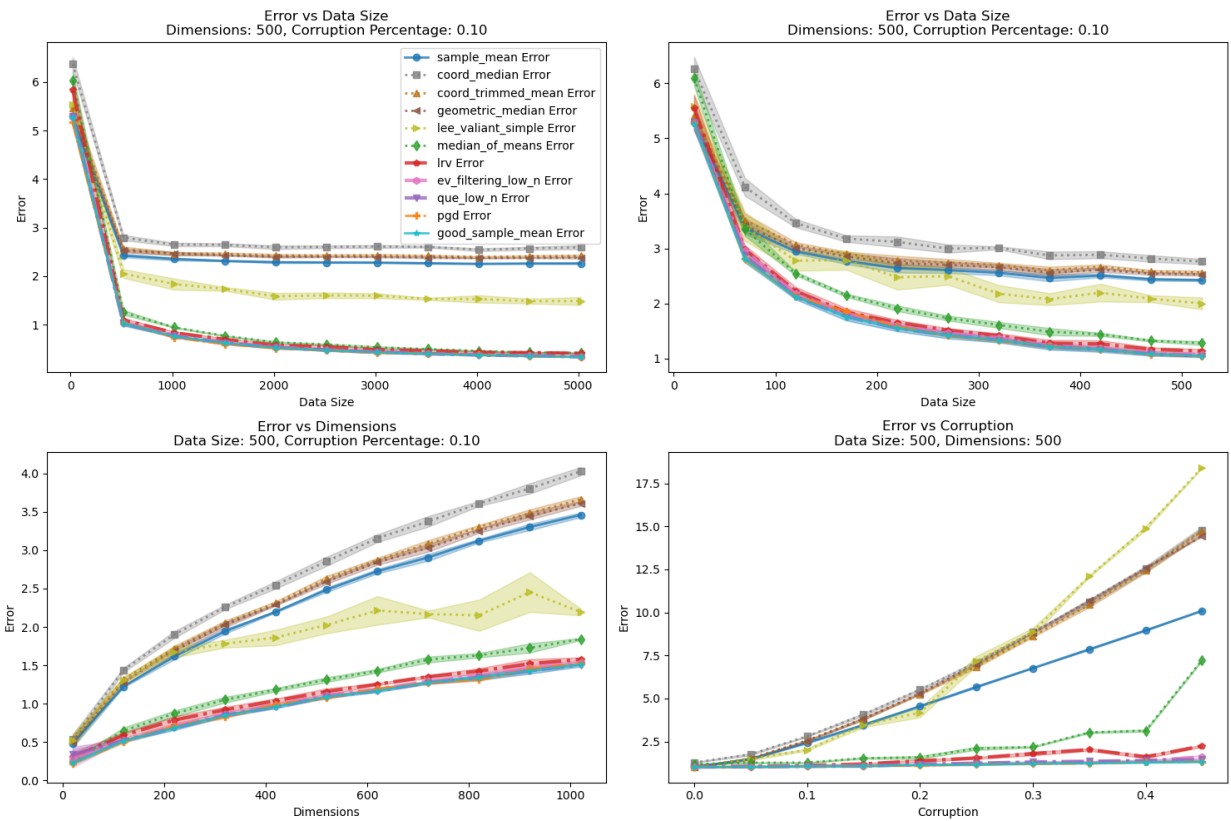

Figure 2: Corrupted Gaussian Identity Covariance: Additive Variance Shell Noise

**Large + Subtle outliers: DKK Noise.** We now recreate the noise distribution from Diakonikolas et al. (2017a), which utilizes a more sophisticated corruption scheme that includes both easier and harder to detect outliers. Half of the noise is drawn from the product distribution over the hypercube where every coordinate is -1 or 0 away from the true mean at that coordinate with equal probability. The other half is drawn from the product distribution where the first coordinate is either 11 or -1 away from the true mean at that coordinate with equal probability, the second coordinate is -3 or -1 away from the corresponding true mean coordinate with equal probability, and all remaining coordinates are -1 away from the true mean. We call this corruption scheme DKK Noise. This is shown in Figure 3, with similar results. ev_filtering_low_n, QUE_low_n, PGD, and LRV achieve performance nearly matching good_sample_mean, with median_of_means also doing almost as well – at least while the dimension $d$ and rate of corruption $\tau$ are on the smaller side. Other than median_of_means, all classic methods perform noticeably worse than good_sample_mean and achieve similar error to sample_mean. The only difference of note here is that lee_valiant_simple exhibits far larger error bars, suggesting that its performance may vary significantly depending on random initializations made within the algorithm. Also, LRV may even outperform good_sample_mean for very large dimensions.

**Subtractive noise.** We additionally consider subtractive noise in Figure 4. Here, an adversary is able to remove a $\eta$ percentage of points from the data distribution. We implement this by removing the $\eta$-percentage of points which are most extreme in some direction. Unlike in the additive corruption case, there is a strict upper bound on the error under subtractive corruption from a standard Gaussian distribution; the error induced is bounded as $O(\eta)$ even using sample_mean, and clustering the subtracted points as most extreme in some direction ensures their effect is $\Omega(\eta)$ under sample_mean. In general, we wish to consider noise distributions that may add outliers and also remove inliers through such subtractive noise. However, we do not find any surprising capabilities among methods in this scenario. As a result, for the remainder of this paper, we focus on additive corruption.

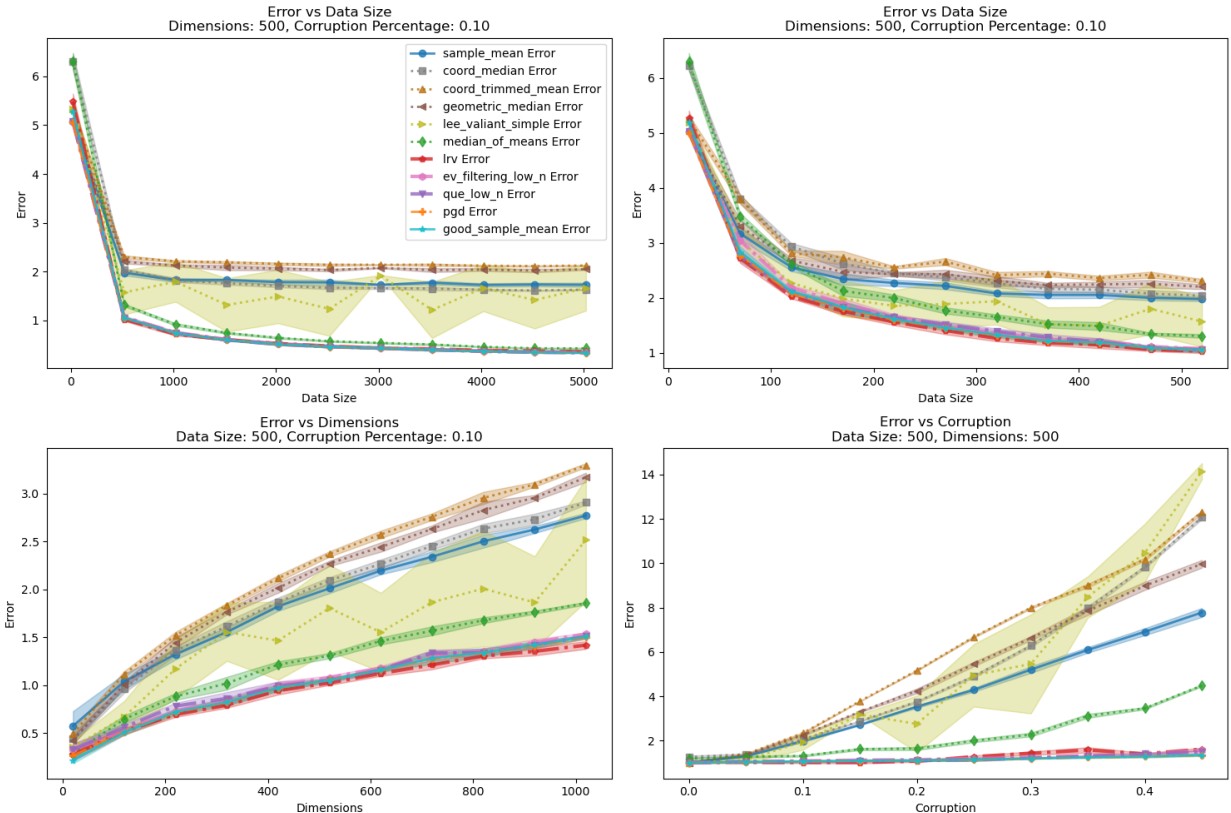

Figure 3: Corrupted Gaussian Identity Covariance: DKK Noise

Under subtractive corruption, nothing outperforms sample_mean; and now nothing can match good_sample_mean in error in most settings. However, ev_filtering_low_n, QUE_low_n, PGD, and LRV all nearly match the performance of sample_mean. Unlike in the previous additive corruption schemes, median_of_means performs significantly worse under subtractive corruption, always achieving error notably worse than sample_mean. Among other estimators, geometric_median nearly matches sample_mean error across all settings, lee_valiant_simple and coord_trimmed_mean perform similarly but degrade much more under larger corruption, while coord_median performs significantly worse.

We find similar results across several other noise distributions. In addition to the hard-to-detect distributions, we also show that ev_filtering_low_n, QUE_low_n, PGD, and LRV are generally robust to arbitrary outliers. These details are deferred to Appendix A.2.

## 4.3 Corrupted Gaussian Data with Unknown Covariance

We now evaluate corrupted Gaussian data for general unknown covariance. Since ev_filtering_low_n and QUE_low_n rely on the identity covariance assumption, we employ a simple heuristic to adapt these algorithms to the unknown covariance case. We estimate the trace of the covariance as $\mathrm{Tr}(\hat{\Sigma}) = \frac{1}{n-1}\sum_{i=1}^{n}||x_i - \hat{\mu}||^2$, where $\hat{\mu}$ is the sample mean and $\hat{\Sigma}$ is the sample covariance. We rescale the data to $X' = \{x_i' = x_i/\sqrt{\frac{\mathrm{Tr}(\hat{\Sigma})}{d}} \mid x_i \in X\}$. We then estimate the mean of $X'$, rescale this estimate by $\sqrt{\frac{\mathrm{Tr}(\hat{\Sigma})}{d}}$, and report the results. This heuristic is used for ev_filtering_low_n and QUE_low_n across all unknown covariance experiments, and not used for any other algorithms. The other standard algorithms are either invariant to this linear rescaling, or themselves account for it; this was supported by our own observations.

Another possible method is to utilize a robust covariance estimate instead of a sample trace estimate, as discussed in Diakonikolas & Kane (2023). We do not evaluate such methods, as this would involve a thorough

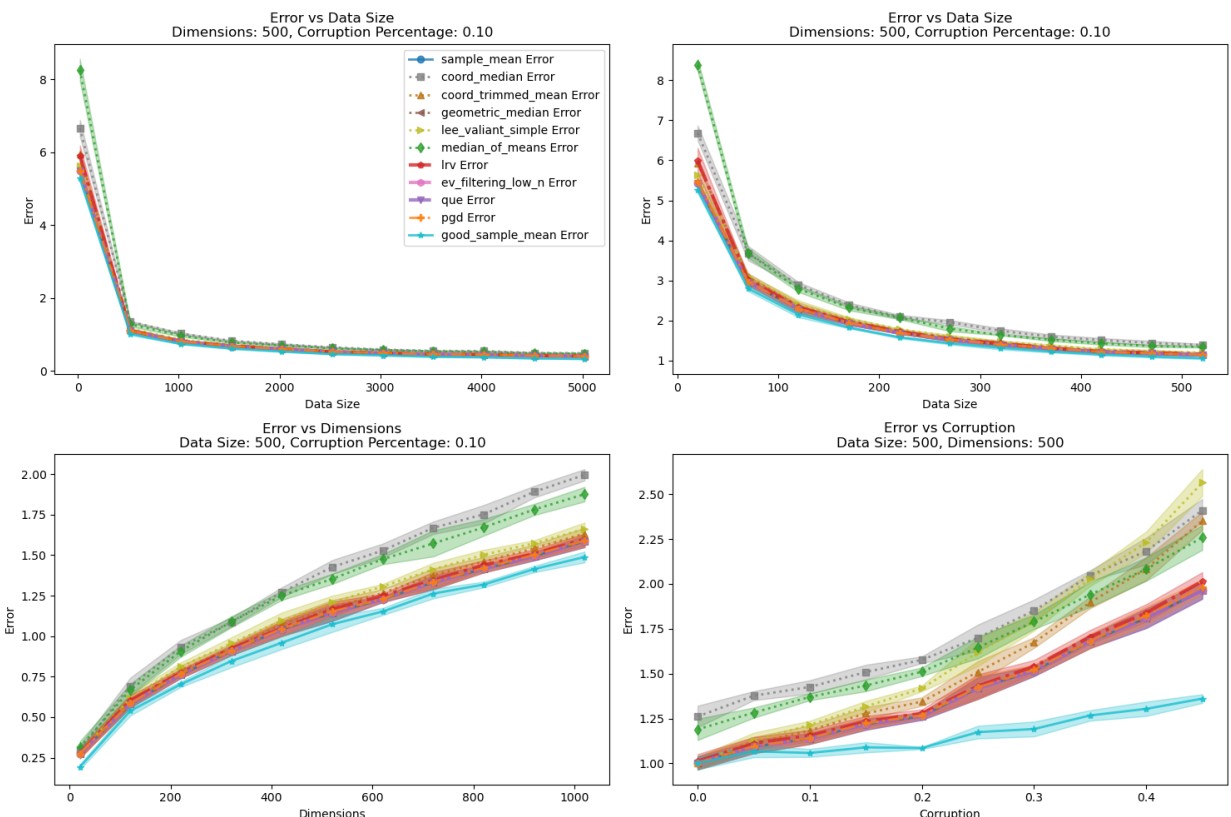

Figure 4: Corrupted Gaussian Identity Covariance: Subtractive Noise

study into robust covariance estimation methods over low data size, which goes beyond the scope of this work. Naively treating robust covariance estimation as robust mean estimation in $d^2$ dimensions further exasperates issues related to low data size. We also choose to use a simple sample trace estimate rather than the robust approach proposed by Lai et al. (2016). We find that the approach proposed often results in significant underestimates across difficult noise distributions, causing ev_filtering_low_n and QUE_low_n to fail catastrophically. We note that these underestimates are potentially more harmful than overestimates as through them, even the inlier data may not pass the threshold, causing continuous pruning. While a sample trace estimate approach is more prone to overestimates, this can be remedied by naively pruning large outliers. Diakonikolas et al. (2017a) also provides an algorithm for unknown covariance mean estimation similar to ev_filtering, but the corruption detection threshold is not easily adapted to the low data size case.

### 4.3.1 Unknown Spherical Covariance

We evaluate corrupting noise added to Gaussian data with spherical covariance. We draw $X \sim (1-\eta)P + \eta Q$ where $P = \mathcal{N}_d(\mu, \sigma^2 I)$ and $Q$ describes the additive corrupted data distribution. We consider $\mu$ to be the all-fives vector and $\sigma = 5$.

**Gaussian noise shifted to scaled variance shell** We adapt the identity covariance noise distribution models by appropriately scaling coordinates by $\sigma$. We first consider the corrupted data distribution $Q = \mathcal{N}(\mu', \frac{1}{10}I)$ so $\|\mu - \mu'\| = \sigma\sqrt{d}$. With $P$ now having covariance $\sigma^2 I$, $\mathbb{E}_{x \sim P}[\|x - \mu\|^2] = \sigma^2 d$, and corrupted data from $Q$ is not easily identified. Results are show in Figure 5.

While the overall error here is higher, matching the theory that even for uncorrupted data, the sample mean is expected to have error of $O(\sigma\sqrt{d/n})$, the relative performance of algorithms is nearly identical to the identity covariance case. ev_filtering_low_n, QUE_low_n, and PGD nearly match good_sample_mean error

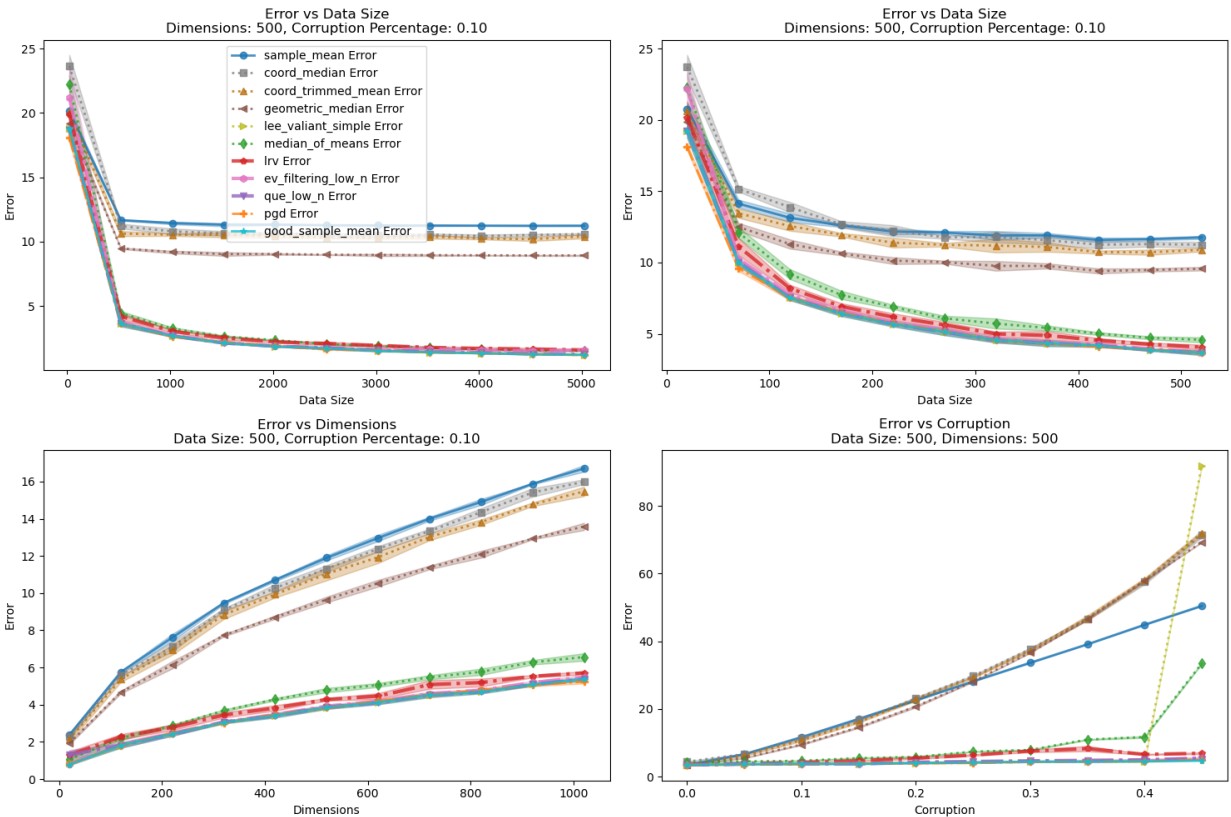

Figure 5: Corrupted Gaussian Large Spherical Covariance: Additive Variance Shell Noise

throughout, with LRV performing only slightly worse. median_of_means lags behind both estimators but still performs noticeably better than sample_mean. However, lee_valiant_simple performs much better in this scenario, nearly exactly matching good_sample_mean except with large enough corruption – where with $\eta = 0.45$, it and median_of_means probably confuse which points are inliers and have much worse error. Other methods perform similarly to sample_mean or worse.

As in the identity covariance case, we find similar results across noise distributions. The only notable exception is for ev_filtering_low_n, which sometimes performs slightly worse and doesn't always converge to good_sample_mean error as $n$ increases, probably due to instabilities in the trace scaling heuristic. We additionally show that relative performance of algorithms is mostly independent of the choice of $\sigma$. The exception to this is LRV, which notably outperforms all other methods, including good_sample_mean, with large enough $\sigma$ across noise distributions. These details are deferred to Appendix A.3.

**Unknown Non-Spherical Covariance**  In Appendix A.4 we also explore the unknown, non-spherical covariance case. This is even more sensitive to the covariance estimate, and so is further outside the primary scope of this study. Nonetheless, we continue to observe that the best robust estimators, including QUE_low_n, continue to perform well.

## 5   Large Language Model Experiment

To evaluate whether robust mean estimation methods are overly sensitive to distributional assumptions, we evaluate performance over real world data. We first study the problem of estimating the mean of vectors from language models. Such "word embeddings" have had an enormous impact on natural language processing, starting from simple sparse term-frequency vectors (Robertson et al., 2009). Second generation word embeddings (e.g., GloVE (Pennington et al., 2014) and word2vec (Mikolov et al., 2013)) made the

advancement of creating a "low" dimensional vector (about 300 dimensions) for each word, where Euclidean (and cosine) distances could be used as a proxy for the similarity between words based on how they are used. As a serendipitous side-effect, structure emerged where dot-products, means, and linear classifiers made sense in this embedding space (Bolukbasi et al., 2016; Dev & Phillips, 2019). Third generation embeddings created representations for each word in the context of the nearby words; that is, each use of a word had a different embedding. These were a first main use of transformer architectures, and implicitly capture more meaning and context with progressive layers of a neural network. As is most common, we use the last layer of the embedding network as the representation of a word. Our first study, shown next, uses these third generation word embeddings. Further experiments over deep pretrained image model embeddings and second generation word embeddings are deferred to Appendix A.7 and Appendix A.8, respectively.

We first examine performance over third generation embeddings of a homonym word where all instances correspond to the same meaning. We then examine performance where embeddings corresponding to one meaning of a word are corrupted by embeddings corresponding to another meaning of the same word. Effectively calculating this mean may be important to many downstream tasks (e.g., topic modeling (Griffiths & Steyvers, 2004; Blei & Lafferty, 2009), bias estimation and attenuation (Bolukbasi et al., 2016; Dev & Phillips, 2019)). This models a realistic form of corruption that may arise within LLM embedded data.

We build a dataset of 400 sentences that use the word "field" corresponding to the following definition: "an area of open land, especially one planted with crops or pasture, typically bounded by hedges or fences". We build another dataset of 400 sentences that use the word "field" corresponding to the following, alternate definition: "a particular branch of study or sphere of activity or interest". From now on, we refer to these as "fields of land" and "fields of study". We generated these sentences using ChatGPT-4o. For more details on how we generate this dataset, and the exact sentences used, see Appendix A.10. We embed these sentences and extract the in-context embeddings for the word "field" using 4 LLMs of varying embedding dimensions: MiniLM (Wang et al., 2020), T5 (Raffel et al., 2023), BERT (Devlin et al., 2019), and ALBERT (Lan et al., 2020). MiniLM has an embedding dimension of 384, T5 has one of 512, BERT and ALBERT have embedding dimensions of 768. We choose these 4 LLMs to sample a variety of models across different dimensionalities.

## 5.1 Common Definition Embeddings

We first consider performance over embeddings corresponding to the same definition. This is analogous to the uncorrupted data case. As an error metric, we use Leave One Out Cross Validaton (LOOCV). We only average over the bottom 90% of errors to account for potential bias introduced by words that less clearly belong to a specific category. LOOCV error is defined here as $\frac{1}{n'}\sum_{i=1}^{n'}\|\mathsf{estimator}(X_{-i}) - x_i\|$ where $n' = 0.9n$ is 90% of the number of data points in the dataset $X$ (those with smallest errors), $x_i$ is the $i$th data point in $X$, and $X_{-i}$ is $X$ excluding $x_i$. LOOCV under the sample mean represents a valuable baseline for comparison as it demonstrates the minimum error to be expected across this data set under the 10% expected corruption ($\tau = 0.1$) modeled by the algorithms. We take the dataset of 400 sentences corresponding to the "field of land" definition. We vary data size from $n = 10$ to $n = 400$, and, as in prior experiments, average results over 5 runs and report shaded regions to denote 1 standard deviation of error. For algorithms that utilize $\tau$, expected corruption, as input, we use the default value of $\tau = 0.1$. We employ the sample trace scaling heuristic for ev_filtering_low_n and QUE_low_n. We additionally halt QUE_low_n whenever more than $2\tau$ percentage of the data has been pruned, regardless of whether or not the threshold is passed. We note that the early halting heuristic is necessary for QUE_low_n to perform well under this setting and that it does not meaningfully improve ev_filtering_low_n; this is further explored in Section 5.3. We show results in Figure 6.

Our results do not match our synthetic experiments, suggesting that some robust mean estimation algorithms are sensitive to (Gaussian) distributional assumptions, at least under small data size. Across LLMs, no algorithm significantly beats the error of sample_mean. Moreover, ev_filtering_low_n performs significantly worse than sample_mean over all embeddings. This is unsurprising due to the sensitivity of ev_filtering_low_n to knowledge of the true covariance. Despite having a similar dependency to knowledge of the true covariance, QUE_low_n achieves performance nearly matching sample_mean error throughout. We also find, that LRV performs meaningfully worse than sample_mean across all LLMs except MiniLM, though not quite

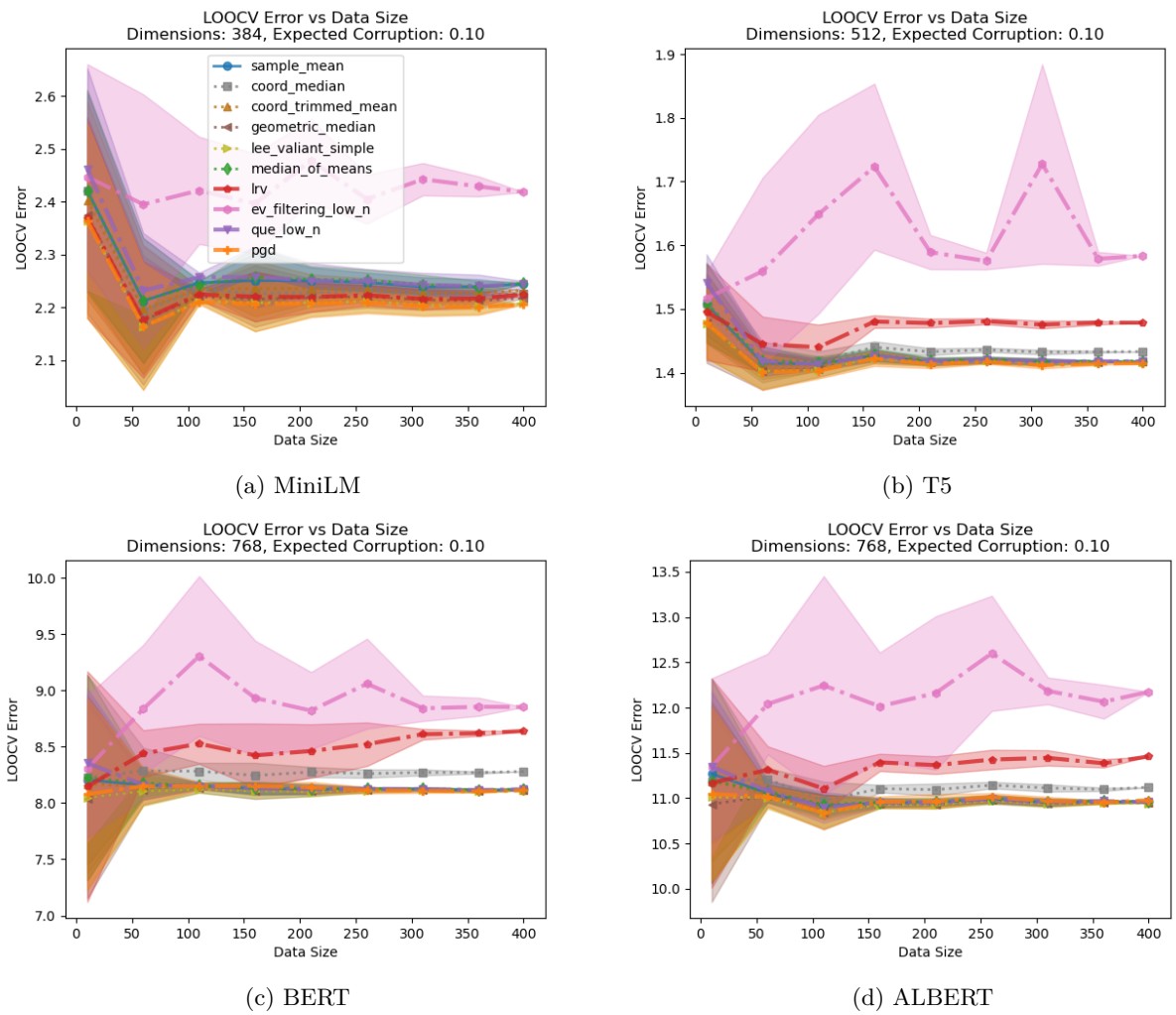

Figure 6: LOOCV Error on "Field of Land" Embeddings

as catostrophically as ev_filtering_low_n. Aside from coord_median, which performs noticeably worse than sample_mean over all LLMs besides MiniLM, all other estimators perform similarly to sample_mean.

## 5.2 Corrupted Embeddings

We examine performance over corrupted embeddings of the word "field". We draw corrupted data $X \sim (1 - \eta)P + \eta Q$, where the inlier distribution, $P$, consists of embeddings of the word "field" corresponding to the "field of land" definition, and the outlier distribution, $Q$, consists of embeddings of the word "field" corresponding to the "field of study" definition. As with previous experiments, we measure the Error $\|\mu - \hat{\mu}\|$ on the $y$-axis, taking $\mu$ as the mean of all 400 "field of land" embeddings, and $\hat{\mu}$ as the estimate returned by a mean estimation algorithm. We measure Error vs $\eta$, vary $\eta$ from 0 to 0.45, and always have $n = 400$. We average results over 5 runs and report shaded regions to represent 1 standard deviation of error. good_sample_mean is plotted to represent the mean of the data before corruption. These results are shown in Figure 7.

We find that mean estimation algorithms can indeed significantly improve performance on this real-world task, but do not observe the same trends as in our synthetic data experiments. QUE_low_n and lee_valiant_simple are the best estimators, with both significantly outperforming sample_mean. In fact, QUE_low_n performs nearly identical to good_sample_mean across all LLMs, and lee_valiant_simple per-

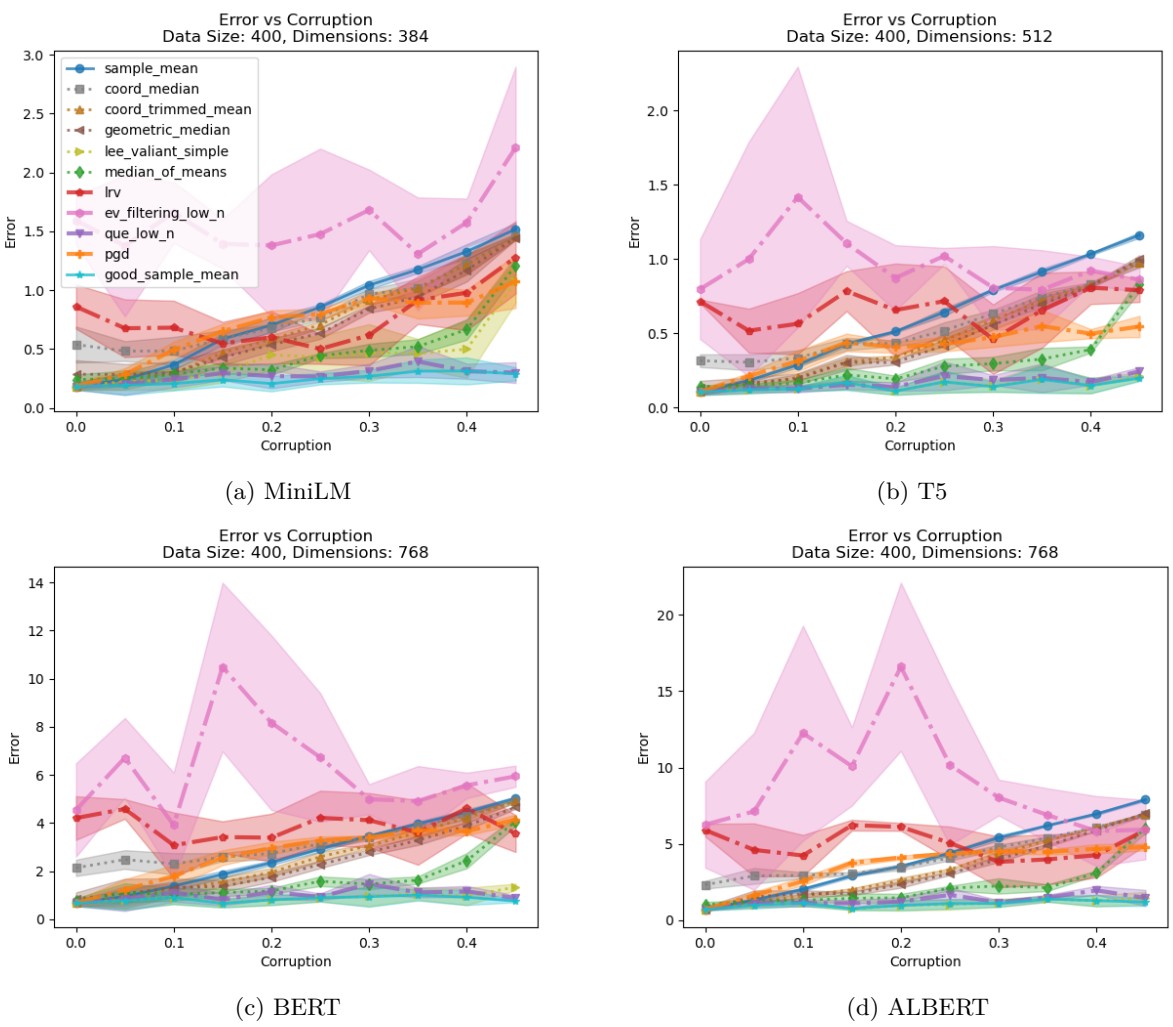

Figure 7: Error on "Field of Land" Embeddings Corrupted with "Field of Study" Embeddings

forms similarly, except on MiniLM, where it degrades worse with larger $\eta$ but still outperforms sample_mean. The performance of lee_valiant_simple supports the observation that it is a more effective naive pruning method, which happens to work among the best in these experiments. Moreover, median_of_means performs very effectively here, always significantly outperforming sample_mean. However, neither ev_filtering_low_n nor LRV perform effectively. LRV outperforms sample_mean with large enough $\eta$, but these results are not consistent across LLMs suggesting sensitivity to distributional assumptions. Additionally, it almost never outperforms median_of_means, lee_valiant_simple, or QUE_low_n and achieves much worse error with lower $\eta$. This may suggest that LRV finds irregularities in the uncorrupted data, causing it to return a mean significantly different from good_sample_mean. While this could be beneficial, suggesting that good_sample_mean isn't the best error metric, this is not supported by the "uncorrupted" LOOCV error results, where LRV also generally results in degraded LOOCV error. ev_filtering_low_n fails catastrophically across LLMs. This matches the "uncorrupted" LOOCV error results, supporting the observation that ev_filtering_low_n may fail catostrophically without sufficient knowledge of the true covariance matrix. As in the "uncorrupted" LOOCV error results, there is the somewhat surprising observation that despite seemingly having a similar dependency on knowledge of the true covariance matrix to ev_filtering_low_n, QUE_low_n performs nearly optimally here. This suggests the superiority of the quantum entropy based scoring method over naively ranking outliers based on the top eigenvalue of the sample covariance.

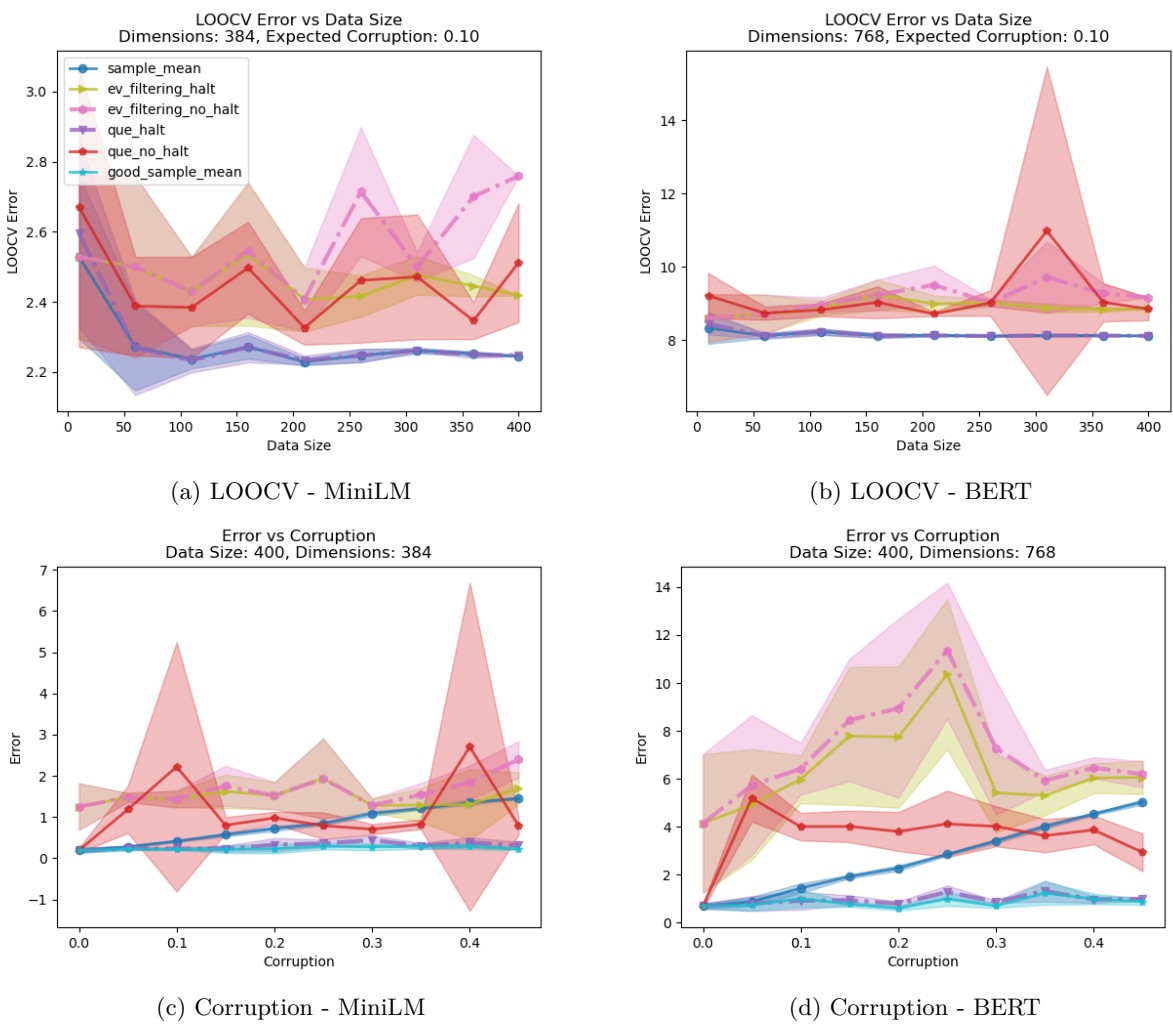

Figure 8: LLM Comparison - With and without *early halting*

## 5.3 Effect Of Early Halting and Ablation

*Early halting* is the following strategy with respect to a given threshold $\tau$: If more than $2\tau$ points have been pruned by an algorithm, then this halts the pruning process (independent of other criteria) and returns the sample mean of remaining data.

Here we examine the effect of early halting on QUE_low_n and ev_filtering_low_n in the context of these real world data where inliers are not generated directly from a prescribed Gaussian distribution. In other settings explored in this paper, this strategy is almost never invoked, so has no visible effect. We compare the performance of both of these algorithm with and without enforcing early halting. We examine LOOCV and Corruption Error over MiniLM and BERT embeddings. Results are shown in Figure 8. Across all 4 experiments the performance of QUE_low_n shows significant degradation without early halting, going from nearly matching sample_mean in LOOCV error and nearly matching good_sample_mean in corrupted error, to yielding error significantly worse than sample_mean and good_sample_mean without early halting. Additionally, QUE_low_n without halting yields far larger variance results. Meanwhile, ev_filtering_low_n performs only slightly better with early halting, and still fails catastrophically across experiments.

We perform further ablations on these LLM experiments in Appendix A.9. We explore the effect of different pruning methods on ev_filtering_low_n and different weighting methods on LRV. In both cases, we find that

LOOCV error can be improved using non-Gaussian pruning and weighting methods, whereas corrupted error is not meaningfully improved. The failure of ev_filtering_low_n across pruning methods suggests the fundamental sensitivity of the outlier scoring method used in ev_filtering_low_n to distributional assumptions, which is not seen in QUE_low_n (with early halting). We additionally examine performance over these same experiments with the roles of "field of study" and "field of land" embeddings flipped, finding nearly identical results despite differences in distribution.

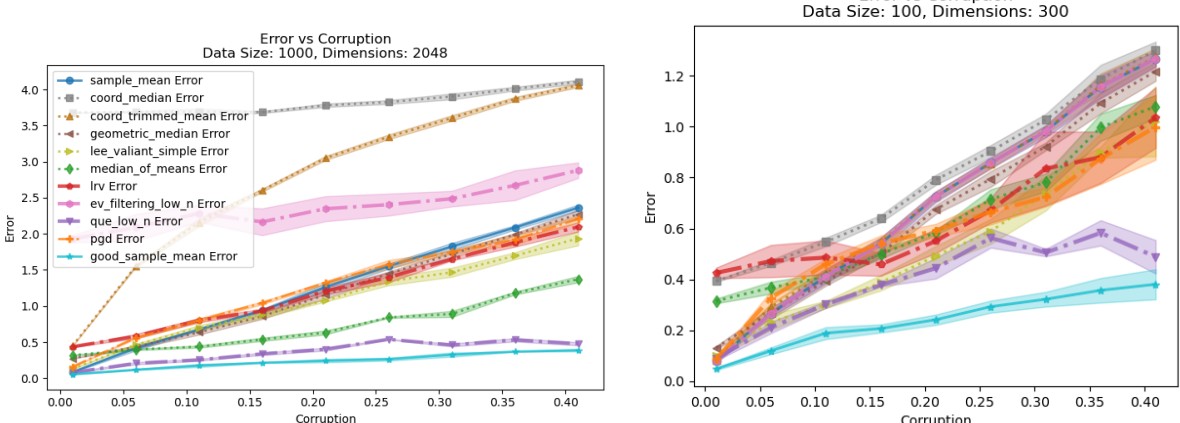

(a) ResNet-50 2048 Dimensional Image Embeddings: Cat Images Corrupted With Dog Images

(b) GloVe 300 Dimensional Word Embeddings: Pleasant Words Corrupted With Unpleasant Words

Figure 9: Additional Real World Corrupted Experiments

## 5.4 Additional Real World Experiments

We additionally examine the performance of robust mean estimation algorithms on corrupted embeddings from deep pretrained image models and non-contextual word embedding models. For the image embedding experiment, we utilize a set of images of cats and dogs from the CIFAR10 dataset (Krizhevsky, 2009) with 2048 dimensional embeddings generated from a pretrained ResNet-50 model (He et al., 2015). For the word embedding experiment, we utilize a dataset of pleasant and unpleasant words from (Aboagye et al., 2023) with 300 dimensional embeddings generated from a pretrained GloVe model (Pennington et al., 2014). Experiments are run analogously to the LLM experiments with identical settings for the mean estimators. For the image embedding experiment, inlier data is defined as embeddings of cat images, outlier data is defined as embeddings of dog images, and data size is fixed at $n = 1000$. For the word embedding experiment, inlier data is defined as embeddings of "pleasant" words, outlier data is defined as embeddings of "unpleasant" words, and data size is fixed at $n = 100$. Results are shown in Figure 9.

We find similar results to the LLM experiments, with QUE_low_n using early halting noticeably outperforming sample_mean across both settings, and nearly matching good_sample_mean over image embeddings. Other robust estimators tend to perform similarly or worse to sample_mean, with ev_filtering_low_n again demonstrating significant degradation without knowledge of distributional assumptions. Additionally, lee_valiant_simple, which performs strongly in the LLM experiments, does not perform as well, demonstrating its sensitivity to distributional assumptions. Similarly, median_of_means, does not perform as well across word embeddings as it does across LLM and image embeddings, no longer noticeably outperforming other estimators.

We find similar results across varying dimensionalities of image embedding and GloVe models. We examine additional image embeddings models of varying dimensionalities under LOOCV and corrupted error in Appendix A.7. We additionally recreate the corrupted data experiment, but vary data size instead of corruption, finding that even with $n \gg d$, only QUE_low_n and median_of_means significantly outperform sample_mean error, with both estimators nearly converging to good_sample_mean error with corruption

$\eta = 0.1$. We also examine GloVe models of varying dimensionalities under LOOCV and corrupted error in Appendix A.8.

## 6 Non-Gaussian Synthetic Data Experiments

We examine the performance of robust mean estimators across a few non-Gaussian synthetic data experiments. As before, we draw $X \sim (1 - \eta)P + \eta Q$, where $P$ is an inlier data distribution and $Q$ is the corrupted data distribution. Similar to real world experiments, we employ the trace scaling heuristic on `ev_filtering_low_n` and `QUE_low_n` and enforce early halting on `QUE_low_n` if more than a $2\tau$ percentage of the data has been pruned.

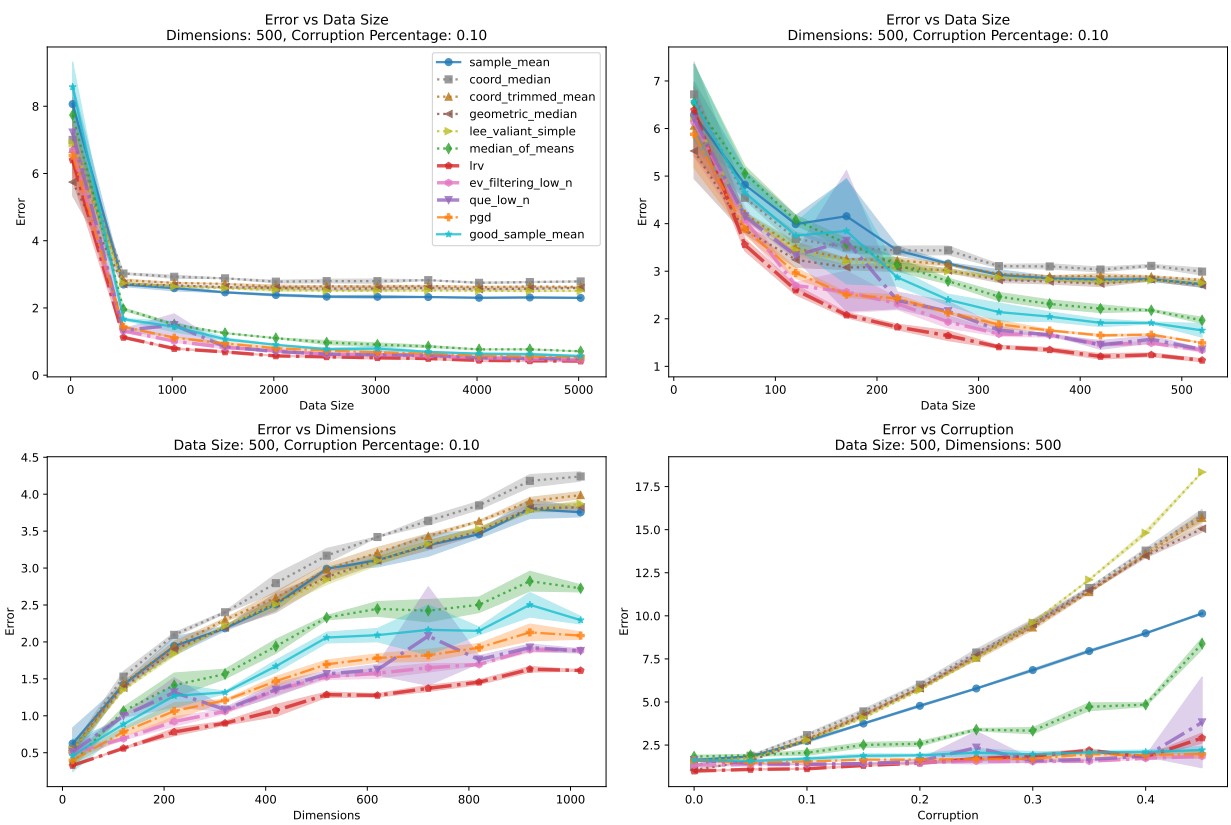

Figure 10: Corrupted Multivariate t-distribution

**Multivariate t-distribution** Define $P$ as the multivariate t-distribution parametrized by $\mu$ as the all-fives vector, $\Sigma$ as the identity matrix, and degrees of freedom $\nu = 3$. Observe that the covariance is $\frac{\nu}{\nu-2}\Sigma$, not $\Sigma$. This is a heavy-tailed distribution with polynomial tail decay. As in other experiments, consider corrupted data distribution $Q = \mathcal{N}_d(\mu', \frac{1}{10}I)$ where $\|\mu - \mu'\| = \sqrt{d}$. Results are shown in Figure 10. As with the other experiments, `QUE_low_n`, `ev_filtering_low_n`, PGD, LRV, and `median_of_means` all notably outperform `sample_mean` here. However here, `QUE_low_n`, `ev_filtering_low_n`, PGD, and LRV all consistently outperform `good_sample_mean`. This trend is particularly notable under lower data size and high dimensions. This is explainable given that the sample mean is known to be a sub-optimal estimator for heavy tailed distributions. These results suggest that robust estimators designed for the Huber contamination model have practical application in the heavy-tailed distribution setting. We leave further investigation into this connection to future work, building on Prasad et al. (2019). We also note that there is a more notable separation between the best performing methods in this settings than in the inlier Gaussian data scenario. In

particular, ev_filtering_low_n and LRV consistently outperform QUE_low_n, and also QUE_low_n exhibits areas of high variance in error.

**Laplace Distribution**    Define $P$ as the product distribution of $d$ independent Laplace distributions, each with mean 5 and scale 1. Then, the true mean $\mu$ is defined as the all-fives vector. Again, consider corrupted data distribution $Q = \mathcal{N}_d(\mu', \frac{1}{10}I)$ where $\|\mu - \mu'\| = \sqrt{d}$. Results are shown in Figure 11. Results among the best estimators are nearly identical to the Gaussian inlier data scenarios, with QUE_low_n, PGD, ev_filtering_low_n all nearly matching good_sample_mean error and with LRV performing marginally worse. We note that although the Laplacian distribution has a heavier tail than the Gaussian, we do not observe the same trends as in the multivariate t-distribution. This can be explained by the fact that Laplacian tails still decay exponentially, whereas the tails in the multivariate t-distribution decay polynomially.

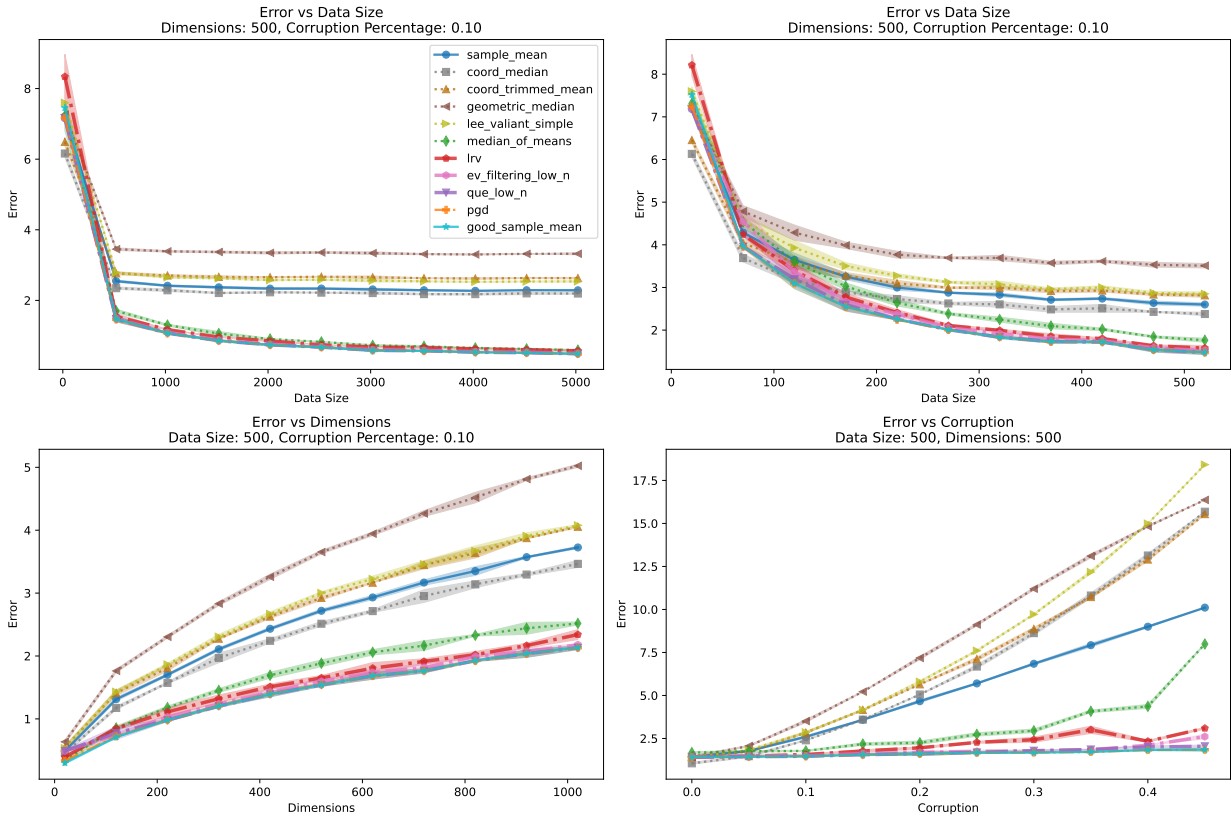

Figure 11: Corrupted Laplace Distribution

**Poisson Distribution**    Define $P$ as the product distribution of $d$ independent Poisson distributions, each with mean 5. Then, the true mean $\mu$ is defined as the all-fives vector. Again consider corrupted data distribution $Q = \mathcal{N}_d(\mu', \frac{1}{10}I)$ where $\|\mu - \mu'\| = \sqrt{d}$. Results are shown in Figure 12. Results among the best estimators are again nearly identical to the Gaussian inlier data scenarios.

**Mixture of Gaussians**    Define $P$ as a mixture of Gaussians with three components, each equally weighted. The components have means $\mu_1 = \vec{1}$, $\mu_2 = \vec{0}$, and $\mu_3 = -\vec{1}$, where $\vec{1}$ and $\vec{0}$ are the $d$-dimensional vectors of all ones and all zeros, respectively. Each component has identity covariance. Define $Q$ as $\mathcal{N}(\vec{2}, \frac{1}{10}I)$ where $\vec{2}$ is the all-twos vector. Results are shown in Figure 13. While robust methods tend to outperform sample_mean, relative performance here differs from the Gaussian inlier data setting. Firstly, note that QUE_low_n does not perform the best and shows irregular areas of high error under lower dimensionality and moderate corruption levels. The observation that QUE_low_n, ev_filtering_low_n, and PGD do not perform as well is likely because they rely significantly on the Gaussianity assumption for the inliers. Methods

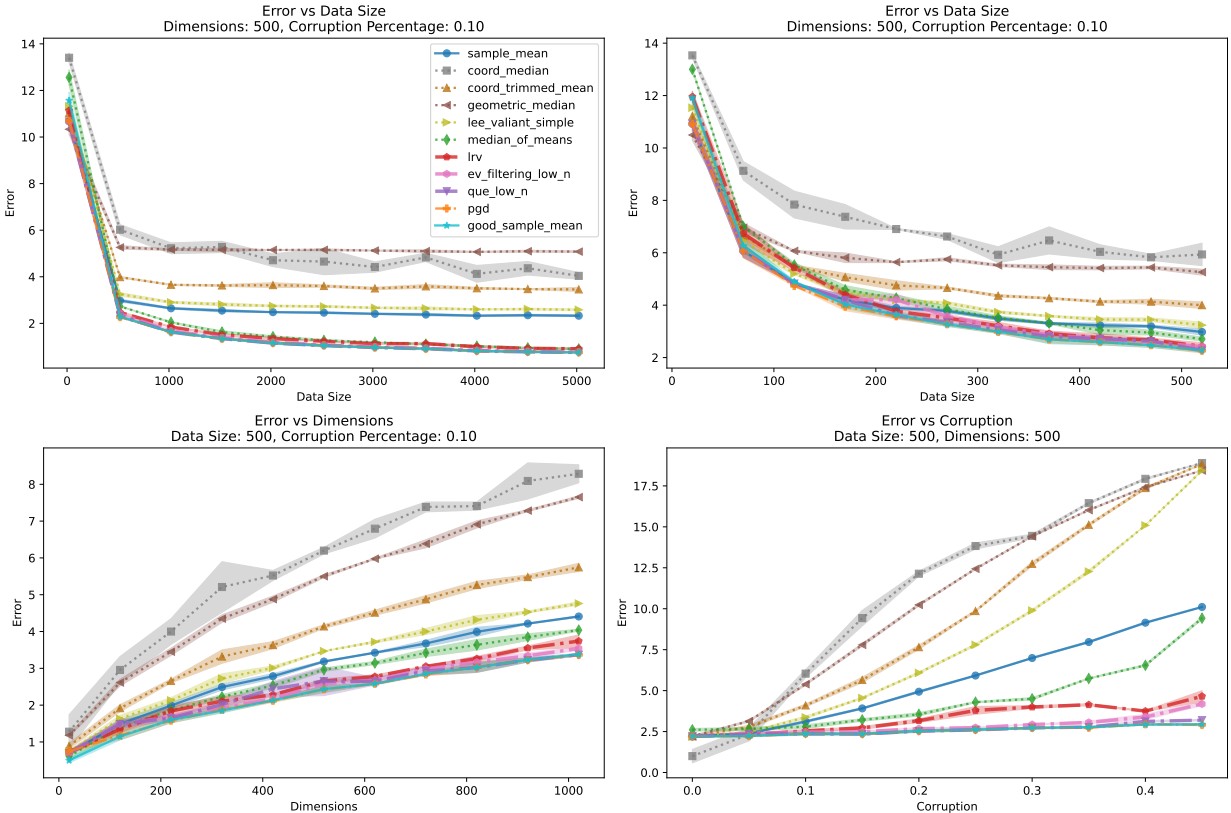

Figure 12: Corrupted Poisson Distribution

such as median_of_means, on the other hand which do not rely as directly on this model are not as effected. Furthermore, ev_filtering_low_n, LRV, and PGD all show significant degradation as corruption levels increase, which is not observed in the Gaussian inlier settings. This may be occurring if they completely filter one of the three "inlier" modes as outliers. Interestingly, LRV notably outperforms all other estimators and even good_sample_mean except with corruption $\tau > 0.2$ and especially for $n < d$. The strong performance of LRV in this setting is interesting, and may be a consequence of the three inlier distributions means lying on a 1-dimensional subspace.

## 7 Comparing Algorithm Variants and Ablation

In this section, we justify and explore our adaptations to ev_filtering, QUE, and lee_valiant. We use these adaptations for the remainder of our experiments.

**Eigenvalue-based Threshold**  Here we compare ev_filtering and ev_filtering_low_n, along with QUE and QUE_low_n. We observe that when we do not have $n$ very large compared to $d$, then ev_filtering and QUE can dramatically shift from low error to abysmal error rates.

We recreate the experiment over corrupted Gaussian data with identity covariance and DKK Noise from Section 4.2. This is shown in Figure 14a. We find that ev_filtering and QUE fail catastrophically with insufficient data, performing far worse than sample_mean. However, ev_filtering_low_n and QUE_low_n never perform worse than sample_mean and achieve near optimal performance regardless of data size. With sufficient data, ev_filtering and QUE abruptly begin to work, and achieve near identical performance to their adjusted threshold counterparts.

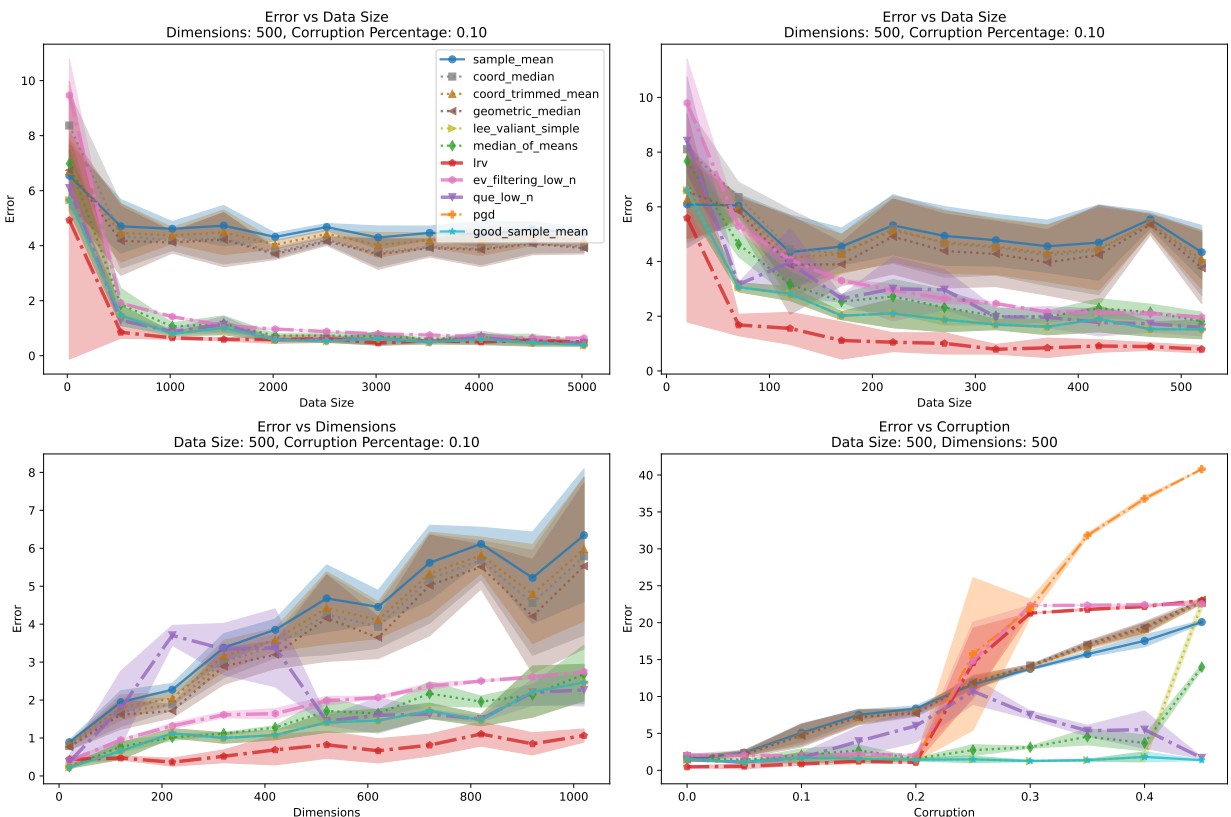

Figure 13: Corrupted Mixture Of Gaussians

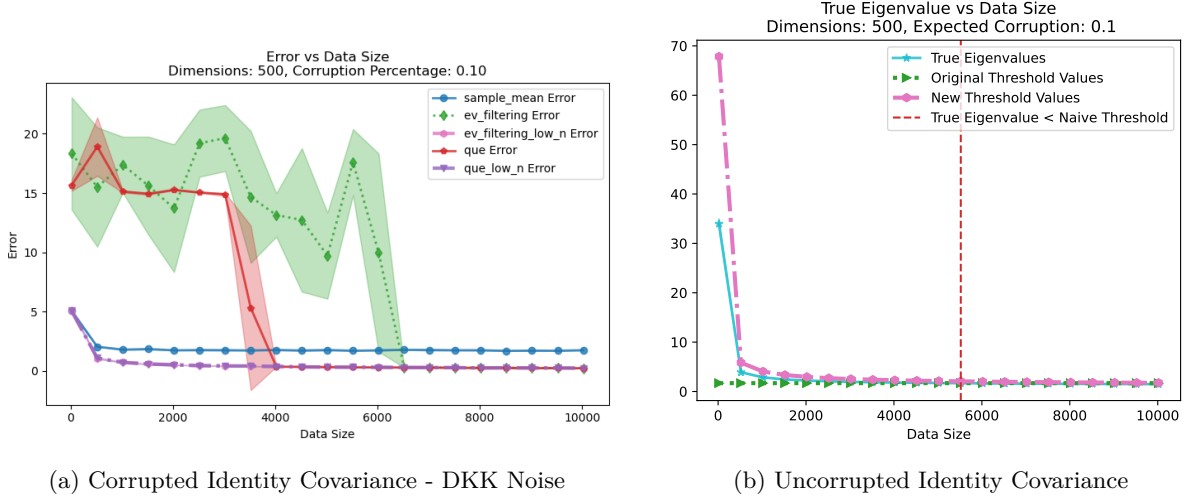

(a) Corrupted Identity Covariance - DKK Noise    (b) Uncorrupted Identity Covariance

Figure 14: Eigenvalue Based Filtering Comparison

The failure of ev_filtering and QUE occurs in the corruption detection step. This corruption detection threshold on the top eigenvalue is initialized as $1 + 3\tau \log(1/\tau)$ in ev_filtering and QUE. This constant threshold uses the fact that with large enough data size, the top eigenvalue of an identity covariance matrix approaches 1 and the $3\tau \log(1/\tau)$ term can account for tolerable noise. However, this does not account for corruption due to low data size, in which the top eigenvalue of the uncorrupted data will necessarily

have a larger expectation as data size decreases. Therefore, ev_filtering and QUE can never be expected to work since even without any corruption, the top eigenvalue will exceed the threshold. As a result, we find that ev_filtering and QUE keep on pruning until there are only very few data points left, resulting in the catastrophic error exhibited. ev_filtering_low_n and QUE_low_n remedy this problem by simply incorporating our new result (Corollary 1.1) as a threshold on the top eigenvalue of the covariance matrix in the threshold. This is empirically shown in Figure 14b. The threshold in ev_filtering and QUE does not become a true upper bound on the top eigenvalue of the uncorrupted data until the vertical red line, which roughly corresponds to the point that ev_filtering_low_n begins to perform better. QUE begins to perform better with much less data size than ev_filtering, but this is unsurprising given the rapid convergence of the inlier top eigenvalue to 1. Meanwhile, our new threshold used in ev_filtering_low_n and QUE_low_n is always an upper bound on the top eigenvalue, and is near-optimal in practice.

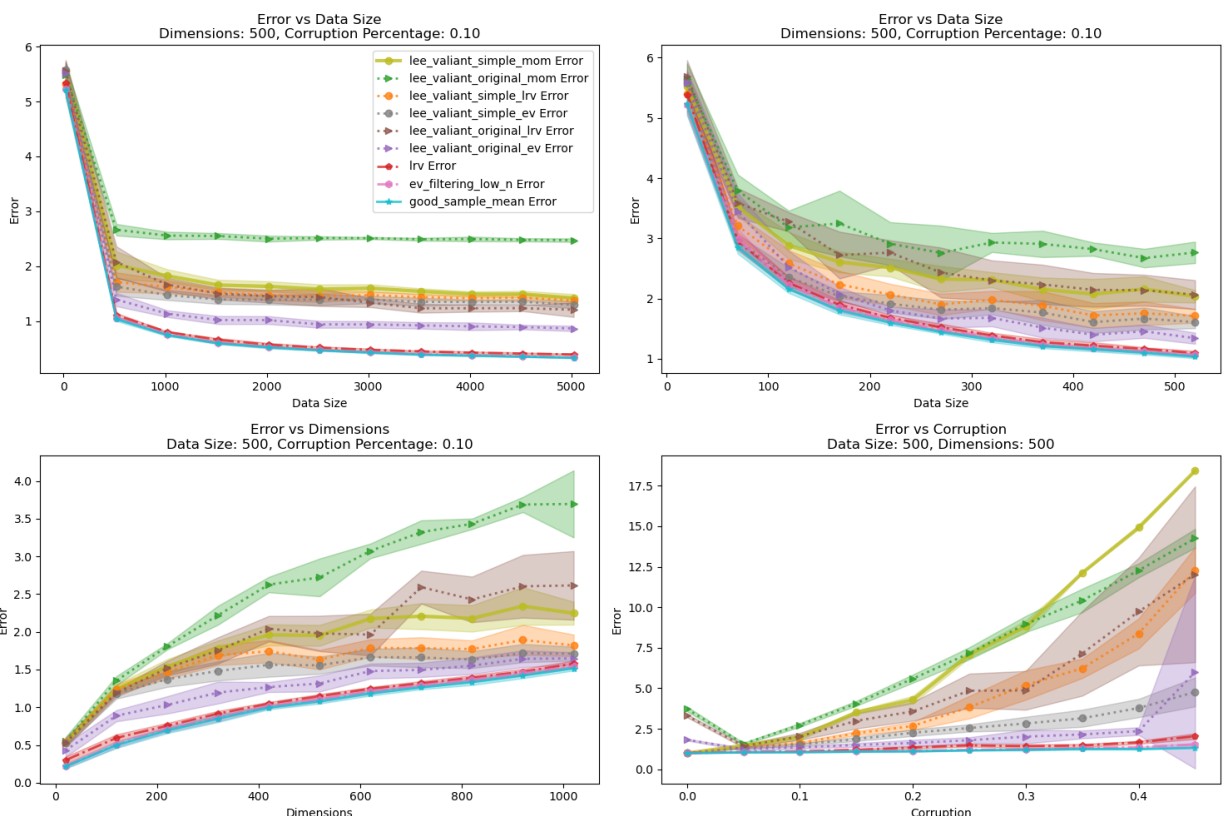

Figure 15: Lee Valiant Variants: Identity Covariance - Additive Variance Shell Noise

**Lee and Valiant variants.** Here we compare different variants of the Lee and Valiant algorithm. We observe that lee_valiant_simple performs a bit better than lee_valiant, and median_of_means is an illustrative choice for initial estimator. We recreate the experiment over Gaussian data with identity covariance and additive variance shell noise from Section 4.2 across different variants of the Lee and Valiant algorithm. We test lee_valiant_simple and lee_valiant using median_of_means, LRV, and ev_filtering_low_n as initial mean estimators. We additionally plot LRV and ev_filtering_low_n as baselines. This is shown in Figure 15.

lee_valiant_simple differs from lee_valiant in two ways: (1) it removes outliers completely instead of down-weighting them and (2) it does not use initial mean estimate in the final result. We see that lee_valiant_simple performs better than lee_valiant in practice, especially with larger $n$ or $d$. Both lee_valiant_simple and lee_valiant see benefit in the use of improved initial estimators, with this improvement being more significant in lee_valiant. The difference in the relative improvement in performance between the algorithms is explained by the fact that lee_valiant additively incorporates the initial estimate directly into its final esti-

mate. However, there is no benefit gained from combining lee_valiant or lee_valiant_simple with an improved initial estimator compared to using the initial estimator alone. As a result of these findings, we only evaluate lee_valiant_simple in our experiments.

## 8 Conclusion

We perform the first wide-scale experimental study of robust mean estimation techniques in high dimensions and relatively-low data size. We showed that under Gaussian data with bounded covariances, robust mean estimation techniques can significantly outperform sample mean, nearly matching the optimal error obtainable, regardless of data size, dimensionality, or corruption level. We provide an updated eigenvalue filtering bound that is useful in this high-dimensional setting, and use it to devise a small but novel and meaningful modification to two existing robust mean estimation algorithms; eigenvalue pruning from Diakonikolas et al. (2019a) and quantum entropy scoring from Dong et al. (2019). This enables these methods to almost exactly match optimal error regardless of data size – that is almost matching the error of the so-called *good sample mean*, which is the mean of the inliers. It seems QUE_low_n works so well because it can identify all ways input distributions deviate from Gaussianity, whereas other methods may require more iterations, which may ultimately prune too many points in the $n \leq d$ setting before it is able to filter outliers in each direction that has them.

However, all methods perform significantly worse than the mean of all inliers under *subtractive corruption* where an $\eta$-fraction of data points can be removed adversarially. This suggests that in this Gaussian modeled data regime, practical improvements may be possible in considering the effect of subtractive corruption.

We also provide a novel evaluation on realistic settings based on the embeddings generated from large language models, deep pretrained image models, and word embedding models. These are representative of real world settings where the data size $n$ may be smaller or not much larger than the dimension $d$. In these settings, quantum entropy scoring with early halting tends to perform near optimally, suggesting its potential application to real world data distributions regardless of data size. However, other robust mean estimation algorithms do not work as well as when the inlier data is not Gaussian, as perhaps foreshadowed by theoretical results leveraging this assumption. This suggests that further valuable results may be derived by moving away from the assumption that inliers are precisely Gaussian. Our initial explorations for corrupted data with non-Gaussian inliers shows the same techniques mostly perform well, but which method performs the best can vary based on the inlier distribution. Notably, some methods can even have less error than good_sample_mean when the sample mean is not the MLE.

Overall, our work demonstrates that there is value in applying robust mean estimation techniques to data, even with insufficient data size for the classic theoretical bounds. We hope that our work inspires researchers to further consider, both experimentally and theoretically, the crucial case of high-dimensional robust statistics under low data size.

### Acknowledgments

The authors thank Meysam Alishahi for early discussions, especially towards the analysis in Theorem 2. We also thank the NSF REU project 224492, under which CA was a participant and initiated this work. JP also thanks funding from NSF 2115677 and 2421782, and Simons Foundation MPS-AI-00010515.

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

# A   Appendix

## A.1   Updated Eigenvalue Threshold

Here we leverage a theorem in Vershynin (2011) to bound the complexity of aggregated high-dimensional Gaussian random variables. We will use it in a couple ways.

**Theorem 2** (Vershynin (2011) Thm. 5.35). *Let $A$ be a $n \times d$ matrix whose entries are independent standard normal random variables. Let $\|A\|_2$ denote the spectral norm of $A$. Then for every $t \geq 0$, with probability of at least $1 - 2\exp(-t^2/2)$, one has*

$$\sqrt{n} - \sqrt{d} - t \leq s_{\min}(A) \leq \|A\|_2 \leq \sqrt{n} + \sqrt{d} + t.$$

*Where $s_{\min}(A)$ is the smallest singular value of $A$. The lower bound assumes $n > d$, if not the roles are reversed.*

We prove the following implication.

**Theorem 3** (restatement of Theorem 1). *Let $X$ be a $n \times d$ matrix whose entries are independently drawn from $\mathcal{N}(\mu, I)$. Let $\Sigma = \frac{1}{n}(X - \bar{\mu})^T(X - \bar{\mu})$ be the sample covariance matrix of $X$, where $\bar{\mu} = \frac{1}{n}\sum_i X_i$ and $X_i$ is the ith row of $X$. Then for every $t > 0$, with probability of at least $1 - 3\exp(-t^2/2)$, one has*

$$\|\Sigma\|_2 \leq \left(1 + \sqrt{d/n} + t/\sqrt{n} + \frac{\sqrt{d + \sqrt{2d}t + t^2}}{n}\right)^2.$$

*Proof.* Let $\bar{X} = X - \mu$ be the centered matrix, equivalent to each entry being drawn from $\mathcal{N}(0, I)$. Let $Z = X - \bar{\mu}$ be the matrix centered by the sample mean. Then $\|Z\|_2 = \|\bar{X} + [\mu - \bar{\mu}]\|_2 \leq \|\bar{X}\|_2 + \|\mu - \bar{\mu}\|_2$ by triangle inequality.

First, by Theorem 2, we have that $\|\bar{X}\|_2 \leq \sqrt{n} + \sqrt{d} + t$, with probability at least $1 - 2\exp(-t^2)$.

Second, to bound $\|\mu - \bar{\mu}\|_2$ we first decompose by coordinate $\|\mu - \bar{\mu}\|_2^2 = \sum_{j=1}^d (\mu_j - \bar{\mu}_j)^2$. Now consider $d$ random variables $B_j = \mu_j - \bar{\mu}_j$ for $j = 1\ldots d$, and further write $B_j = \frac{1}{n}\sum_{i=1}^n F_i$ where $F_i \sim \mathcal{N}(0, 1)$. As a result $B_j \sim \mathcal{N}(0, 1/n) = \frac{1}{\sqrt{n}}\mathcal{N}(0, 1)$, since the average of $n$ normals is still normal with variance reduced by factor $n$. As a result $B_j$ is a squared normal distribution, and $B = n\|\mu - \bar{\mu}\|^2 = n\sum_{j=1}^d B_j^2$ is a chi-squared distribution $\chi^2(d)$. Hence we have (Laurent & Massart, 2000)

$$\Pr[B \geq d + 2\sqrt{d}t + 2s^2] \leq \exp(-s^2).$$

Inside the probability expression, using $\sqrt{B/n} = \|\mu - \bar{\mu}\|$, and letting $t = \sqrt{2}s$, we can rewrite this as

$$\Pr\left[\|\mu - \bar{\mu}\| < \sqrt{(d + \sqrt{2}\sqrt{d}t + t^2)/n}\right] \geq 1 - \exp(-t^2/2).$$

So now if both of these events hold, which by union bound occurs with probability at least $1 - 3\exp(-t^2/2)$, we have that

$$\|Z\|_2 \leq \|\bar{X}\|_2 + \|\mu - \bar{\mu}\| \leq (\sqrt{n} + \sqrt{d} + t) + \sqrt{\frac{d + \sqrt{2d}t + t^2}{n}}$$

Notice that $\|\Sigma\|_2 = \|\frac{1}{n}Z^T Z\|_2 = \frac{\|Z\|_2^2}{n}$. Thus we have

$$\|\Sigma\|_2 = \frac{\|Z\|_2^2}{n} \leq \left(1 + \sqrt{d}/\sqrt{n} + t/\sqrt{n} + \frac{\sqrt{d + \sqrt{2d}t + t^2}}{n}\right)^2$$

$\square$

Note that the fourth term in this bound, coming from the error in $\|\mu - \bar{\mu}\|$, is a lower order effect. This is captured in the following corollary.

**Corollary 3.1** (restatement of Corollary 1.1)**.** *Let $X$ be a $n \times d$ matrix whose entries are independently drawn from $\mathcal{N}(\mu, I)$. Let $\Sigma = \frac{1}{n}(X - \bar{\mu})^T(X - \bar{\mu})$ be the sample covariance matrix of $X$, where $\bar{\mu} = \frac{1}{n}\sum_i X_i$ and $X_i$ is the ith row of $X$. If one assumes $d/n \leq 16, n \geq 16, t \geq 5$, then with probability of at least $1 - 3\exp(-t^2/8)$, one has*

$$\|\Sigma\|_2 \leq \left(1 + \sqrt{d/n} + t/\sqrt{n}\right)^2.$$

*Proof.* Starting with the bound in Theorem 3 we have

$$\|\Sigma\|_2 \leq \left(1 + \sqrt{d/n} + t/\sqrt{n} + \frac{\sqrt{d + \sqrt{2d}t + t^2}}{n}\right)^2$$

$$= \left(1 + \sqrt{d/n} + t/\sqrt{n} + \frac{t}{\sqrt{n}} \cdot \frac{\sqrt{d/t^2 + \sqrt{2d}/t + 1}}{\sqrt{n}}\right)^2$$

$$= \left(1 + \sqrt{d/n} + \frac{t}{\sqrt{n}}\left(1 + \sqrt{(d/n)/t^2 + \sqrt{2}\sqrt{d/n}/t/\sqrt{n} + 1/n}\right)\right)^2$$

$$\leq \left(1 + \sqrt{d/n} + \frac{t}{\sqrt{n}}\left(1 + \sqrt{(16)/t^2 + \sqrt{2}\sqrt{16}/t/\sqrt{n} + 1/n}\right)\right)^2$$

$$\leq \left(1 + \sqrt{d/n} + \frac{t}{\sqrt{n}}\left(1 + \sqrt{16/25 + \sqrt{32}/(5\sqrt{n}) + 1/n}\right)\right)^2$$

$$\leq \left(1 + \sqrt{d/n} + \frac{t}{\sqrt{n}}\left(1 + \sqrt{16/25 + \sqrt{32}/(5 \cdot 4) + 1/16}\right)\right)^2$$

$$< \left(1 + \sqrt{d/n} + 2t/\sqrt{n}\right)^2$$

Adjusting $t$ to $2t$ in the probability of failure, so it is $3\exp(-t^2/8)$ instead of $3\exp(-t^2/2)$, completes the proof. $\square$

## A.2 Corrupted Gaussian Data Identity Covariance: Additional Noise Schemes

We examine the performance of robust mean estimators across additional corruption schemes. We still draw $X \sim (1-\eta)P + \eta Q$ where $P = \mathcal{N}_d(\mu, I)$ and $Q$ describes the corrupted data distribution, where $\mu$ is the all-fives vector. We utilize the following additional corruption schemes:

**Two Gaussian clusters shifted to variance shell.** Consider corrupted data distribution $Q = 0.7\mathcal{N}_d(\mu^0, \frac{1}{10}I) \cup 0.3\mathcal{N}_d(\mu^1, \frac{1}{10}I)$ where $\|\mu - \mu^0\| = \sqrt{d}$, $\|\mu - \mu^1\| = \sqrt{d}$, and $\theta = 75°$ where $\theta$ is the angle between $\mu^0$ and $\mu^1$. The location of $\mu^0$ is determined by a random rotation matrix to prevent any coordinate-axis specific biases. Results over this noise distribution are shown in Figure 16.

**In Distribution Noise.** Consider corrupted data distribution, $Q$, where for each corrupted data point $q_i \in Q$, each coordinate $j$ of $q_i$ is drawn from $\mathsf{Uniform}(\mu_j, \mu_j + 2)$. Here $\mu_j$ represents the $j$th coordinate of the true mean $\mu$. Results over this noise distribution are shown in Figure 17.

**Large Outlier Noise.** Consider corrupted data distribution $Q = 0.7\mathcal{N}_d(\mu^0, \frac{1}{10}I) \cup 0.3\mathcal{N}_d(\mu^1, \frac{1}{10}I)$ where $\|\mu - \mu^0\| = 10\sqrt{d}$, $\|\mu - \mu^1\| = 20\sqrt{d}$, and $\theta = 75°$ where $\theta$ is the angle between $\mu^0$ and $\mu^1$. The location of $\mu^0$ is determined by a random rotation matrix to prevent any coordinate-axis specific biases. Results over this noise distribution are shown in Figure 18.

**Large Outlier Noise Mixes.** Consider corrupted data distribution $Q = 0.5L \cup 0.5Q'$ where $L$ is the large outlier corruption scheme previously described and $Q'$ is a subtle corruption scheme. We examine two settings for $Q'$: additive variance shell corruption with one cluster, shown in Figure 19, and DKK corruption, shown in Figure 20.

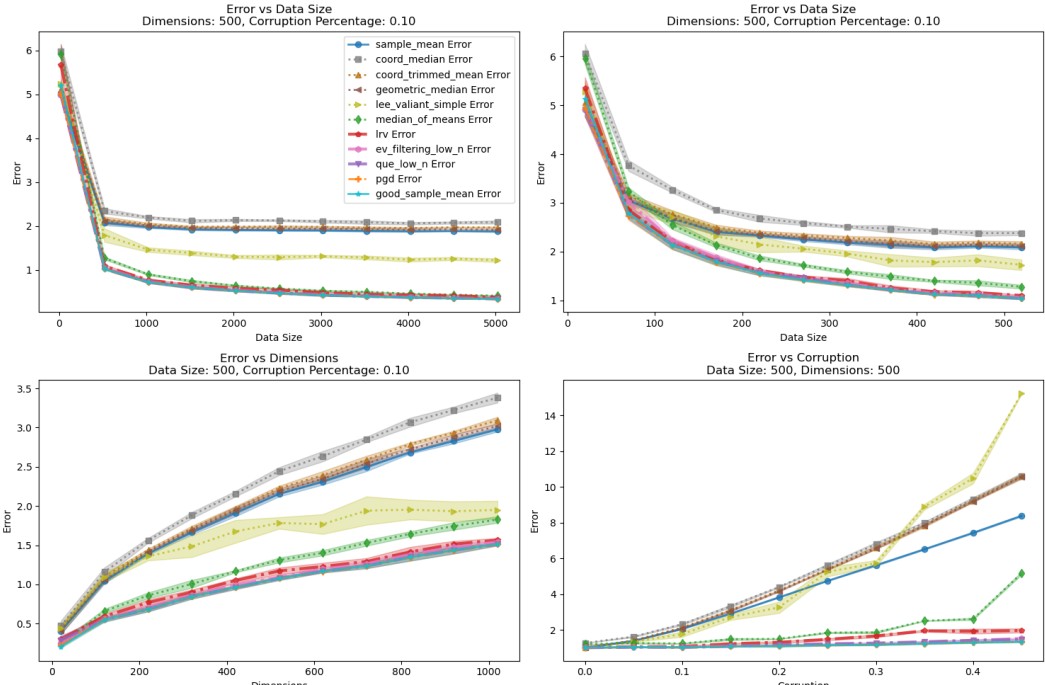

Figure 16: Corrupted Gaussian Identity Covariance: Two Variance Shell Clusters

Across all of these distributions, including those with large outliers, ev_filtering_low_n, QUE_low_n, PGD, and LRV perform the best, suggesting that their performance is not overly sensitive to the noise distribution. However, we note that across schemes with large outliers, PGD sees areas of higher variance and slightly worse performance, and LRV degrades worse as $\eta$ increases with large outliers. This downgrade in performance can be remedied by first preprocessing data by removing large outliers through a naive pruning method, but this step doesn't appear necessary for other methods. We remark that LRV requires outlier weights to be clipped to avoid numerical instability issues under large outlier schemes. Otherwise, it will degrade poorly over large outliers and large $\eta$ as predicted outliers will be assigned near-zero weights. We also see again that median_of_means significantly outperforms other simple estimators, especially under large data size, although its performance degrades poorly under certain conditions, such as with larger $\eta$. With large outliers, lee_valiant_simple nearly matches good_sample_mean error across conditions, achieving much better performance than it does across subtle noise distributions. As it additionally outperforms coord_trimmed_mean, this suggests that lee_valiant_simple may operate as a more effective naive pruning method, as seen in the LLM experiments.

**Dependence on true mean** We additionally verify that performance does not depend on the choice of true mean, $\mu$. We recreate experiments over Additive Variance Shell Noise and DKK Noise over different choices of $\mu$. We replicate the same experimental setup as before, but draw every coordinate of $\mu$ from $\mathcal{N}(0, 50)$ at every iteration in the experiment rather than fixing $\mu$ as the all-fives vector. As a reminder, this occurs for every choice of the independent variable over every run. If performance depended on $\mu$, we would expect to achieve high variance results. Instead, we find nearly identical results to the original experiments across both distributions. These results are shown in Figure 21 and Figure 22.

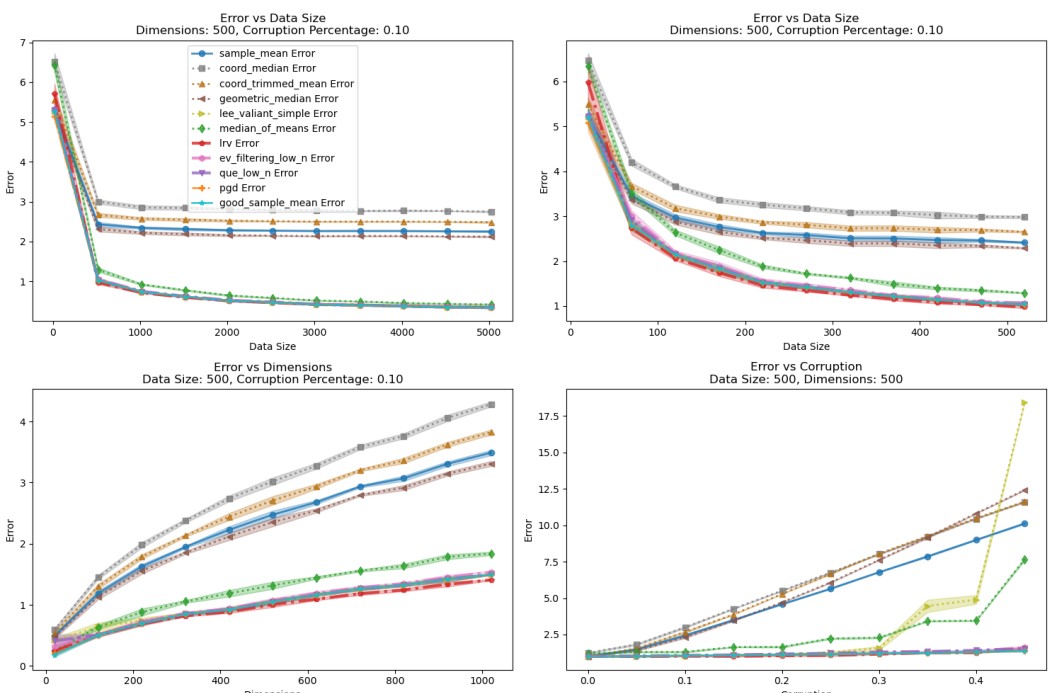

Figure 17: Corrupted Gaussian Identity Covariance: In Distribution Noise

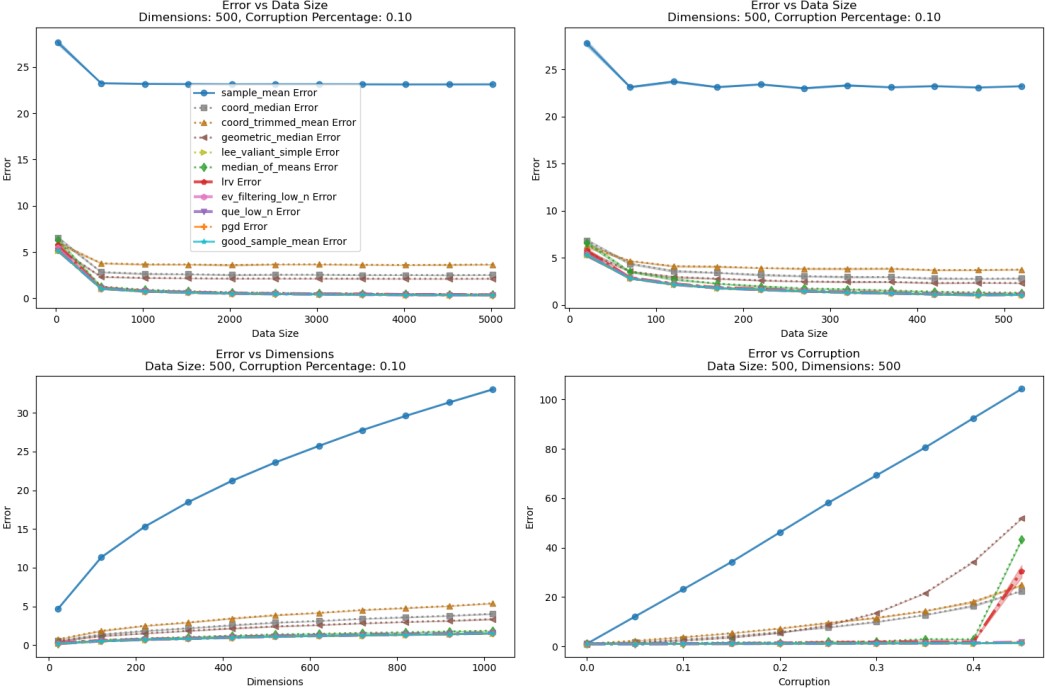

Figure 18: Corrupted Gaussian Identity Covariance: Large Outliers

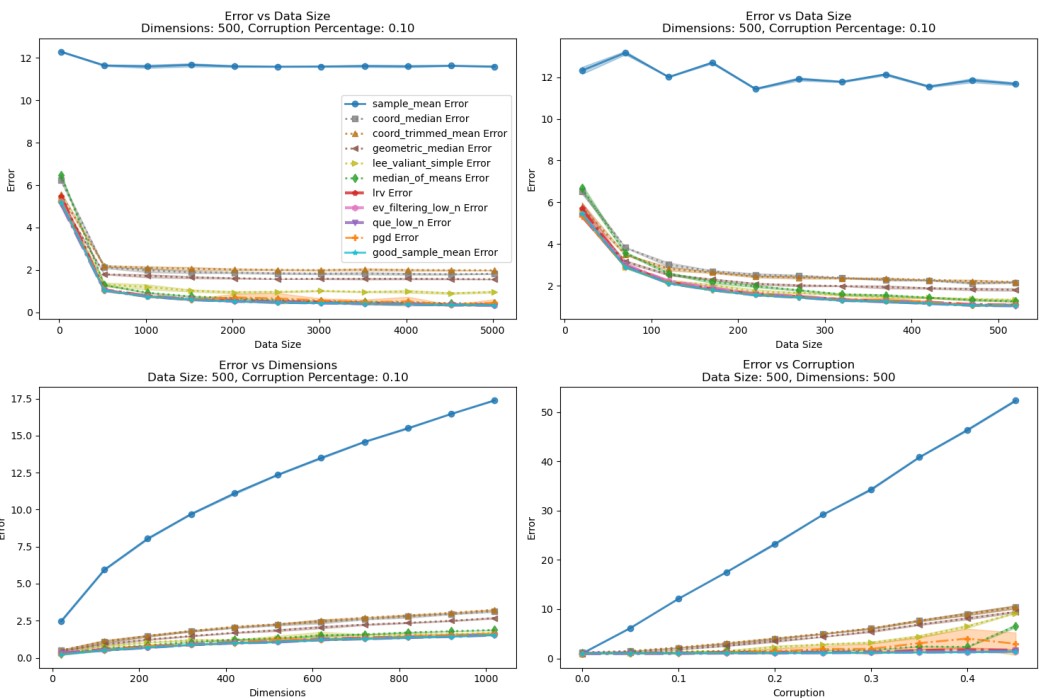

Figure 19: Corrupted Gaussian Identity Covariance: Large Outliers w/ Additive Variance Shell Noise

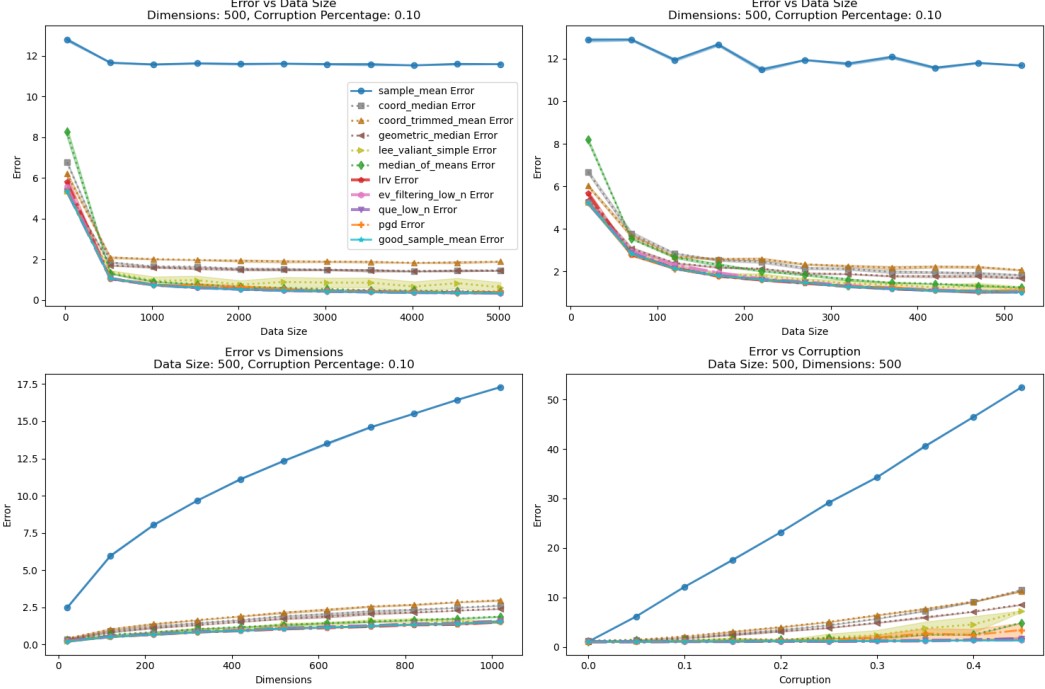

Figure 20: Corrupted Gaussian Identity Covariance: Large Outliers w/ DKK Noise

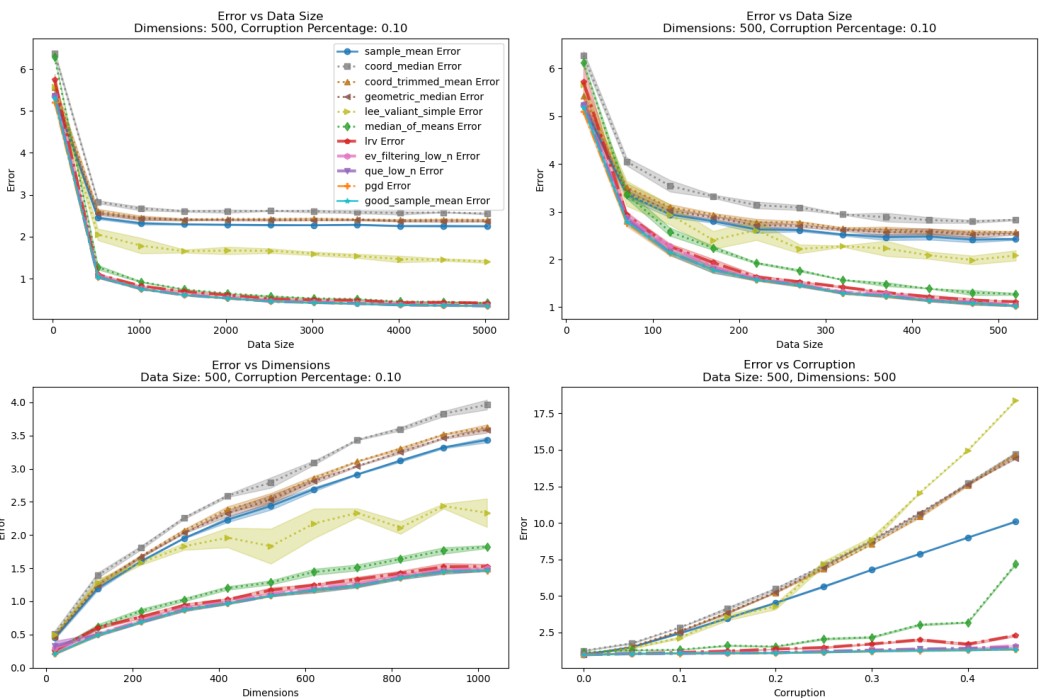

Figure 21: Dependence On True Mean: Identity Covariance, Additive Variance Shell Noise

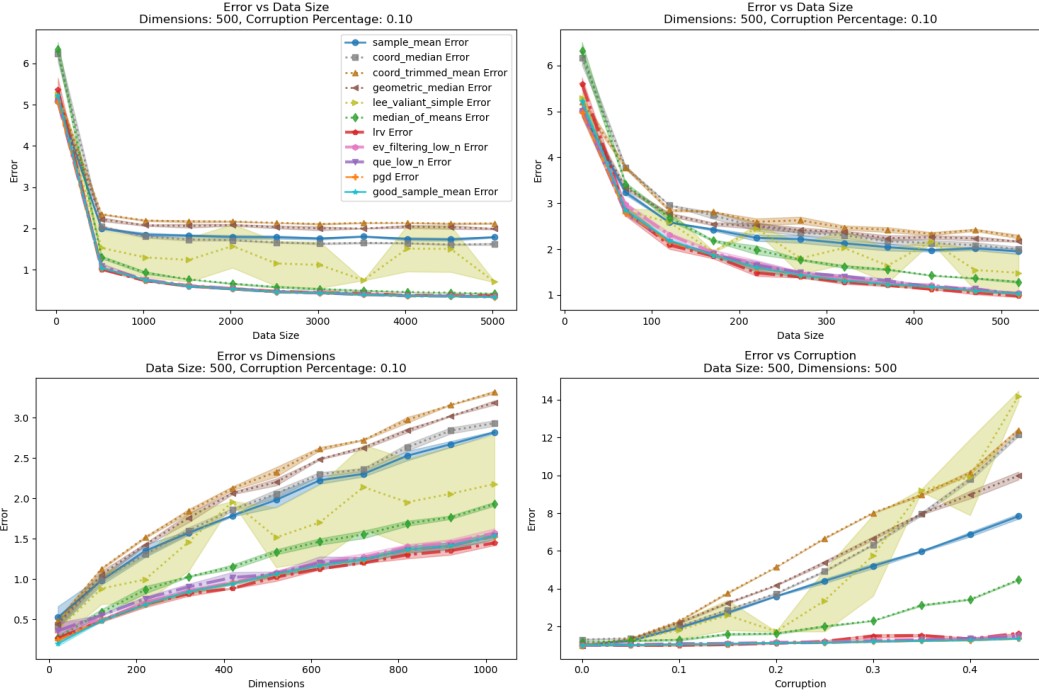

Figure 22: Dependence On True Mean: Identity Covariance, DKK Noise

### A.3 Corrupted Gaussian Data Unknown Spherical Covariance: Additional Corruption Schemes

We examine the unknown spherical covariance case across additional corruption schemes. We utilize the same uncorrupted distribution as before, $P = \mathcal{N}_d(\mu, \sigma^2 I)$ where $\mu$ is the all-fives vector and $\sigma = 5$. We find similar performance across the distributions we test.

**Adapting noise distributions to spherical covariance**  As in the Gaussian noise shifted to variance shell case, we utilize the well know Gaussian concentration inequality that for data $X \sim \mathcal{N}_d(\mu, \sigma I)$, $\mathbb{E}_{x \sim X}[\|x - \mu\|^2] = \sigma^2 d$. This observation is used to adapt noise distributions in the identity covariance case to this case. For two additive variance shell clusters, each cluster has mean $\mu^i$ for $i \in [0, 1]$ where $\|\mu - \mu^i\| = \sigma\sqrt{d}$, with other conditions remaining the same; results are shown in Figure 23. For DKK noise, half the noise is drawn over the hypercube where every coordinate is $-\sigma$ or $0$ away from the true mean at that coordinate with equal probability. The other half is drawn from the product distribution where the first coordinate is either $11\sigma$ or $-\sigma$ away from the true mean at that coordinate with equal probability, the second coordinate is $-3\sigma$ or $-\sigma$ away from the corresponding true mean coordinate with equal probability, and all remaining coordinates are $-\sigma$ away from the true mean. Results are shown in Figure 24. For in distribution corruption, we draw each coordinate $j$ of a corrupted data point from $\mathsf{Uniform}(\mu_j, \mu_j + 2\sigma)$; results are shown in Figure 25. We also perform subtractive corruption, using the same scheme as in the identity covariance case; results are shown in Figure 26.

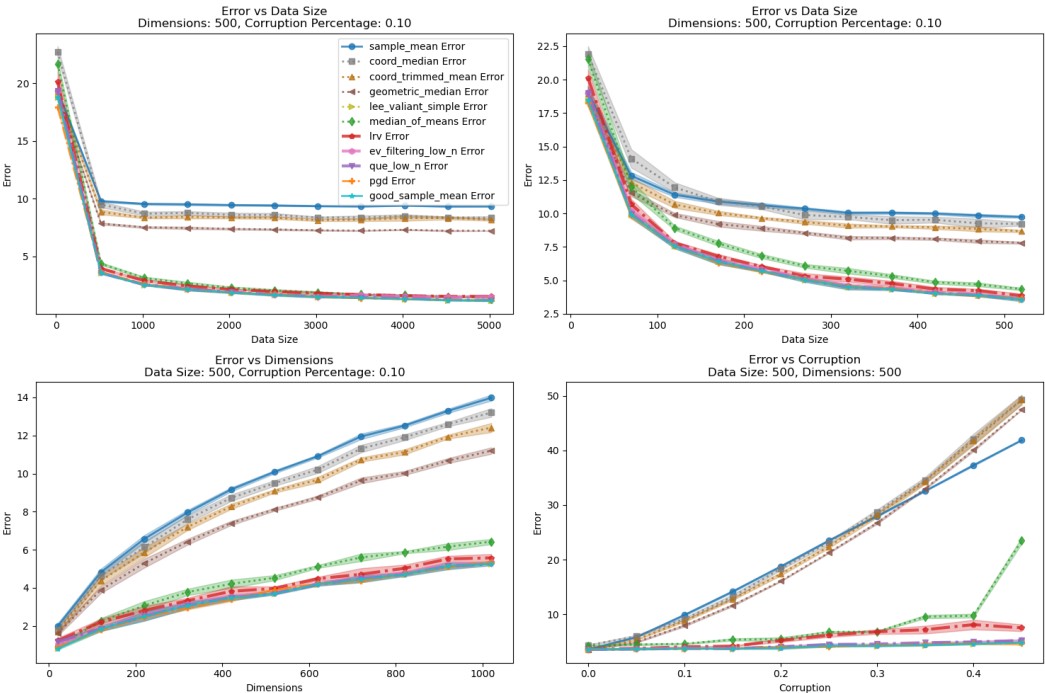

Figure 23: Corrupted Gaussian Large Spherical Covariance: Two Variance Shell Clusters

We find similar results to the identity covariance case across the best estimators, again observing the near optimal performance of QUE_low_n and PGD, along with the slightly worse but still near optimal performance of LRV. As a result of using the scaling data heuristic, ev_filtering_low_n degrades slightly. This is especially noticeable with DKK noise, where it does not converge to good_sample_mean error as data size increases. QUE_low_n appears to be less sensitive to the trace scaling heuristic, retaining its performance in these experiments. There is some variance among other estimators. Notably, lee_valiant_simple performs near optimally across two variance shell corruption and in distribution noise with spherical covariance (its performance in these plots is hidden amongst the best estimators which approximately match good_sample_mean error), whereas it performs comparably worse across analogous noise distributions for identity covariance

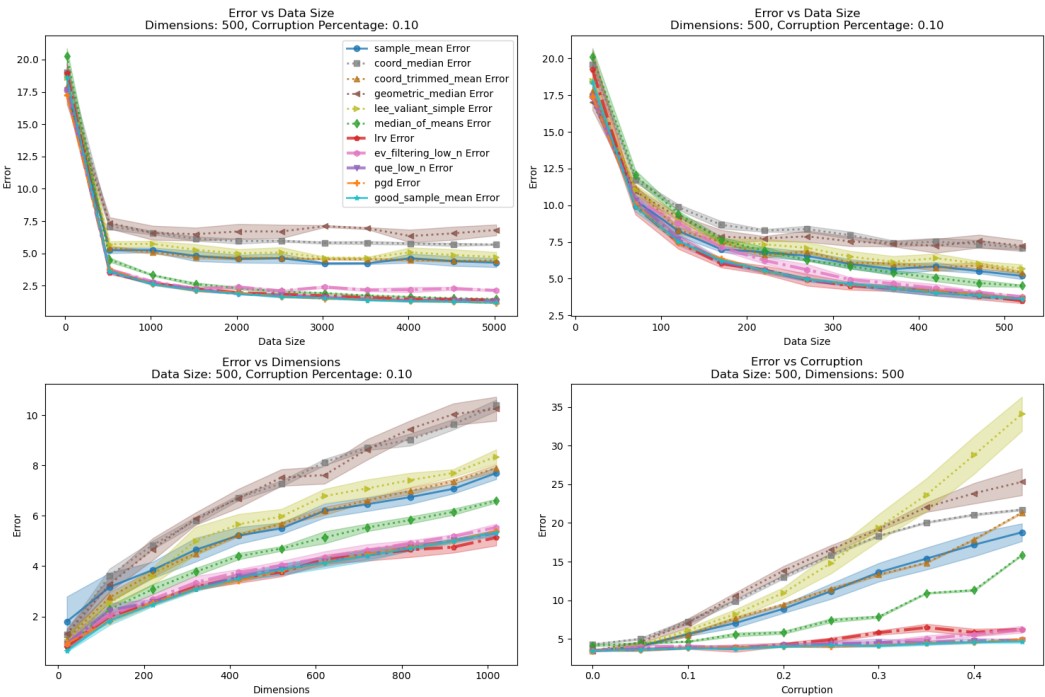

Figure 24: Corrupted Gaussian Large Spherical Covariance: DKK Noise

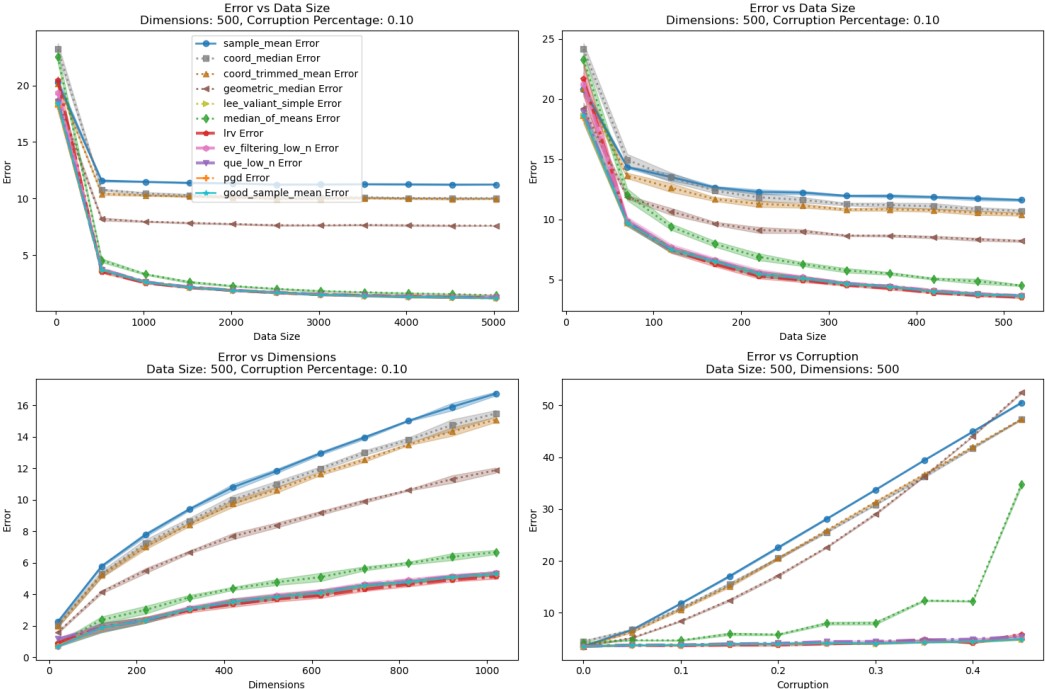

Figure 25: Corrupted Gaussian Large Spherical Covariance: In Distribution Noise

data. Still, lee_valiant_simple does not generally perform better in the spherical covariance case compared to the (known) identity covariance case; performing consistently worse than sample_mean compared to out-performing sample_mean except with large $\eta$ under identity covariance.

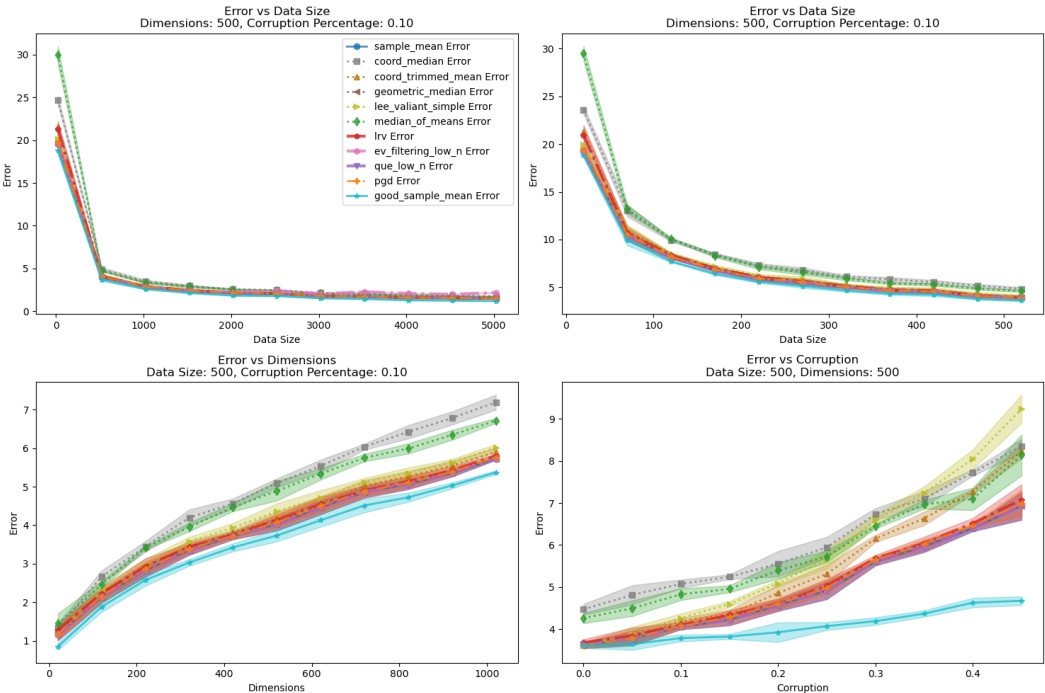

Figure 26: Corrupted Gaussian Large Spherical Covariance: Subtractive Noise

**Varying** $\sigma$ We rerun several experiments as we vary $\sigma$ from $\sigma = 0.1$ to $\sigma = 200$ – the coordinate wise standard deviation of the true covariance matrix – and fix other variables as their default values. In particular, we examine Additive Variance Shell Noise, DKK Noise, In Distribution Uniform Noise, and Two Variance Shell Clusters Noise. Results are shown in Figure 27. As expected, error tends to increase linearly with $\sigma$. Generally, relative performance of the algorithms remains the same, with ev_filtering_low_n, QUE_low_n, and PGD nearly identically matching good_sample_mean error throughout. Surprisingly, LRV error does not grow linearly with $\sigma$, consistently outperforming even good_sample_mean with large enough choices of $\sigma$. A similar trend is also seen for coord_median, but only across DKK Noise, in which it is noticeably the best estimator with larger values of $\sigma$.

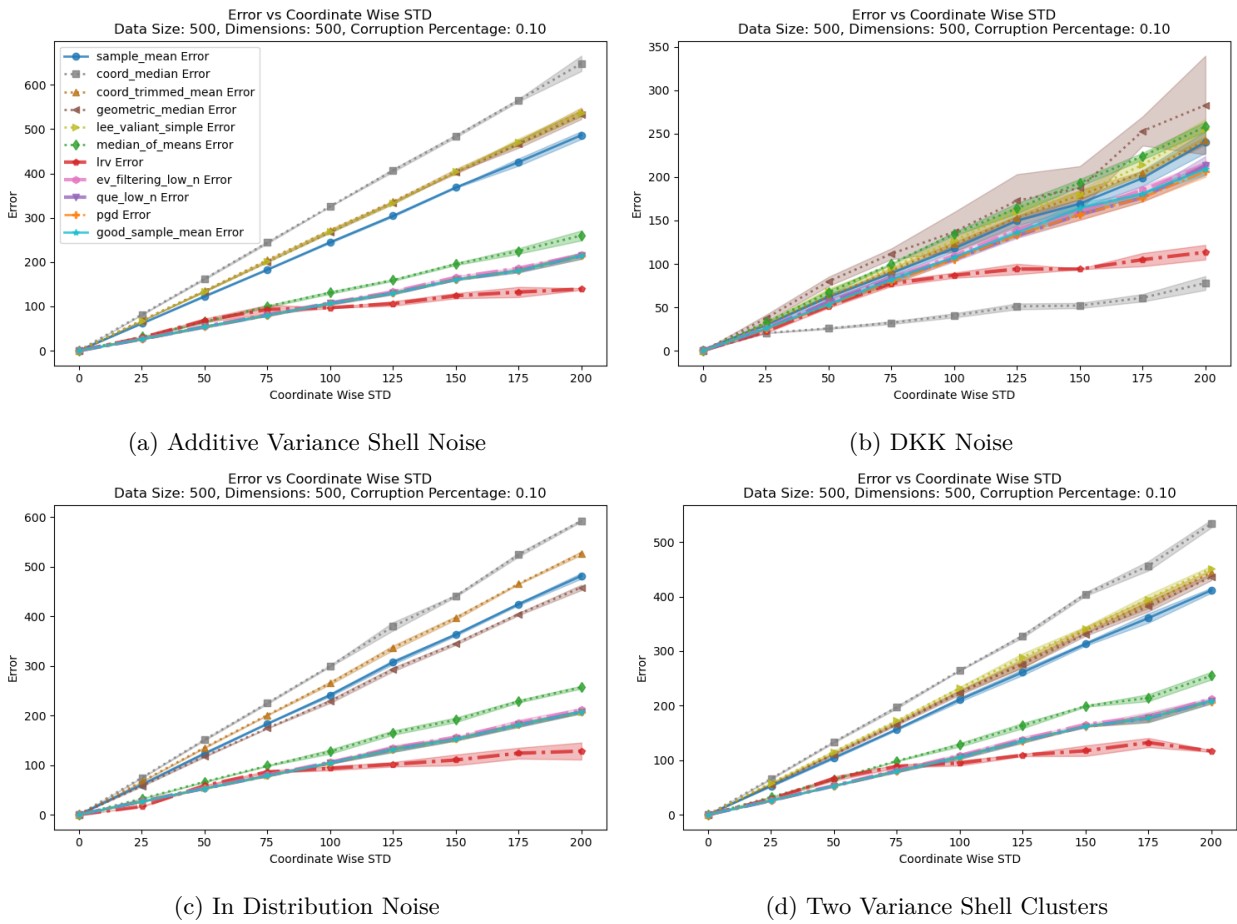

(a) Additive Variance Shell Noise

(b) DKK Noise

(c) In Distribution Noise

(d) Two Variance Shell Clusters

Figure 27: Corrupted Gaussian Large Spherical Covariance: Varying Coordinate-Wise Standard Deviation $\sigma$

### A.4 Corrupted Gaussian Data Unknown Non Spherical Covariance

#### A.4.1 Unknown Diagonal Covariance

Here we consider the performance of mean estimators on corrupted Gaussian data with unknown diagonal non-spherical covariance. We draw uncorrupted data from $\mathcal{N}_d(\mu, \Sigma)$ where $\mu$ is the all fives-vector and $\Sigma$ has large diminishing covariance. In particular, the diagonal elements uniformly decrease from 25 to 0.1.

**Noise Distributions** We adapt the variance shell additive noise distribution to cluster outliers to be a standard deviation away from the true mean along every coordinate axis. That is, consider corrupted data distribution $Q = \mathcal{N}_d(\mu', \frac{1}{10}I)$ with $|\mu'_j - \mu_j| = \Sigma_j$, where $\Sigma_j$ is the $j$th diagonal element in $\Sigma$, $\mu'_j$ is the $j$th coordinate of $\mu'$, and $\mu_j$ is the $j$th coordinate of the true mean $\mu$; results are shown in Figure 28. We adapt in distribution uniform noise to draw each coordinate $j$ of a corrupted data point from $\mathsf{Uniform}(\mu_j, \mu_j + \Sigma_j)$; results are shown in Figure 29. For large outlier noise we weight the distance of clusters from $\mu$ by $\sqrt{\frac{\mathrm{Tr}(\Sigma)}{d}}$. That is, consider corrupted data distribution $Q = 0.7\mathcal{N}_d(\mu^0, \frac{1}{10}I) \cup 0.3\mathcal{N}_d(\mu^1, \frac{1}{10}I)$ where $\|\mu - \mu^0\| = 10\sqrt{\frac{\mathrm{Tr}(\Sigma)}{d}}\sqrt{d}$, $\|\mu - \mu^1\| = 20\sqrt{\frac{\mathrm{Tr}(\Sigma)}{d}}\sqrt{d}$, and $\theta = 75°$ where $\theta$ is the angle between $\mu^0$ and $\mu^1$.; results are shown in Figure 30. We also utilize subtractive noise which already works in this case; results are shown in Figure 31.

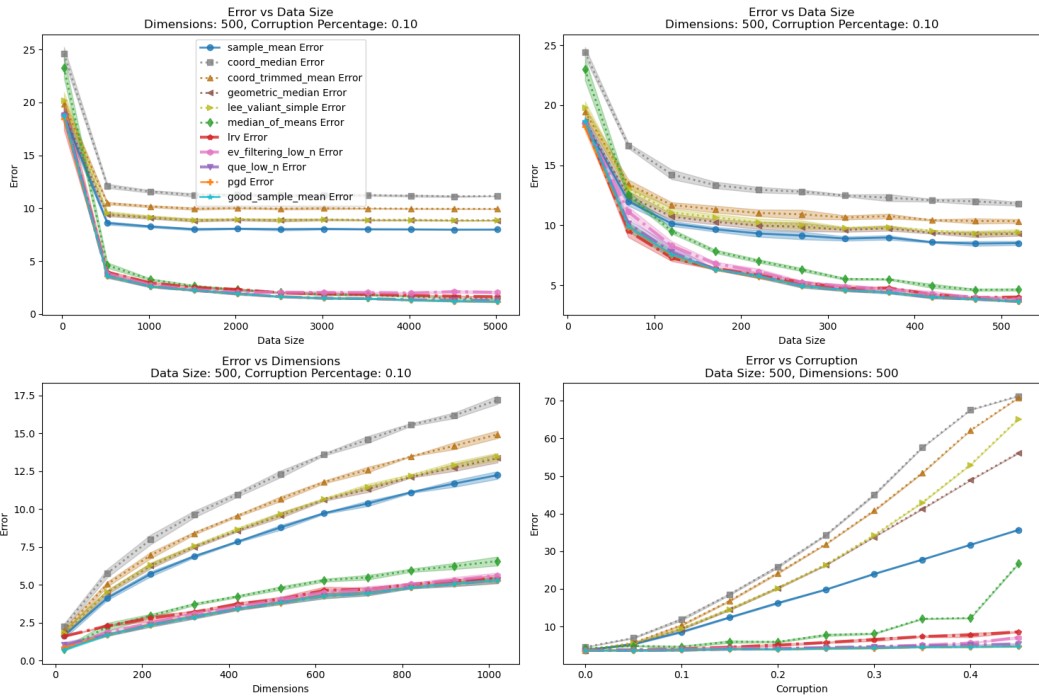

Figure 28: Corrupted Gaussian Large Diminishing Diagonal Covariance: Additive Variance Shell Noise

Again, we find that QUE_low_n and PGD nearly match good_sample_mean error across distributions, with LRV doing slightly worse but still significantly outperforming sample_mean. ev_filtering_low_n still does among the best here, but, as in the large spherical covariance case, sees slight degradation as a result of the scaling data heuristic. In particular, error does not clearly converge to good_sample_mean error as $n$ increases over Additive Variance Shell Noise and In Distribution Noise. QUE_low_n does not encounter this issue despite employing the same heuristic to generalize to non-identity covariance data, suggesting that it is more robust to distributional assumptions. Additionally, median_of_means performs best among simpler estimators, outperforming the sample_mean with $n \approx d$ and performing similarly to good_sample_mean with sufficiently large $n$.

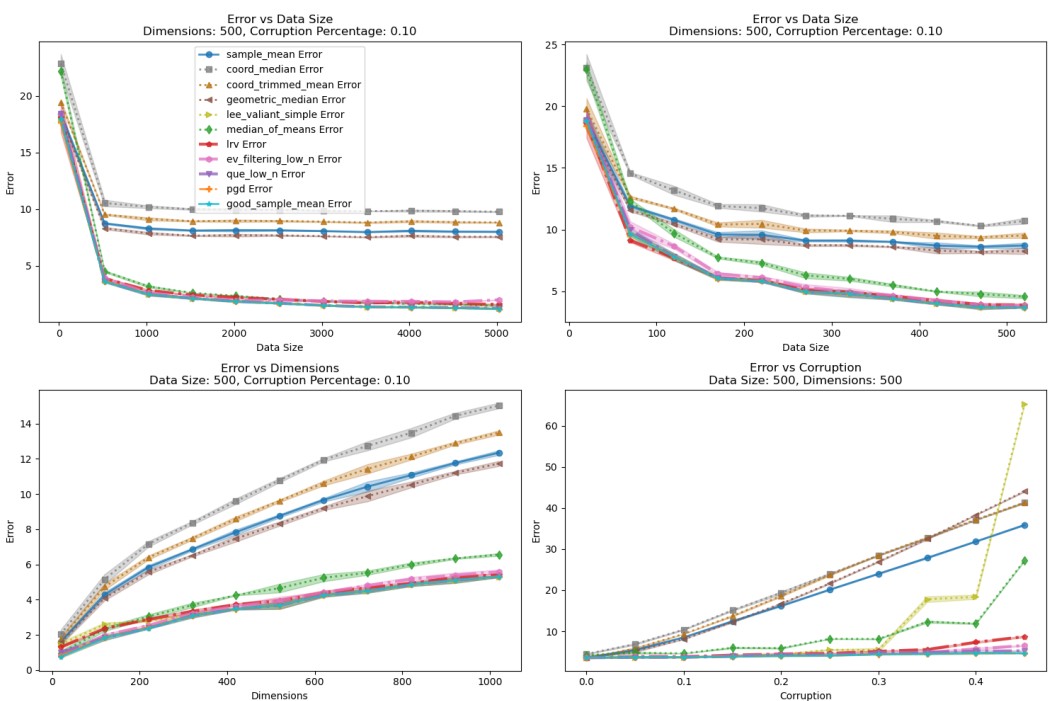

Figure 29: Corrupted Gaussian Large Diminishing Diagonal Covariance: In Distribution Noise

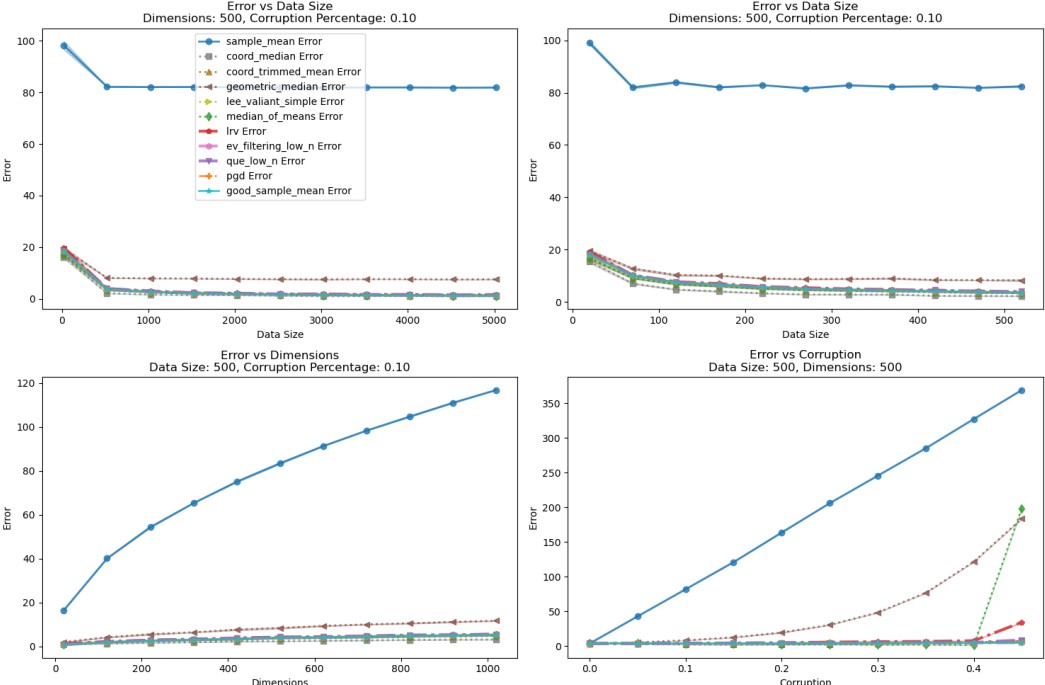

Figure 30: Corrupted Gaussian Large Diminishing Diagonal Covariance: Large Outliers

**Varying top Eigenvalue** We rerun Additive Variance Shell Noise and In Distribution Noise as we vary the squareroot of the top eigenvalue of the true covariance matrix, labeled as $\sigma$, from $\sigma = 0.1$ to $\sigma = 200$. In particular, the diagonal of the covariance will uniformly decrease from $\sigma^2$ to 0.1. For every choice of $\sigma$,

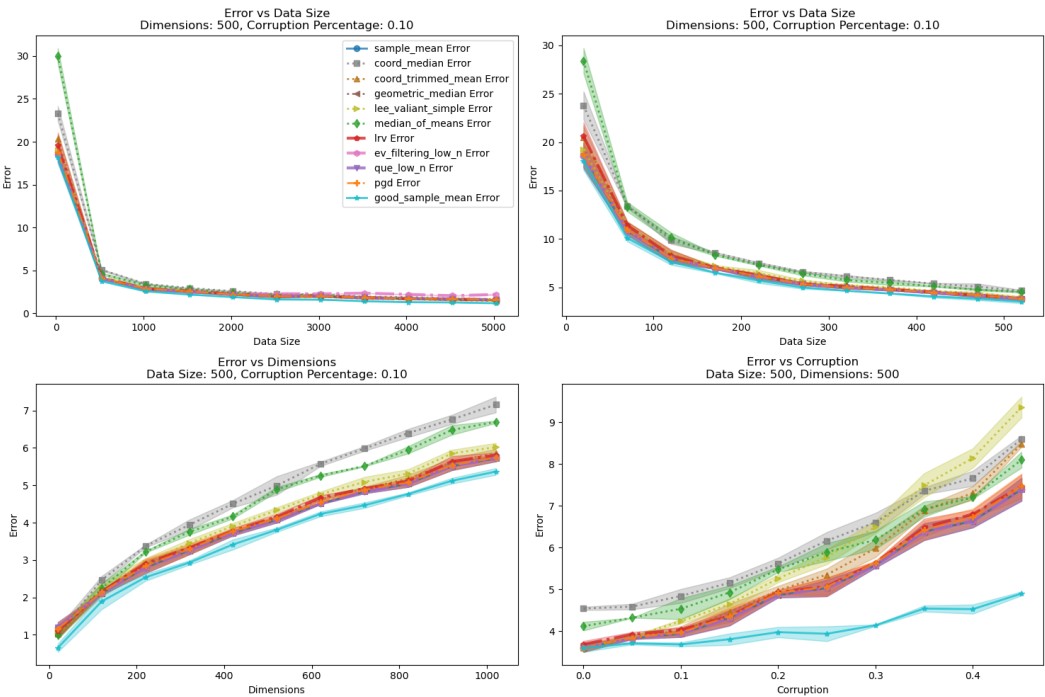

Figure 31: Corrupted Gaussian Large Diminishing Diagonal Covariance: Subtractive Noise

the noise is scaled as described previously. These results are shown in Figure 32. Like in the spherical case, we find that the relative performance of algorithms remains nearly identical throughout choices of $\sigma$.

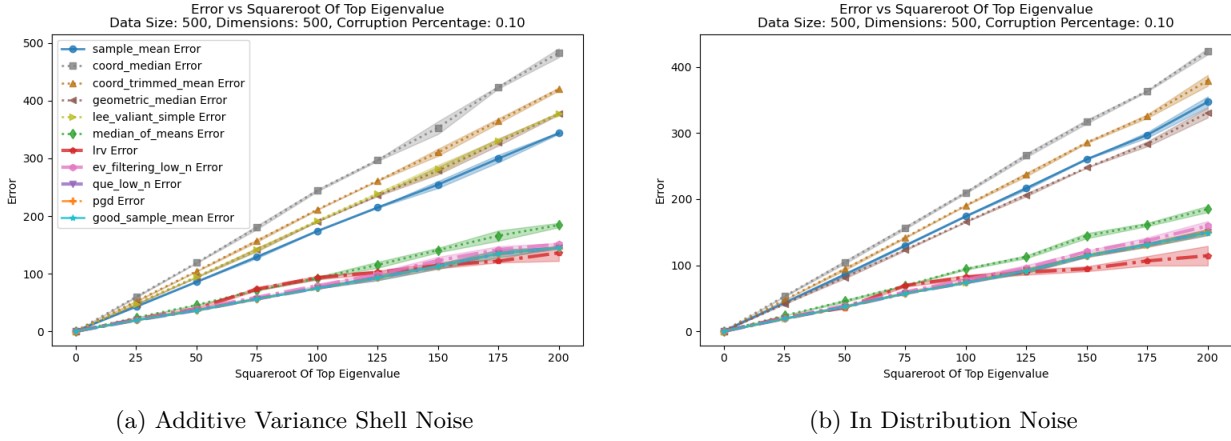

(a) Additive Variance Shell Noise

(b) In Distribution Noise

Figure 32: Corrupted Gaussian large diminishing diagonal covariance: Varying the square root of the top eigenvalue: $\sigma$

### A.4.2 Unconstrained Covariance

So far, we have only examined inlier data with diagonal covariance matrices. However, in line with the intuition that there is nothing inherently special about the standard orthonormal basis, we hope for a robust estimator to work well regardless of the choice of coordinate axis. Since the covariance matrix is always symmetric, it is also diagonalizable by taking the eigenvectors as the orthonormal basis. Then, any possible data distribution over unconstrained covariance can be framed as a data distribution over a diagonal matrix by using these eigenvectors as the orthonormal basis. As a result, any robust estimator that does not leverage the standard orthonormal basis should perform equally well on unconstrained covariance. However, this does not necessarily hold for the estimators that we examine. We employ a trace estimate to adapt ev_filtering_low_n and QUE_low_n to the unknown covariance case. coord_median, coord_trimmed_mean, and median_of_means all directly utilize coordinate wise calculations. LRV utilizes a trace estimate when downweighting points. In this section, we evaluate the performance of robust mean estimators over data with non-diagonal covariance matrices.

**Rotated Data Noise**   Because the covariance matrix is always symmetric, it is diagonalizable, and experiments over unconstrained covariance can be framed as an ablation on noise distributions over inliers with diagonal covariances. We reuse data and noise distributions, but randomly rotate everything before estimation, resulting in unconstrained true covariances and appropriately difficult noise distributions. Random rotation is implemented by generating a standard normal matrix and utilizing its QR decomposition. We examine the performance on Rotated Identity Covariance with DKK Noise in Figure 33; Rotated Identity Covariance with Subtractive Noise in Figure 34; Rotated Large Spherical Covariance with Additive Variance Shell Noise (with coordinate-wise standard deviation $\sigma = 5$) in Figure 35; and Rotated Large Diminishing Covariance with Additive Variance Shell Noise (with squareroot of the top eigenvalue $\sigma = 5$) in Figure 36. As in the original experiments, we set the true mean, $\mu$, to be the all-fives vector.

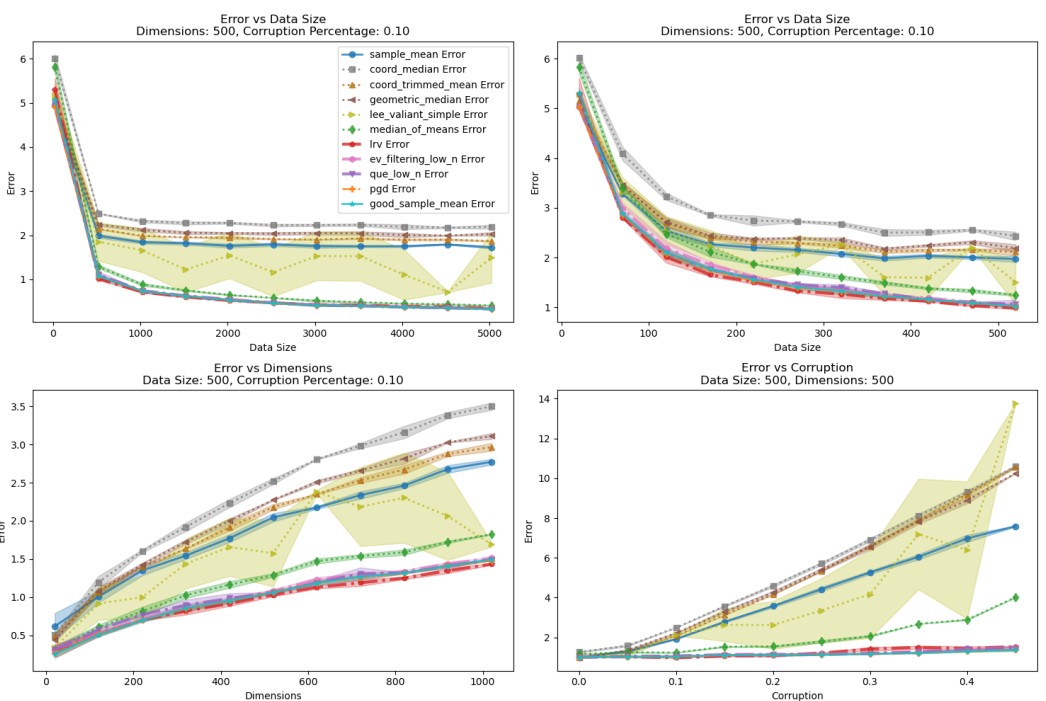

Figure 33: Corrupted Rotated Identity Covariance - DKK Noise

We find nearly identical results among the best estimators to the corresponding non-rotated data experiment. While many of these algorithms induce a bias to the coordinate axis, they are not enough to significantly skew results in the distributions that we examine. There is some variation between coord_median and coord_trimmed_mean with the corresponding non-rotated data experiments, but no major changes in their

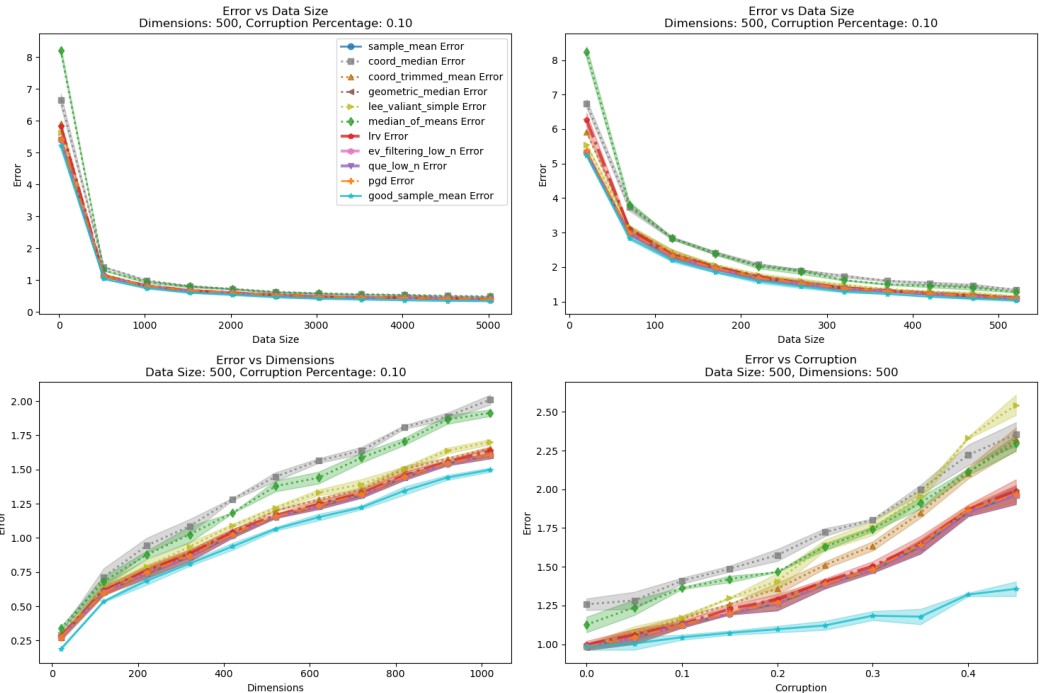

Figure 34: Corrupted Rotated Identity Covariance - Subtractive Noise

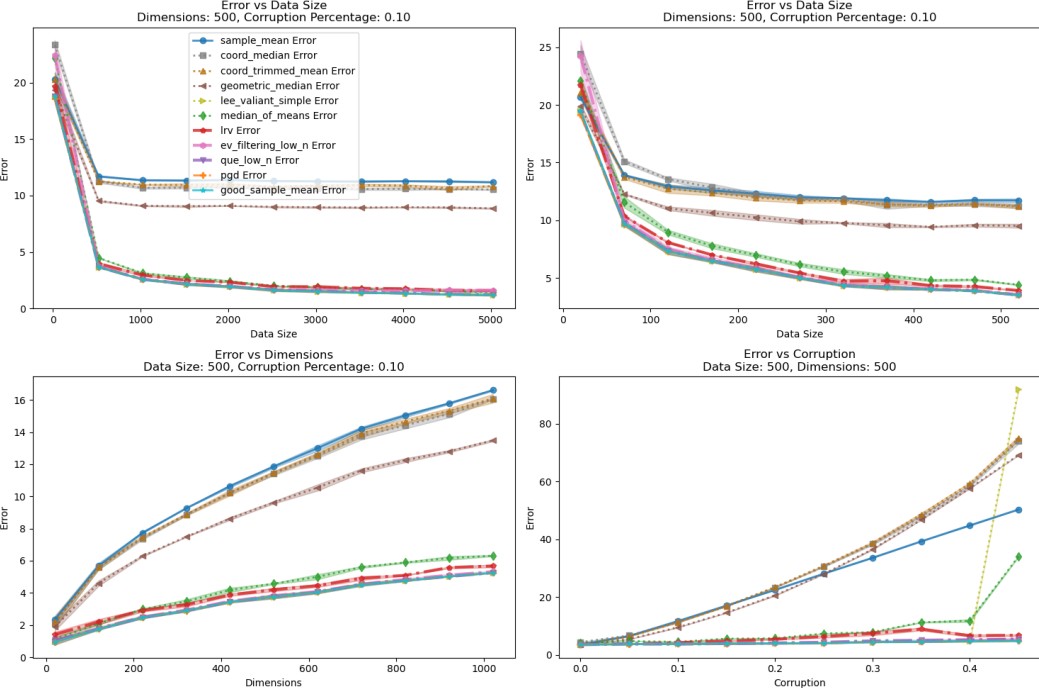

Figure 35: Corrupted Rotated Large Spherical Covariance - Additive Variance Shell Noise

trends. There is no such variation for QUE_low_n, ev_filtering_low_n, or median_of_means among the settings that we test.

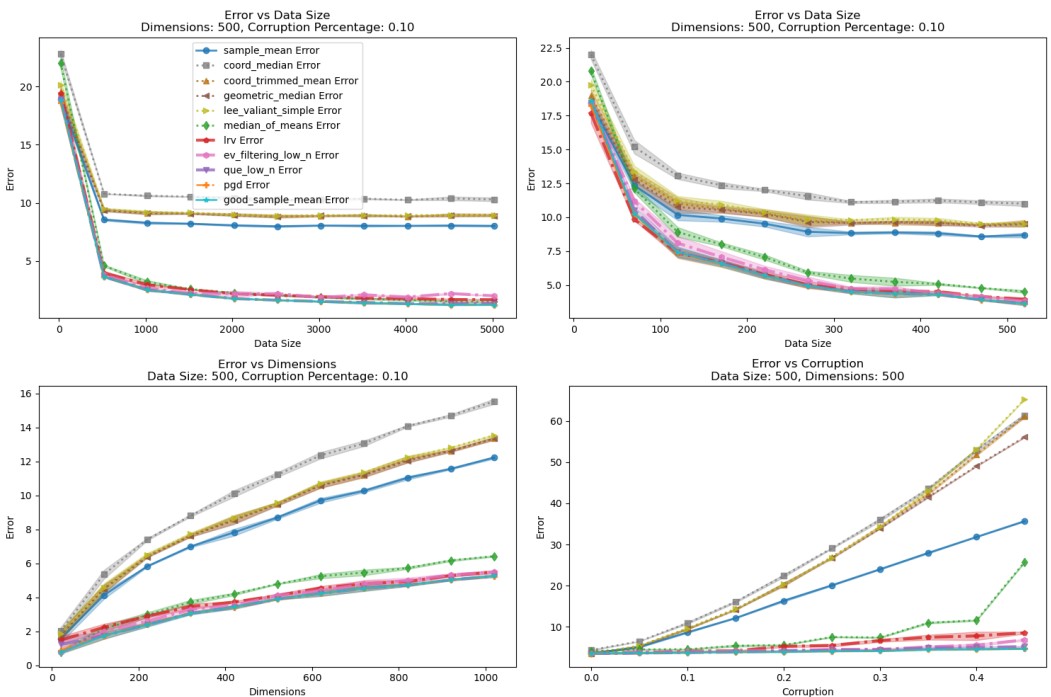

Figure 36: Corrupted Rotated Large Diminishing Covariance - Additive Variance Shell Noise

## A.5 Hyperparameter Tuning

In this section we tune the hyperparameters of some of the most interesting algorithms, median_of_means, LRV, ev_filtering_low_n, and PGD. We utilize the best hyperparameters in this section across all other experiments. In general, we find that none of these algorithms are overly sensitive to choices of hyperparamaters, as long as they lay within a reasonable range.

We evaluate performance over a subset of 4 corrupted data distributions previously discussed: Identity Covariance with DKK Noise; Identity Covariance with In Distribution Noise; Large Spherical Covariance with Variance Shell Additive Noise; Large Diminishing Covariance with Variance Shell Additive Noise. We manually pick the hyperparameters that achieve the best performance across these distributions, or when similar use default ones from the corresponding paper.

**Median of how many means?** Here we explore the parameter $k$ in median_of_means algorithm. This parameter controls the number of chunks that we split the data into; then we take the median of $k$ means determined by these chunks. We vary $k$ in the set $[3, 5, 10, 15, 20, 30]$. For the case where $n < k$, we simply set $k = n$. These results are shown in Figures 37, 38, 39, 40. We find that although there is not always an obvious choice for $k$, that $k = 10$ tends to perform well throughout most settings. However, we find that this and larger choices of $k$ are more prone to error as $\eta$ increases than smaller choices of $k$. Approximately when $\eta > 0.15$, $k = 3$ becomes the best choice of $k$. However, with smaller corruption, such as $\eta = 0.10$ which we generally test, $k = 3$ performs notably worse, making $k = 10$ a better choice. Since we utilize $\eta = 0.1$ as a default value, we set $k = 10$ throughout our experiments.

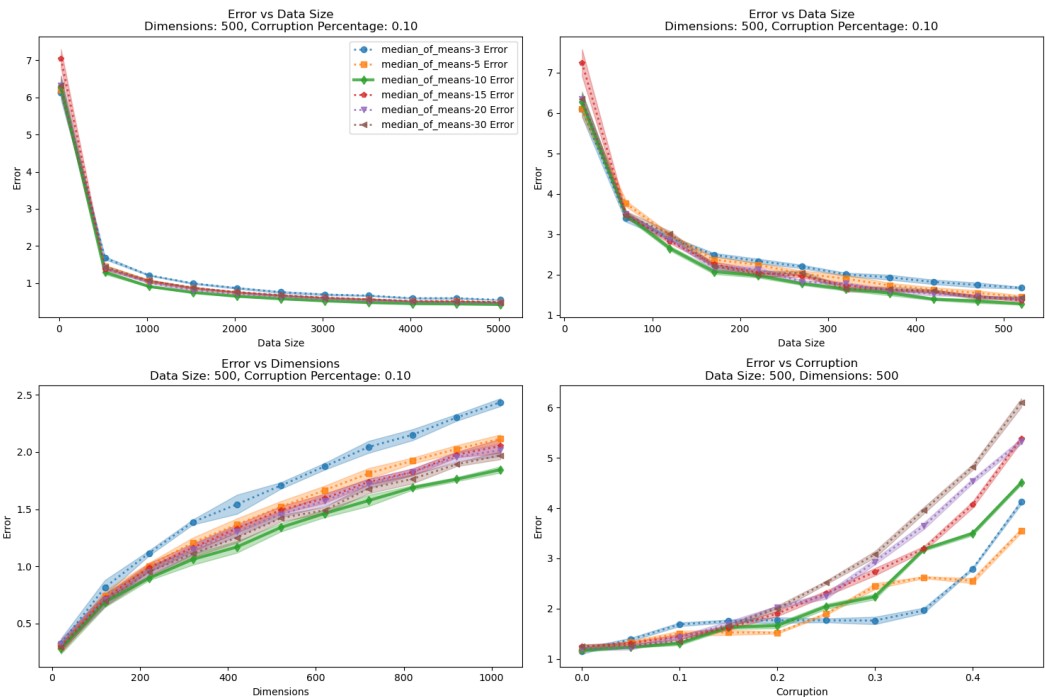

Figure 37: Median Of Means - Number of Chunks $k$: Identity Covariance, DKK Noise

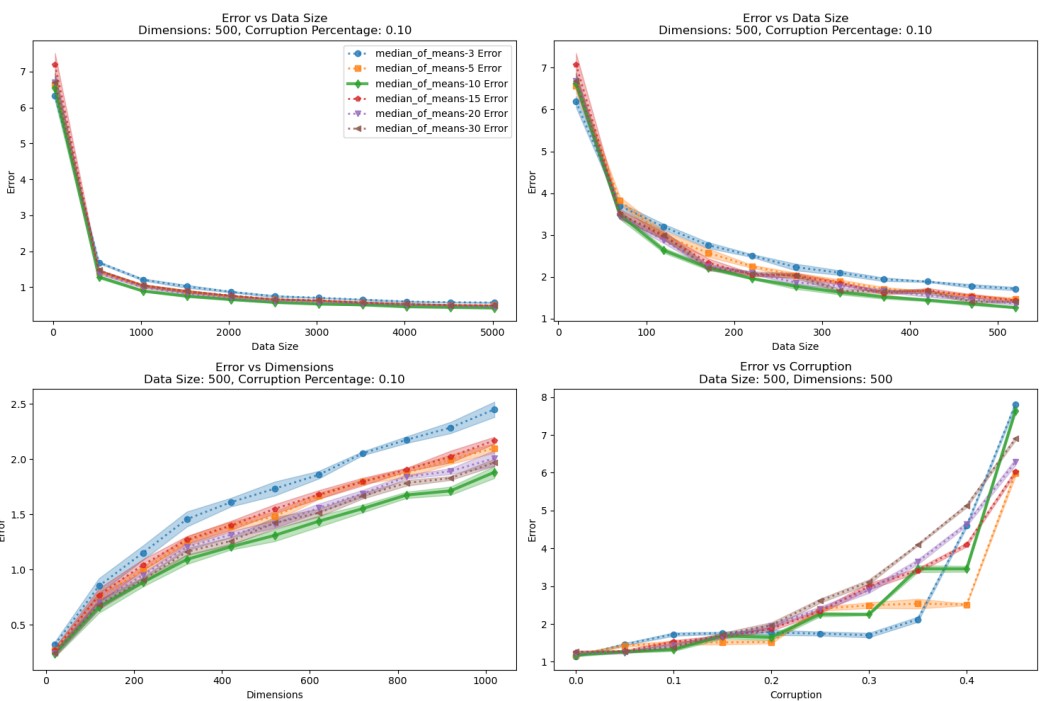

Figure 38: Median Of Means - Number of Chunks $k$: Identity Covariance, In Distribution Noise

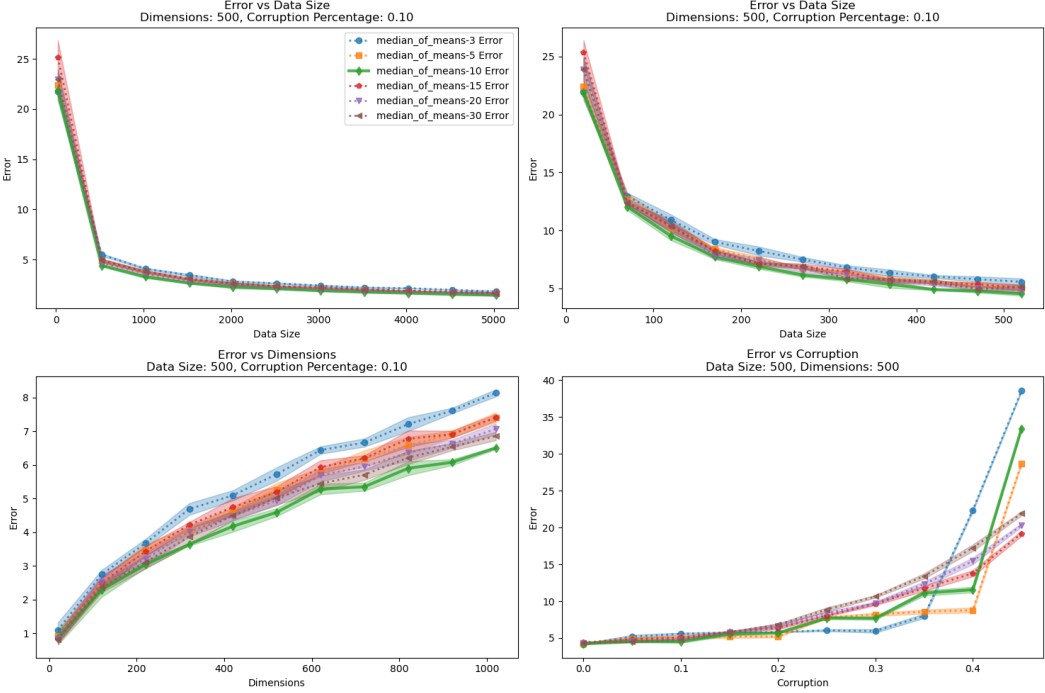

Figure 39: Median Of Means - Number of Chunks $k$: Large Spherical Covariance, Additive Variance Shell Noise

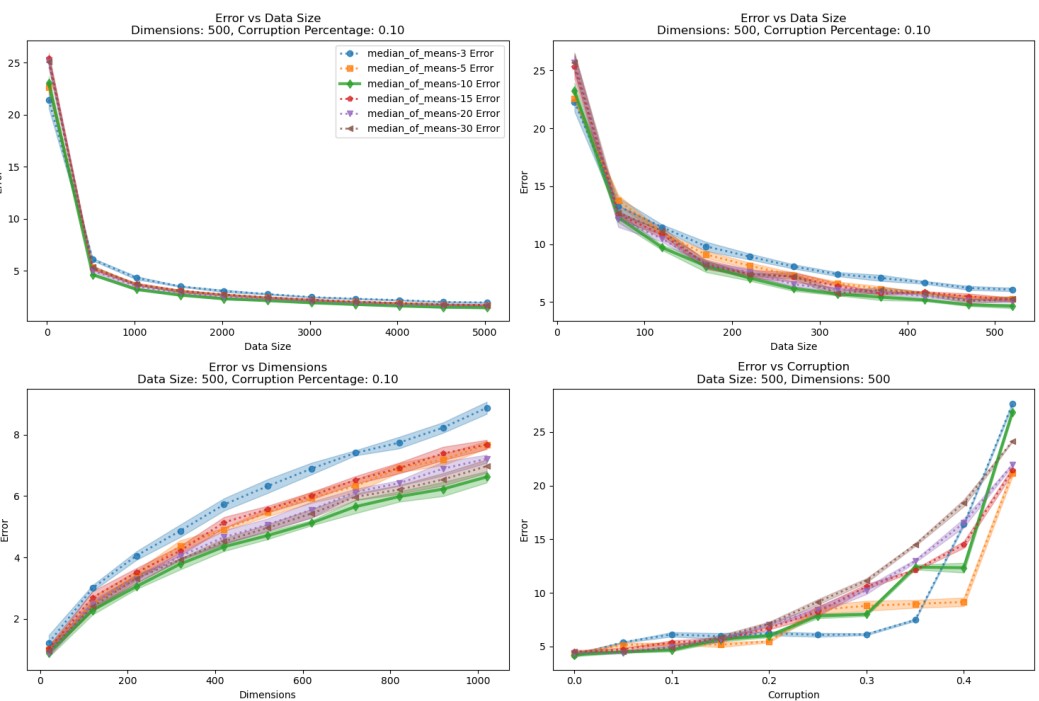

Figure 40: Median Of Means - Number of Chunks $k$: Large Diminishing Covariance, Additive Variance Shell Noise

**LRV Weighting Procedure** Here we explore the weighting procedure in LRV. First, we examine the parameter $C$ in the weighting procedure for LRV. This parameter is used when we calculate weights for each point, $x_i$ as $w_i = \exp(-\|x_i - a\|^2/(C * s^2))$. We vary $C$ in the set $[0.1, 0.5, 1, 5, 10, 20, 50]$. Results are shown in Figures 41, 42, 43, 44. We notice that performance may degrade with choices of $C$ that are too high or too low, such as with $C = 0.5$ and 50. We also notice that smaller choices of $C$ tend to degrade worse with greater corruption. To strike a balance, we select $C = 1$ throughout our experiments, which consistently performs among the best throughout the hyperparameter trials that we test, and is the default value used the original author's implementation of LRV. Although there are cases where larger choices of $C$ noticeably outperform $C = 1$, this is not robust as such choices may perform meaningfully worse over different noise distributions. For example, $C = 20$ noticeably outperforms $C = 1$ over Large Diminishing Covariance with Additive Variance Shell Noise, especially with larger $\eta$, but performs much worse over Identity Covariance with DKK Noise and Large Spherical Covariance with Additive Variance Shell Noise.

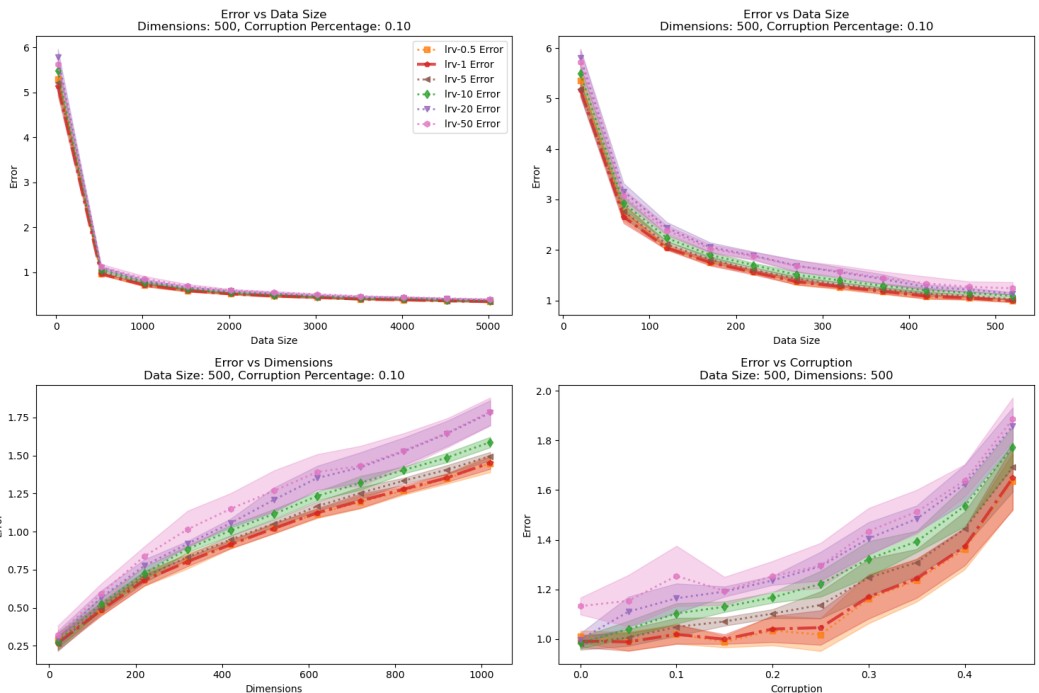

Figure 41: LRV - Choice Of $C$: Identity Covariance, DKK Noise

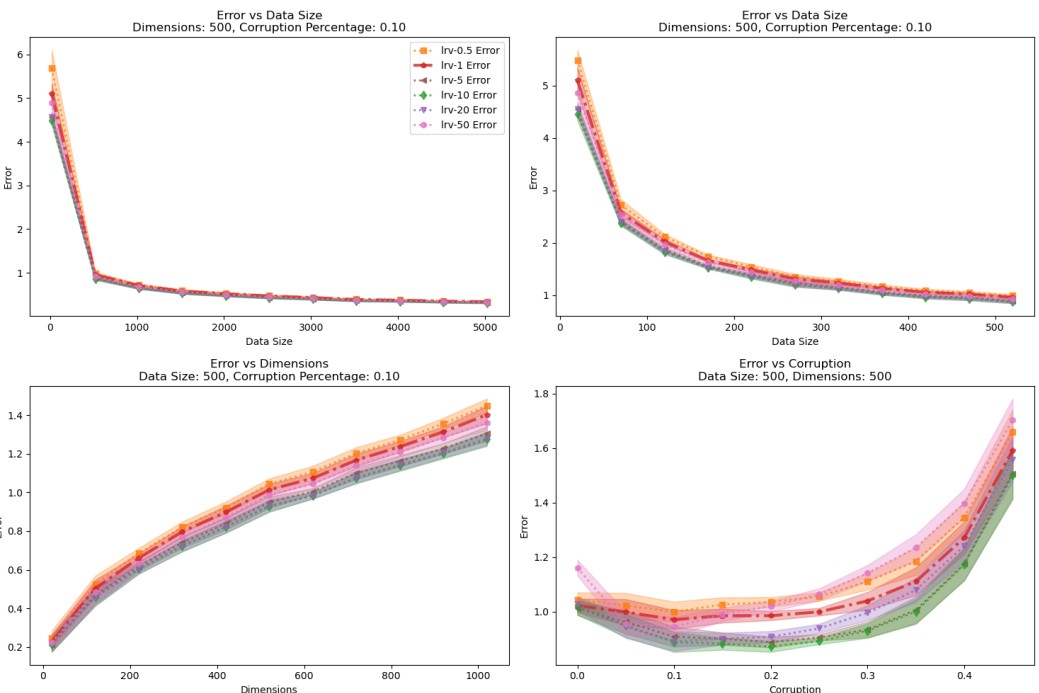

Figure 42: LRV - Choice Of $C$: Identity Covariance, In Distribution Noise

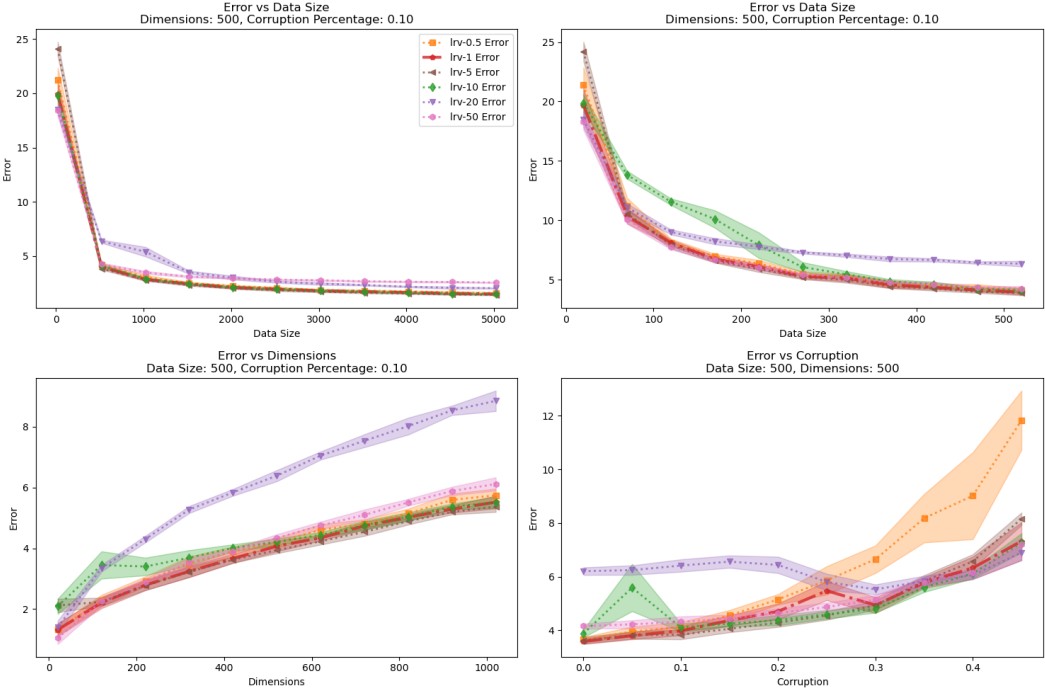

Figure 43: LRV - Choice Of $C$: Large Spherical Covariance, Additive Variance Shell Noise

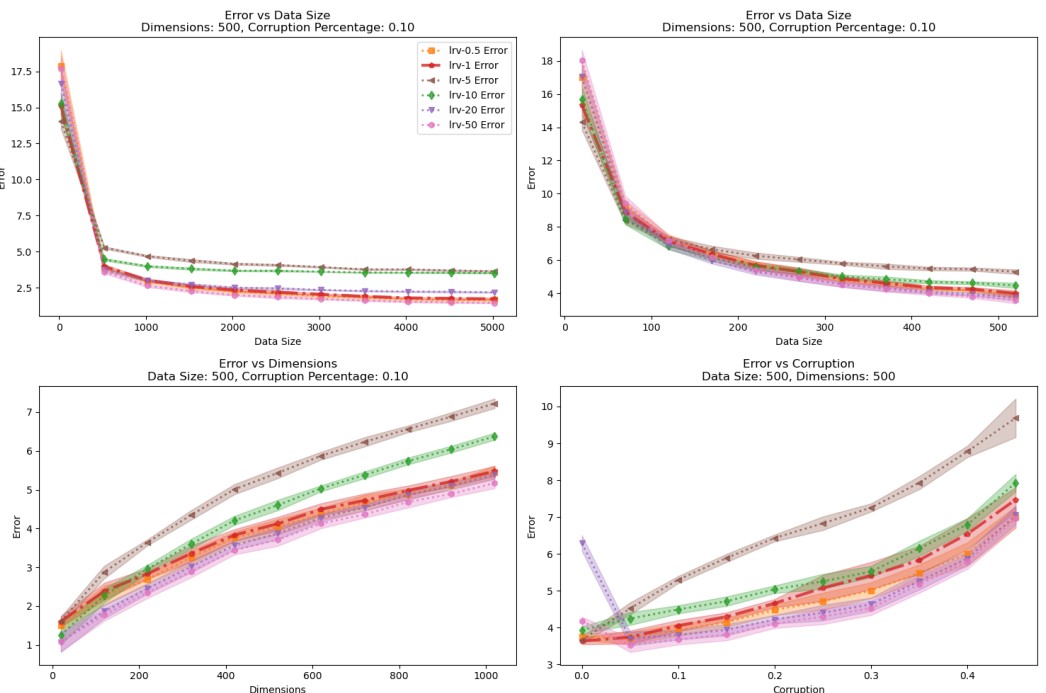

Figure 44: LRV - Choice Of $C$: Large Diminishing Covariance, Additive Variance Shell Noise

We additionally compare the weighting procedure of LRV that we consider with one meant for more general distributions discussed by Lai et al. (2016). Rather than downweighting outliers, this alternate procedure completely prunes outliers by calculating a point, $\mu'$, analogous to the coordinate wise median, finding a ball centered at $\mu'$ that contains $1-\tau$ percentage of data points, and throwing away all points outside of this ball. Results are shown in Figures 45, 46, 47, 48. This general weighting procedure performs meaningfully worse than the Gaussian weighting procedure and degrades significantly worse with larger corruption across all of the distributions considered. However, it achieves similar results as data size increases. Since we focus on the low data size regime and synthetic data with Gaussian inliers, we only evaluate LRV with Gaussian-based outlier downweighting.

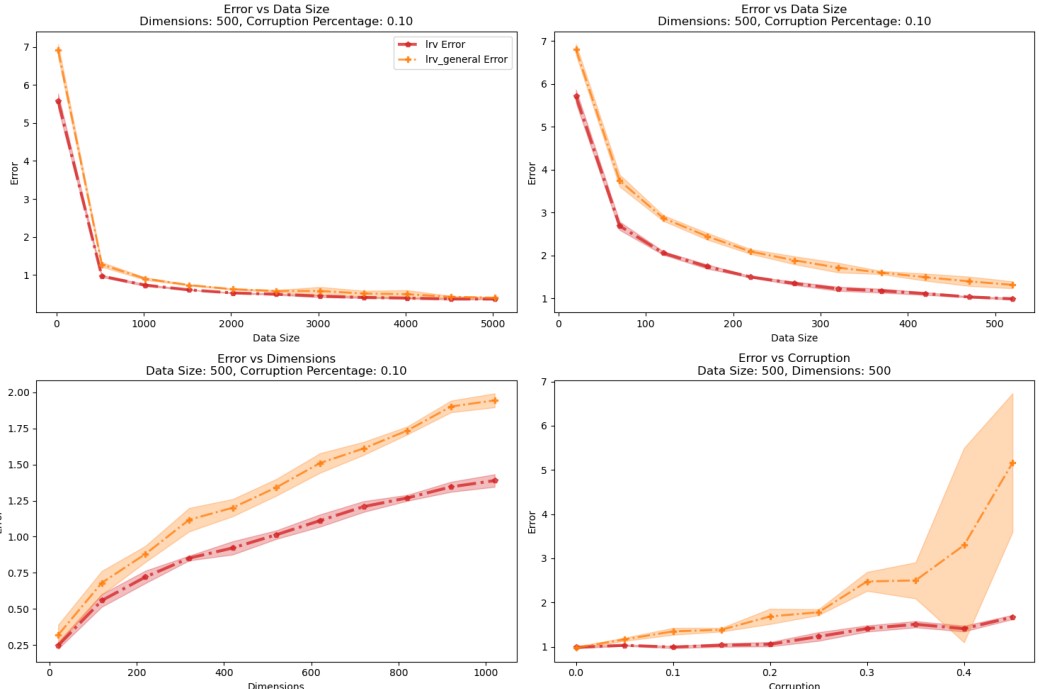

Figure 45: LRV - Gaussian Vs General Weighting: Identity Covariance, DKK Noise

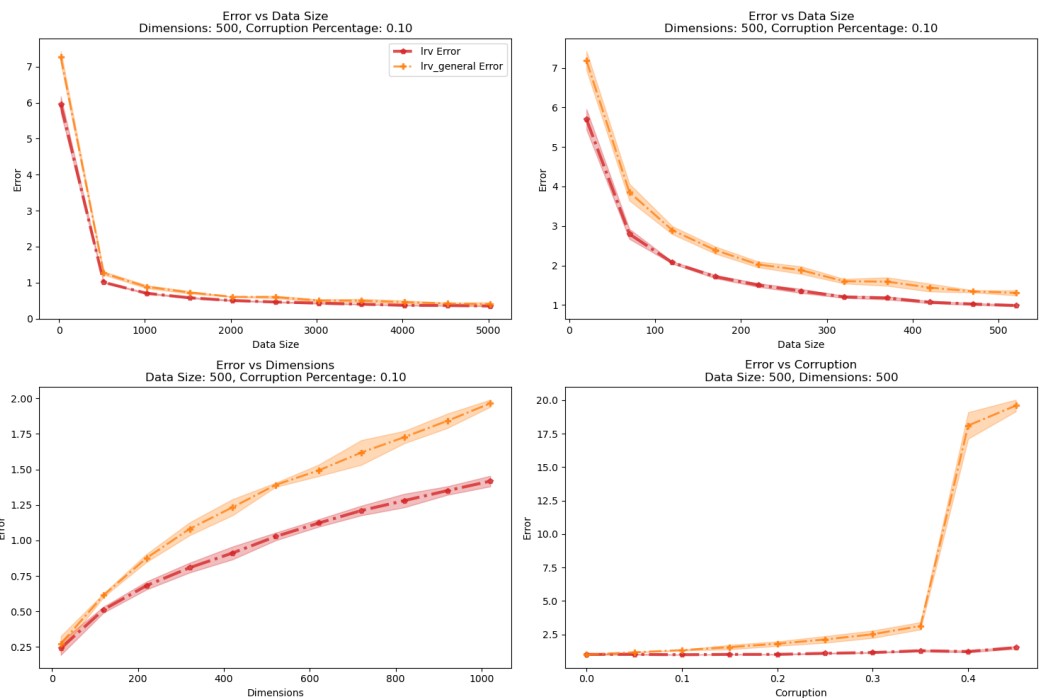

Figure 46: LRV - Gaussian Vs General Weighting: Identity Covariance, In Distribution Noise

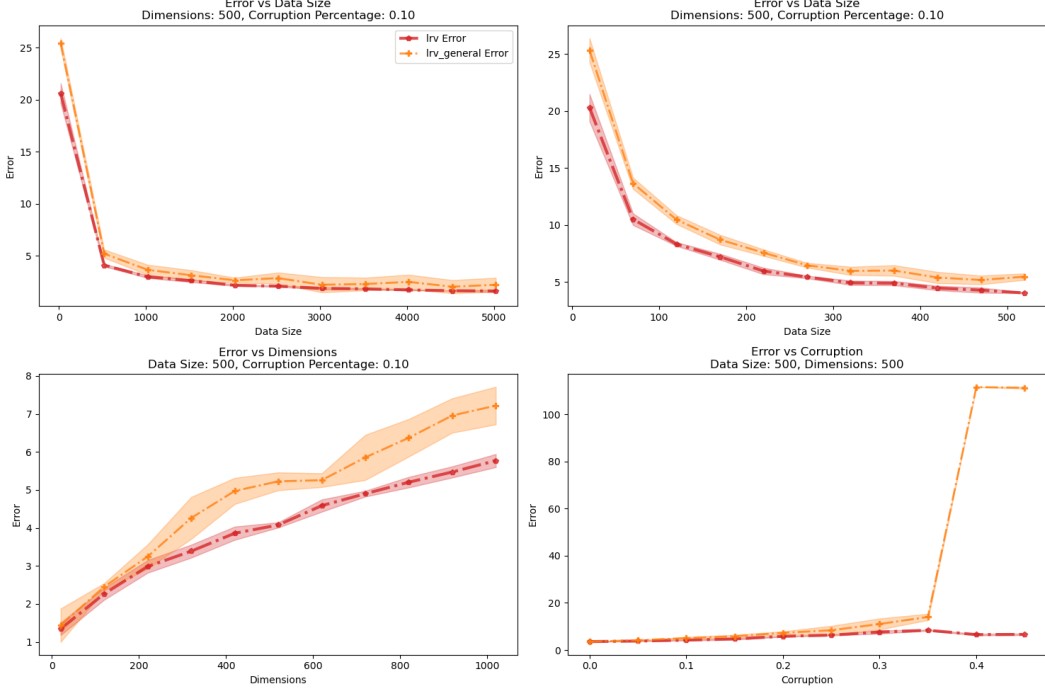

Figure 47: LRV - Gaussian Vs General Weighting: Large Spherical Covariance, Additive Variance Shell Noise

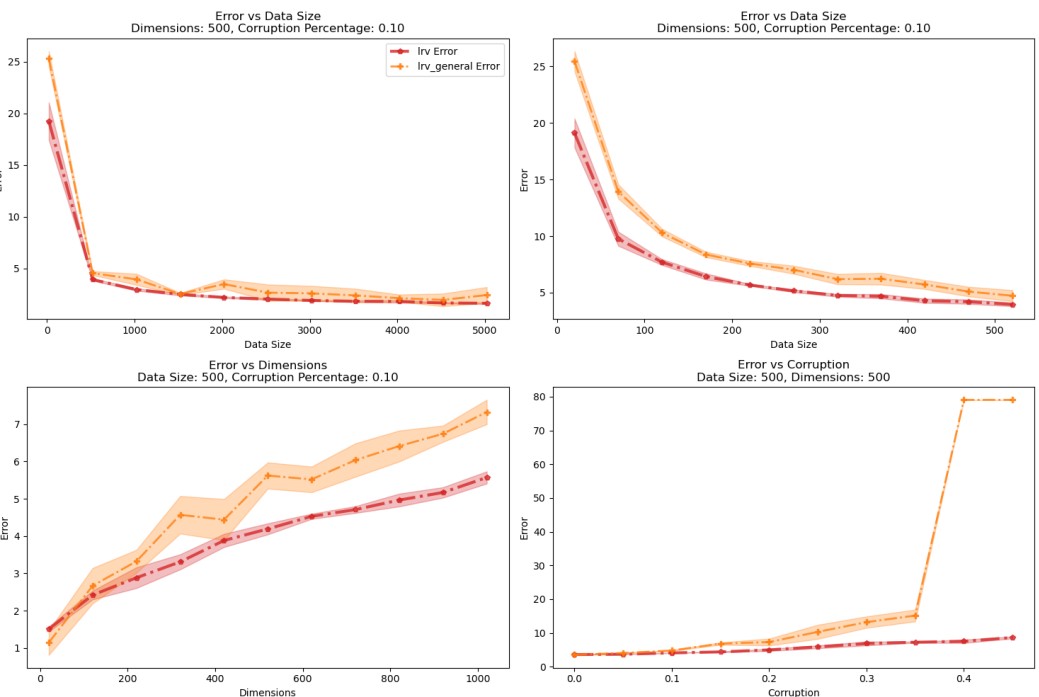

Figure 48: LRV - Gaussian Vs General Weighting: Large Diminishing Covariance, Additive Variance Shell Noise

**Eigenvalue Pruning Tail Threshold**   Here we explore the pruning routine in ev_filtering_low_n. First, we examine the parameter $\gamma$ in the pruning threshold for ev_filtering_low_n. $\gamma$ weights the expectation that the Gaussian concentration inequality gives for how many points will surpass a certain value; larger values correspond to less aggressive pruning. We vary $\gamma$ in the set $[0.5, 1, 2.5, 5, 10, 20, 50]$. Results are shown in Figures 49, 50, 51, 52. We find that using values of $\gamma$ that are too small result in significantly worse error. Setting $\gamma = 0.5$ or $\gamma = 1$ both achieve performance identical to sample_mean because the pruning threshold is too sensitive, performing significantly worse than all other choices of $\gamma$, as it determines all data to be outliers. We note that when all data is determined to be outliers, we simply return sample_mean. However, using reasonably sized $\gamma$ results in mostly similar performance across distributions. Notably, larger values of $\gamma$ tend to perform better over large diminishing covariance with additive variance shell noise, especially with larger $n$. We select $\gamma = 5$ throughout our experiments.

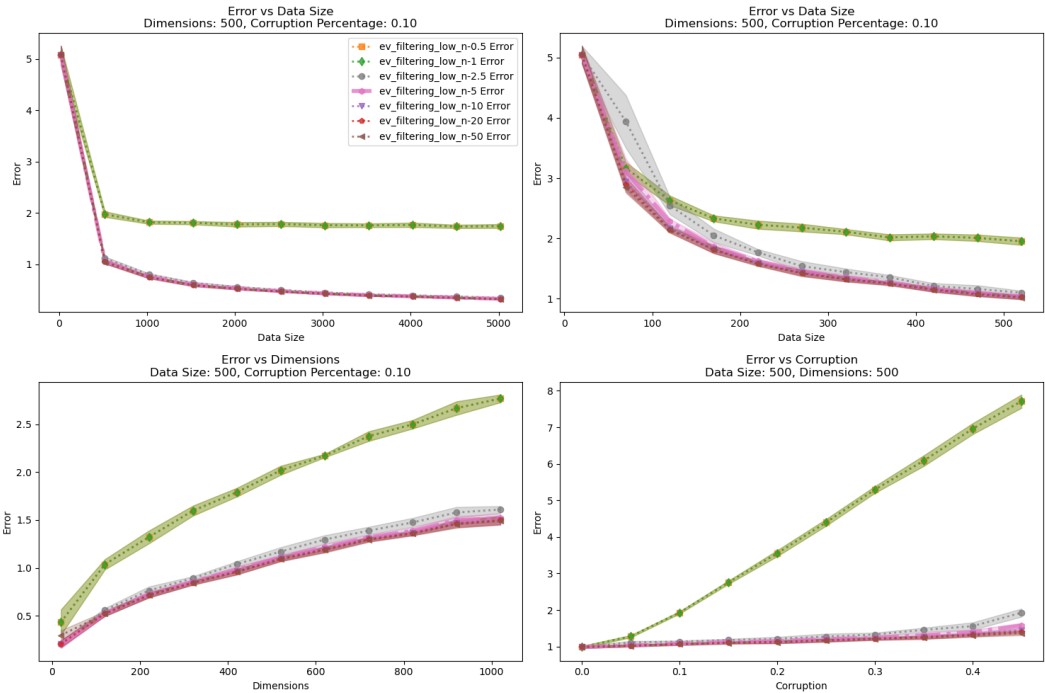

Figure 49: Eigenvalue Pruning - Choice Of $\gamma$: Identity Covariance, DKK Noise

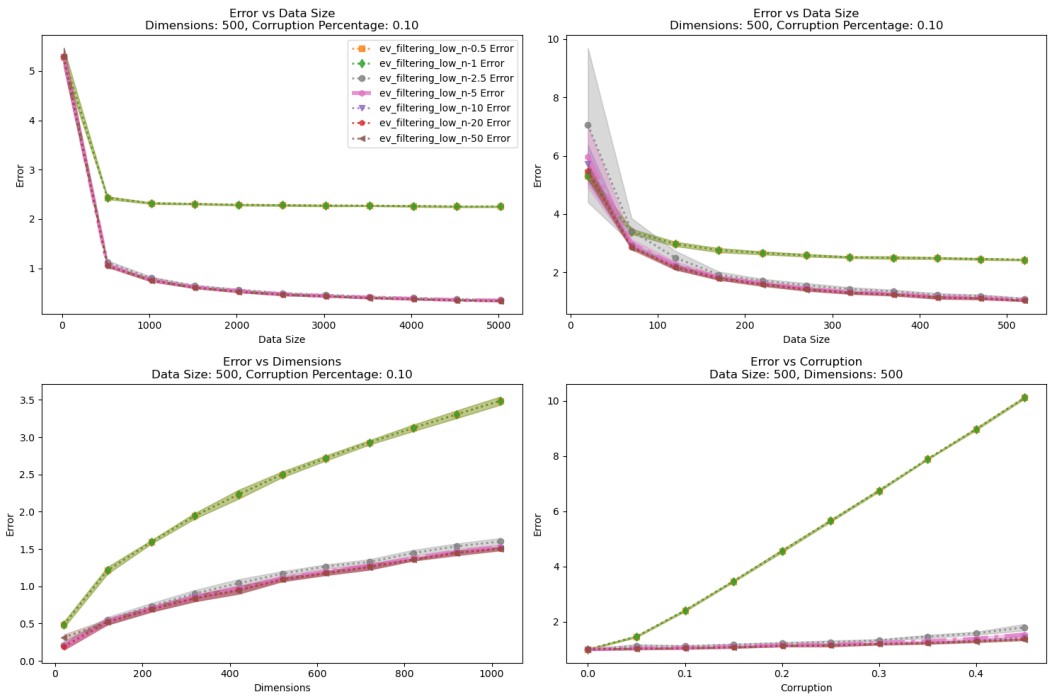

Figure 50: Eigenvalue Pruning - Choice Of $\gamma$: Identity Covariance, In Distribution Noise

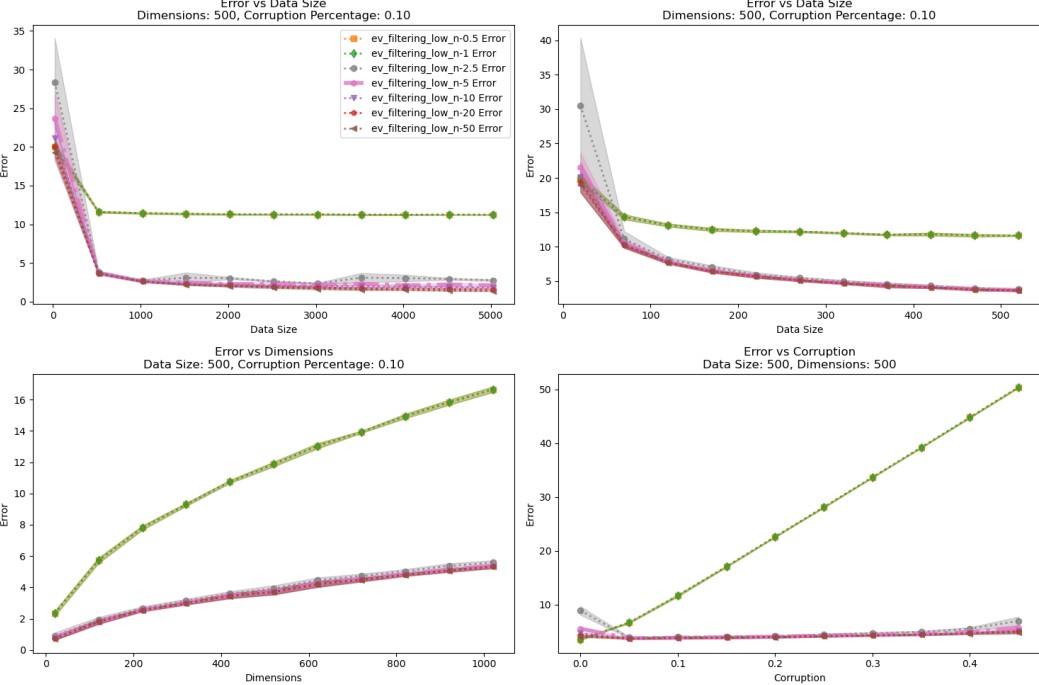

Figure 51: Eigenvalue Pruning - Choice Of $\gamma$: Large Spherical Covariance, Additive Variance Shell Noise

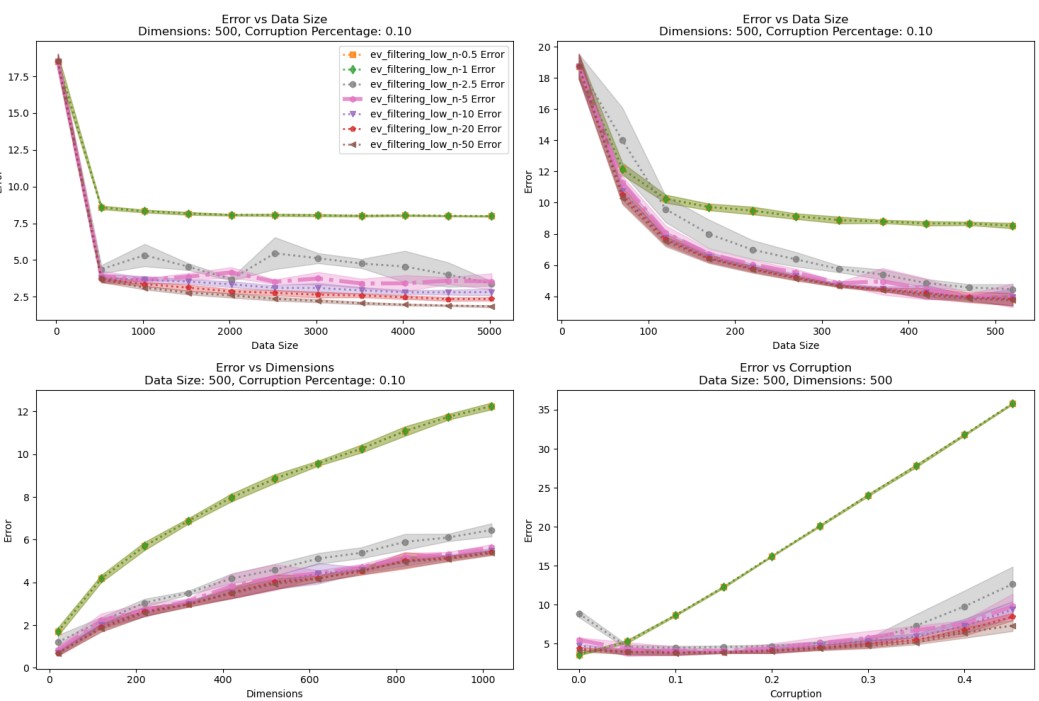

Figure 52: Eigenvalue Pruning - Choice Of $\gamma$: Large Diminishing Covariance, Additive Variance Shell Noise

We also explore ev_filtering_low_n using two alternate pruning methods not explicitly based on the Gaussian assumption as the current one is: randomized pruning and fixed pruning,. Randomized Pruning removes points based on a random scaling of the largest deviation in the dataset. Define $T$ as the largest deviation of a point projected onto the top eigenvector from the median of the projected points. Draw $Z$ from the distribution on $[0, 1]$ with probability density function $2x$. Then, prune all points whose projected distance onto the top eigenvector is at least $TZ$. This randomized pruning method is derived from the mean estimation algorithm for unknown covariance distributions by Diakonikolas et al. (2017a). Fixed Pruning simply prunes the $0.5\tau$ percentage of points whose projection onto the top eigenvector is furthest from the median of the projected points at every iteration. This is identical to the pruning method in QUE_low_n, with projected deviations being used as "outlier scores", instead of the quantum entropy scores used in QUE_low_n. Results are shown in Figures 53, 54, 55, 56. We find that both randomized and fixed pruning are able to match or slightly outperform the standard Gaussian pruning method. However, we note that unlike in QUE_low_n, fixed pruning could potentially result in catastrophic error. In particular, if corruption is uniformly distributed across $O(d)$ orthogonal clusters, then ev_filtering_low_n may take $O(d)$ runs to return an outlier, since it can only prune in one direction at once. But with fixed pruning, each iteration will prune too many outliers in each direction. We only evaluate Gaussian pruning to follow the conventions of Diakonikolas et al. (2017a).

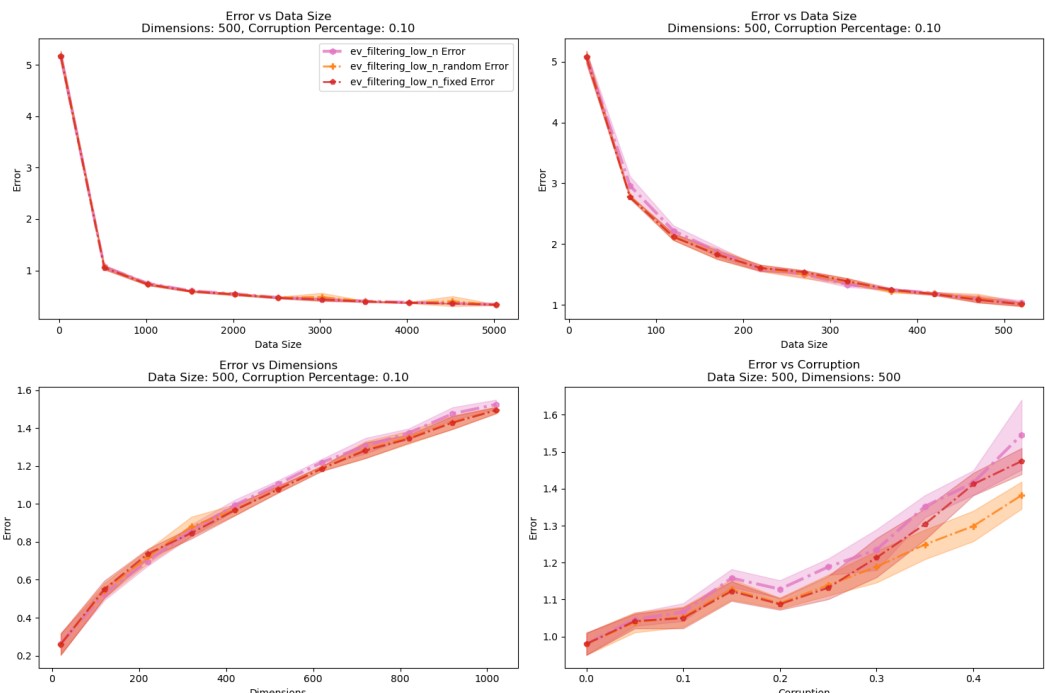

Figure 53: Eigenvalue Pruning - Pruning Method: Identity Covariance, DKK Noise

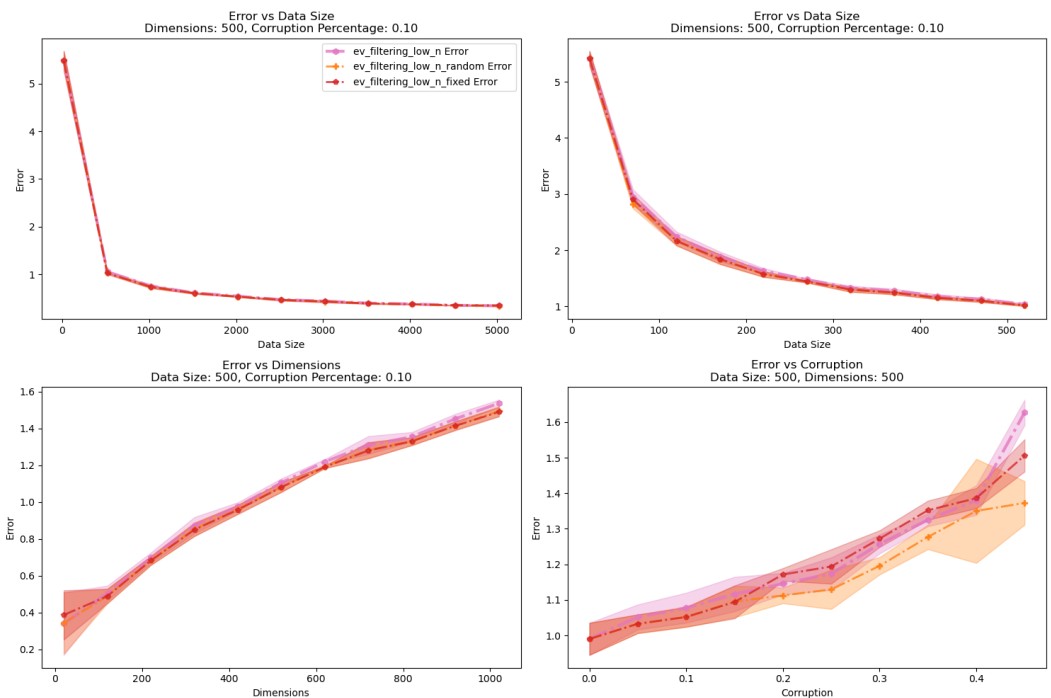

Figure 54: Eigenvalue Pruning - Pruning Method: Identity Covariance, In Distribution Noise

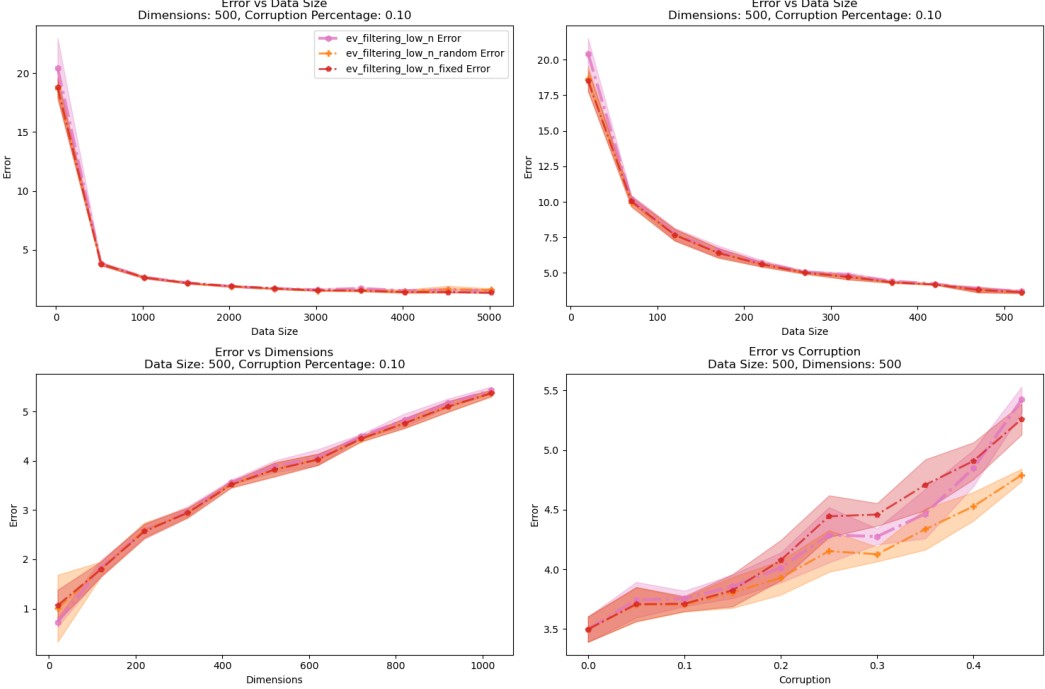

Figure 55: Eigenvalue Pruning - Pruning Method: Large Spherical Covariance, Additive Variance Shell Noise

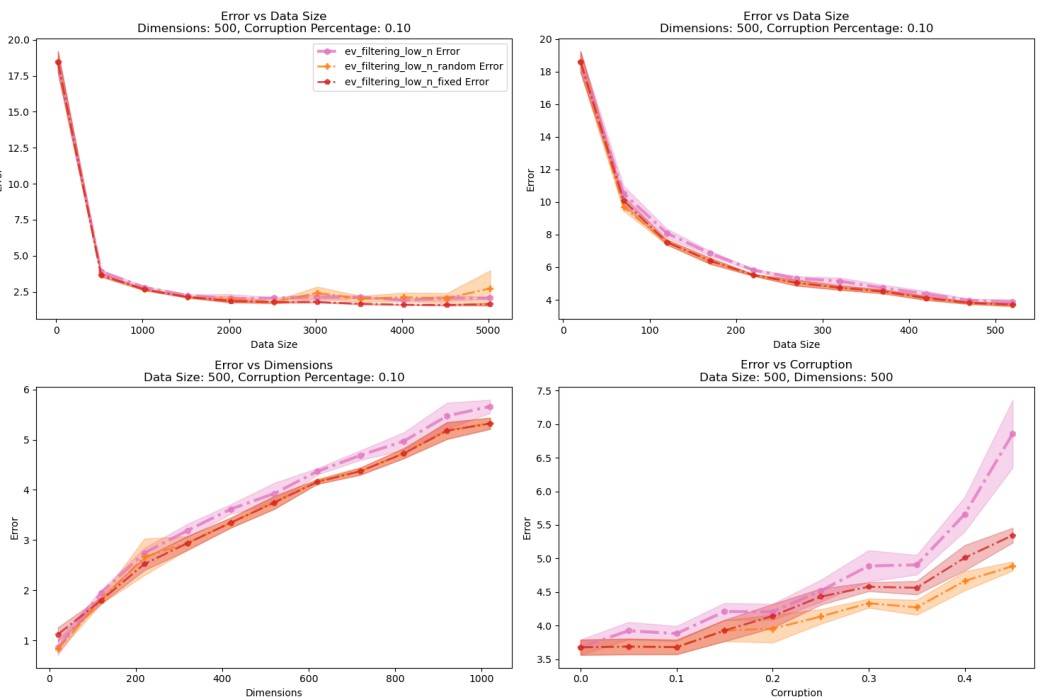

Figure 56: Eigenvalue Pruning - Pruning Method: Large Diminishing Covariance, Additive Variance Shell Noise

**Projected Gradient Descent Iterations** Here we explore the number of iterations parameter, $\gamma$ for PGD. We choose values for $\gamma$ in the set $[1, 5, 10, 15, 20]$. Results are shown in Figures 57, 58, 59, 60. We find that low choices of $\gamma$ result in significantly worse performance, while higher choices perform roughly equally. Notably, when $\gamma$ is set equal to 10, PGD performs much worse over Identity Covariance with DKK Noise, especially under large $n$, despite this choice of $\gamma$ performing among the best across other distributions. This suggests that larger choices of $\gamma$ may be necessary for PGD to be robust across different corruption schemes. We note that the runtime of PGD increases approximately linearly with respect to $\gamma$, so there is a meaningful tradeoff when using larger values of $\gamma$. We set $\gamma = 15$ across our experiments because it is the lowest $\gamma$ that performs among the best across the distributions tested.

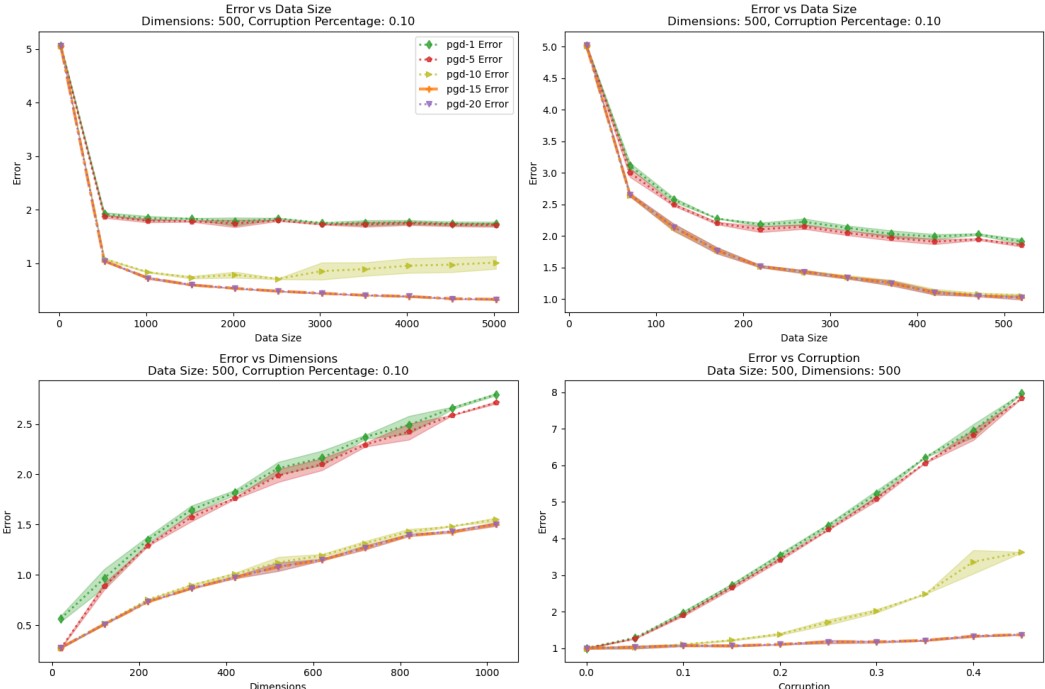

Figure 57: Projected Gradient Descent - Number Of Iterations $\gamma$: Identity Covariance, DKK Noise

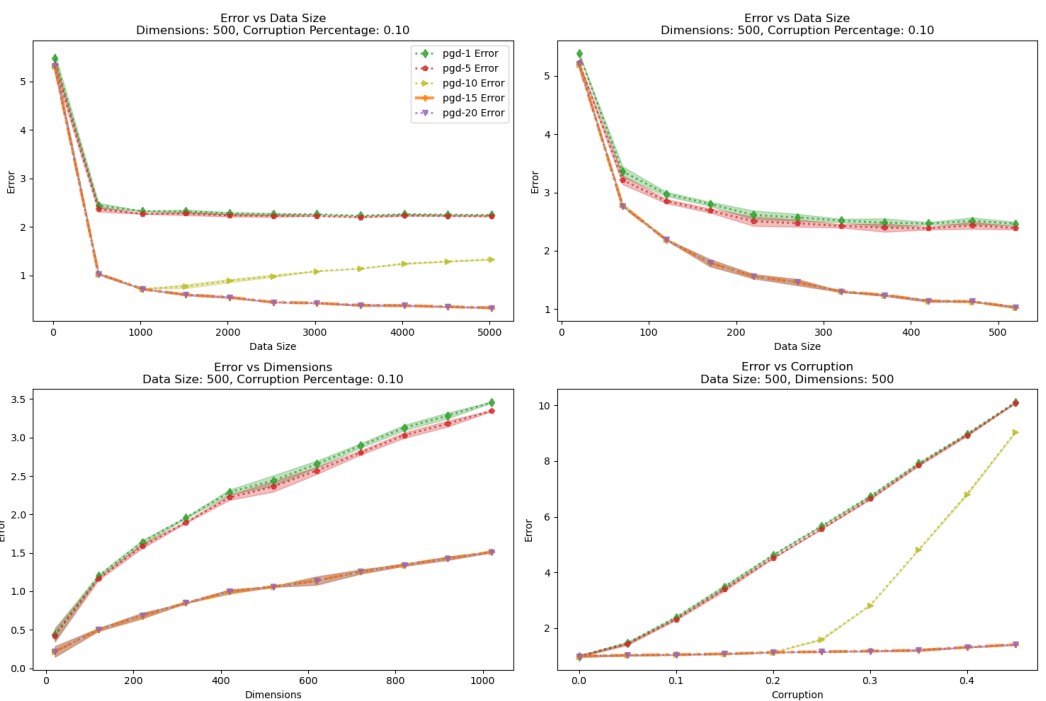

Figure 58: Projected Gradient Descent - Number Of Iterations $\gamma$: Identity Covariance, In Distribution Noise

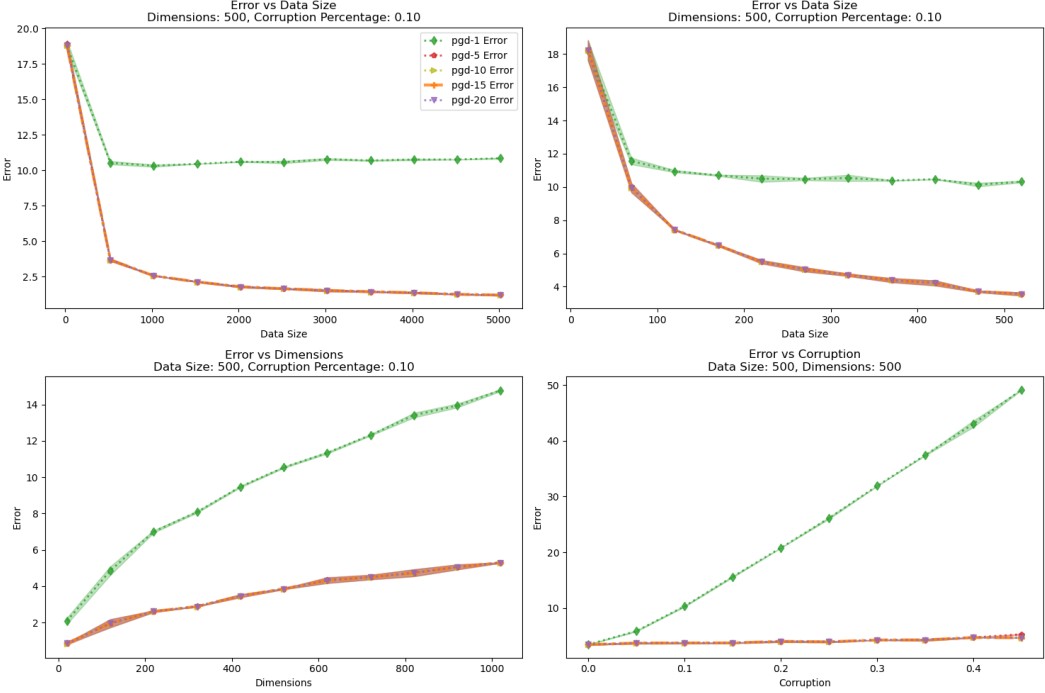

Figure 59: Projected Gradient Descent - Number Of Iterations $\gamma$: Large Spherical Covariance, Additive Variance Shell Noise

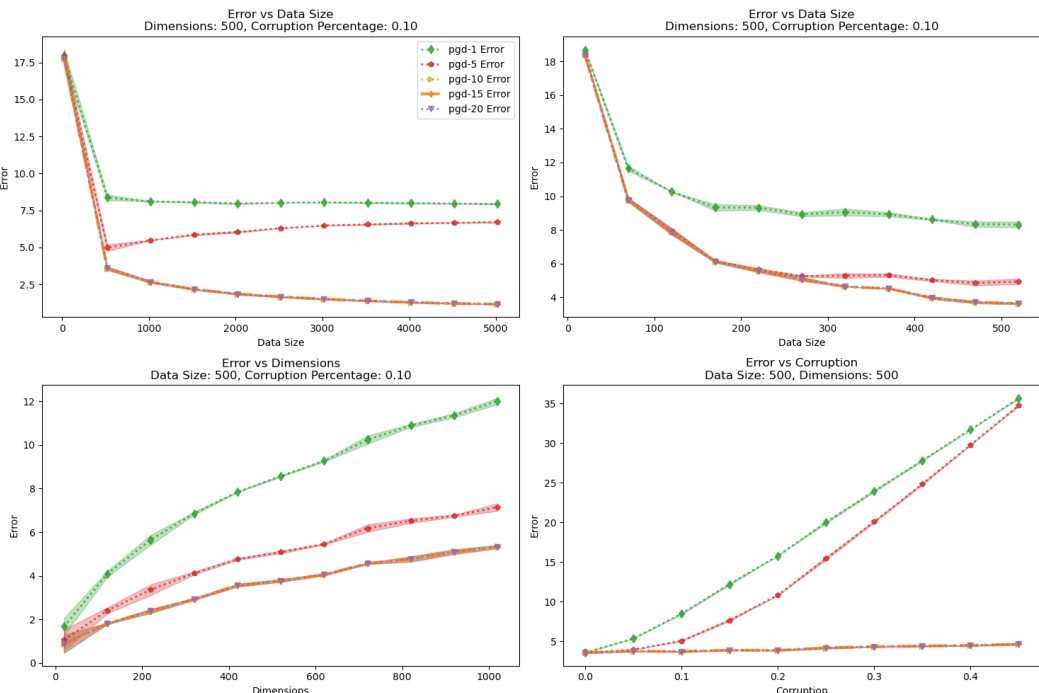

Figure 60: Projected Gradient Descent - Number Of Iterations $\gamma$: Large Diminishing Covariance, Additive Variance Shell Noise

## A.6 Robustness To Expected Corruption

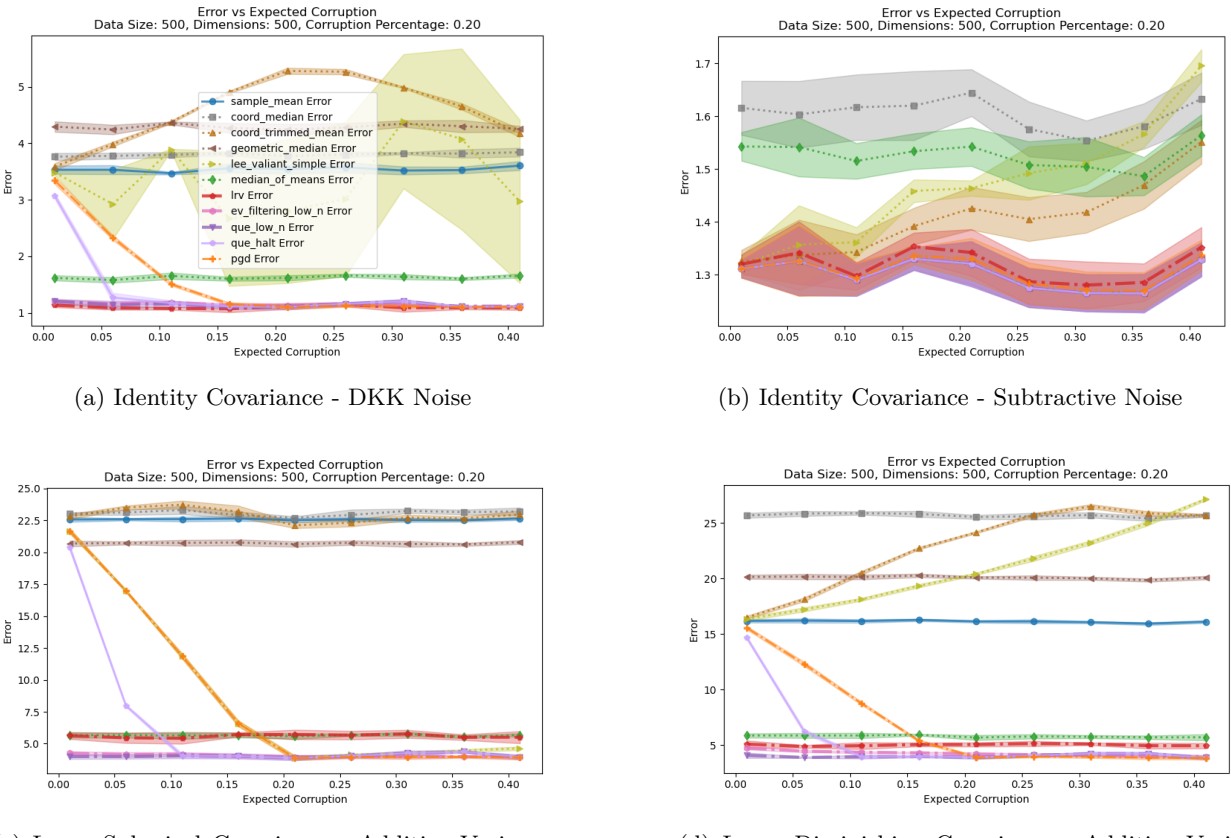

(a) Identity Covariance - DKK Noise

(b) Identity Covariance - Subtractive Noise

(c) Large Spherical Covariance - Additive Variance Shell Noise

(d) Large Diminishing Covariance - Additive Variance Shell Noise

Figure 61: Robustness To Expected Corruption: Error vs Expected Corruption $\tau$

We examine robustness to expected corruption, $\tau$. This is a hyperparameter for ev_filtering_low_n, QUE_low_n, PGD, lee_valiant_simple, and coord_trimmed_mean. In ev_filtering_low_n, $\tau$ only plays a soft role as a slack term in the filtering step. In QUE_low_n, $\tau$ controls the number of points that are pruned in every iteration of the algorithm, but the number of iterations is unbounded. In lee_valiant_simple, and coord_trimmed_mean, $\tau$ explicitly controls the amount of data that is pruned in total. In PGD, $\tau$ controls the space of feasible outlier weights. We evaluate error as expected corruption, $\tau$, varies from $\tau = 0.01$ to $\tau = 0.46$ with true corruption fixed as $\eta = 0.20$. Otherwise, the experiment setup remains the same as seen previously. As in Appendix A.5, we replicate experiments over Identity Covariance with DKK Noise and with Subtractive Noise; Large Spherical Covariance with Additive Variance Shell Noise; and Large Diminishing Covariance with Additive Variance Shell Noise. These results are shown in Figure 61, with all estimators included for reference. We include QUE_low_n with and without early halting.

We find that most estimators perform nearly identically regardless of the choice of $\tau$, except for PGD, which performs nearly identically when $\tau$ is an upper bound on true corruption $\eta$ but degrades with underestimates of $\eta$. QUE_low_n without early halting performs well throughout choices of $\tau$. With smaller choices of $\tau$, it will prune significantly less points at each iteration, but will run for more iterations until the corruption detection threshold is passed, while for larger choices of $\tau$, it will prune more points at each iteration, but will run for less iterations until the corruption detection threshold is passed. QUE_low_n with early halting, which is used throughout the real world experiments, sees degradation with underestimates of $\tau$, but identical performance with overestimates. ev_filtering_low_n also performs nearly identically regardless of the choice of $\tau$, as expected by the soft dependency of the pruning threshold on $\tau$. lee_valiant_simple and coord_trimmed_mean both degrade noticeably as $\tau$ increases over Identity Covariance data with Subtractive

Noise and Large Diminishing Covariance data with Additive Variance Shell Noise; in both cases yielding error worse than sample_mean the more points they prune. Surprisingly, over Large Spherical Covariance with Additive Variance Shell Noise, lee_valiant_simple nearly exactly matches the performance of PGD, except with slightly worse degradation with large overestimates of $\tau$.

### A.7 Image Embedding Experiments

We evaluate algorithms on the problem of estimating the mean of embeddings of images generated by deep pretrained image models. As in the LLM experiment, we first examine the problem of mean estimation of image embeddings belonging to the same category, reporting LOOCV error. We then examine a corrupted distribution where images belonging to one category are considered inliers and those belonging to another are considered outliers. We utilize a set of images of cats and dogs from the CIFAR10 dataset (Krizhevsky, 2009). We embed these images using 4 deep pretrained image models of varying embedding dimensions: ResNet-18, ResNet-50 (He et al., 2015), MobileNet V3 (Howard et al., 2019), and EfficientNet B0 (Tan & Le, 2020). ResNet-18 has an embedding dimension of 512, MobileNet V3 has one of 960, EfficientNet B0 has one of 1280, and ResNet-50 has one of 2048.

**Common Category Images**  Here we examine LOOCV error vs data size on embeddings of images of cats. We vary data size from $n = 10$ to $n = 1000$. Otherwise, experiments are run identically to the LLM experiment, fixing expected corruption $\eta = 0.1$, employing the trace scaling heuristic on ev_filtering_low_n and QUE_low_n, the halting heuristic on QUE_low_n, and averaging results over 5 runs. We note that, as in the LLM experiments, employing the halting heuristic on ev_filtering_low_n does not improve performance. Results are shown in Figure 62.

As in the LLM experiment, we observe that no algorithm significantly outperforms sample_mean, despite the nontrivial LOOCV error in each setting. As in the LLM experiment, ev_filtering_low_n tends to perform worse than other algorithms, which is unsurprising given its sensitivity to knowledge of the true covariance. Other robust mean estimation algorithms, including QUE_low_n, perform near identically to sample_mean.

**Corrupted Images**  For the corrupted case, we draw data $X \sim (1-\eta)P + \eta Q$, where the inlier distribution, $P$, consists of embeddings of images of cats, and the outlier distribution, $Q$, consists of embeddings of images of dogs. We fix data size $n = 1000$ to focus on the $n \approx d$ and $n < d$ regime. Otherwise, the experimental setup is identical to in the LLM experiments. Results are shown in Figure 63.

These results demonstrate similar trends to the LLM experiment. One key difference is that coord_trimmed_mean performs much worse than sample_mean here, compared to the LLM experiment where it tends to slightly outperform sample_mean. This suggests that naive pruning does not work well in this setting as outliers are not obvious, reinforcing that this is a difficult setting for robust mean estimation. Nonetheless, several robust mean estimators are able to perform well in this case. In particular, QUE_low_n is again the strongest performer, noticeably outperforming all other estimators and nearly matching good_sample_mean error across settings. Notably, this strong performance occurs even with $n$ much less than $d$, with relative performance remaining the same even with $n = 1000$ and $d = 2048$ in the ResNet-50 embedding case. Among the best estimators in the synthetic data case, PGD tends to perform similarly to sample_mean, except with large enough corruption across MobileNet V3 embeddings where it outperforms sample_mean; LRV tends to perform similarly to sample_mean except under low corruption, where it noticably degrades; and ev_filtering_low_n fails catastrophically throughout. As in the LLM experiments, median_of_means outperforms robust estimators that tend to perform better in the synthetic data cases. However, lee_valiant_simple no longer performs near optimally, suggesting its sensitivity to distributional assumptions, as expected due to its general poor performance over synthetic data experiments.

**Corruption vs Data Size**  We repeat experiments over the same corrupted data scheme but examine error vs data size. We fix true corruption $\eta = 0.1$, set expected corruption $\tau = \eta$, and vary data size $n$. We examine the performance of all estimators with data size ranging from $n = 100$ to $n = 5000$. We additionally provide a zoomed in plot, examining the performance of estimators excluding ev_filtering_low_n, coord_median, and

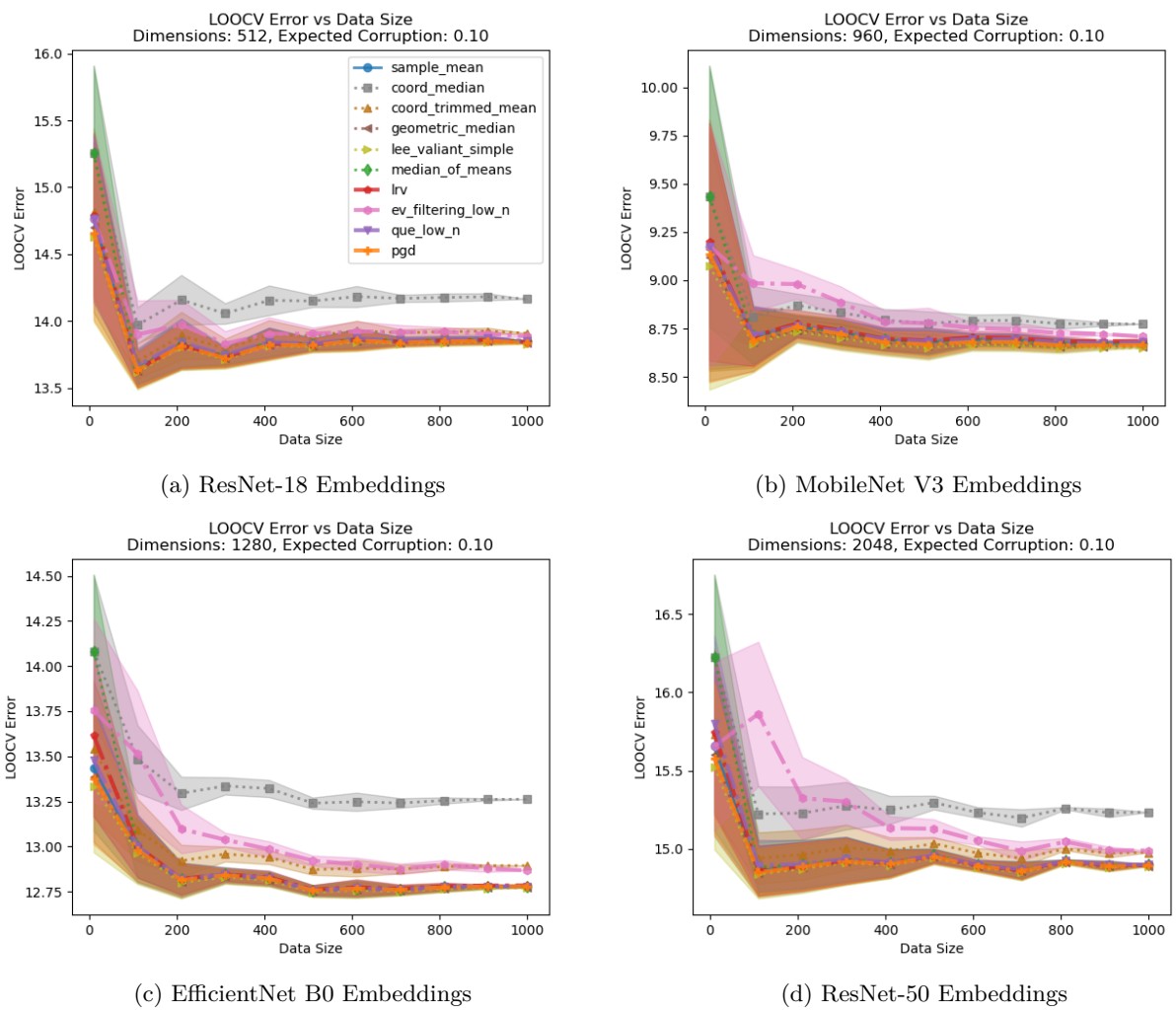

(a) ResNet-18 Embeddings

(b) MobileNet V3 Embeddings

(c) EfficientNet B0 Embeddings

(d) ResNet-50 Embeddings

Figure 62: LOOCV Error on Cat Image Embeddings

coord_trimmed_mean– which all fail catastrophically – with data size from $n = 100$ to $n = 1000$. Results are shown in Figure 64.

We find that the relative performance of algorithms remains similar across data sizes. Particularly, even with very large $n$, such as $n = 5000$ and $d = 512$ under ResNet-18 Embeddings, only QUE_low_n and median_of_means consistently outperform sample_mean. PGD, LRV, and lee_valiant_simple tend to perform slightly worse than sample_mean. ev_filtering_low_n fails catastrophically regardless of data size, though the error stabilizes with larger $n$. These results suggest that the weakness of robust mean estimators over real world data distributions is not just confined to the low data size regime. Yet again, we find that QUE_low_n is the best performer, outperforming all other estimators and achieving near optimal performance throughout settings. Additionally, median_of_means does not show this same sensitivity to distributional assumptions as other estimators, and as in the synthetic data experiments, tends to perform near optimally with large enough $n$.

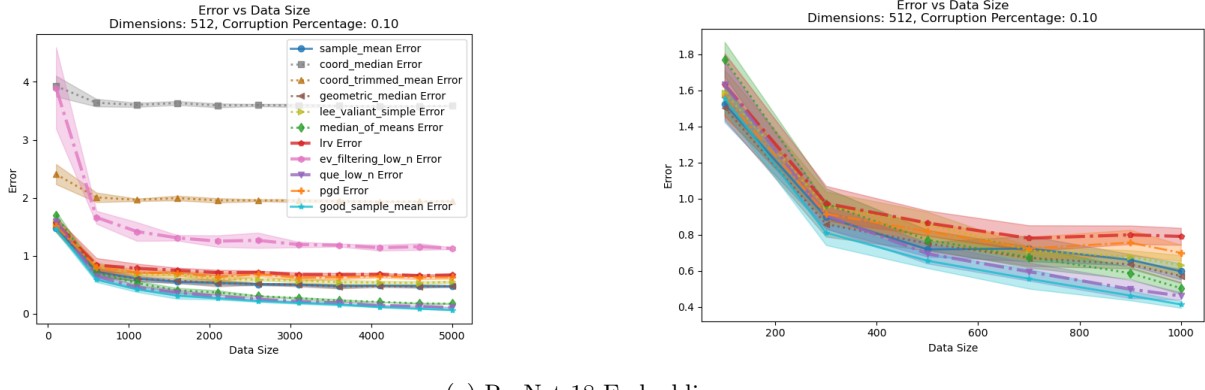

(a) ResNet-18 Embeddings

(b) MobileNet V3 Embeddings

(c) EfficientNet B0 Embeddings

(d) ResNet-50 Embeddings

Figure 63: Error on Cat Image Embeddings Corrupted with Dog Image Embeddings

(a) ResNet-18 Embeddings

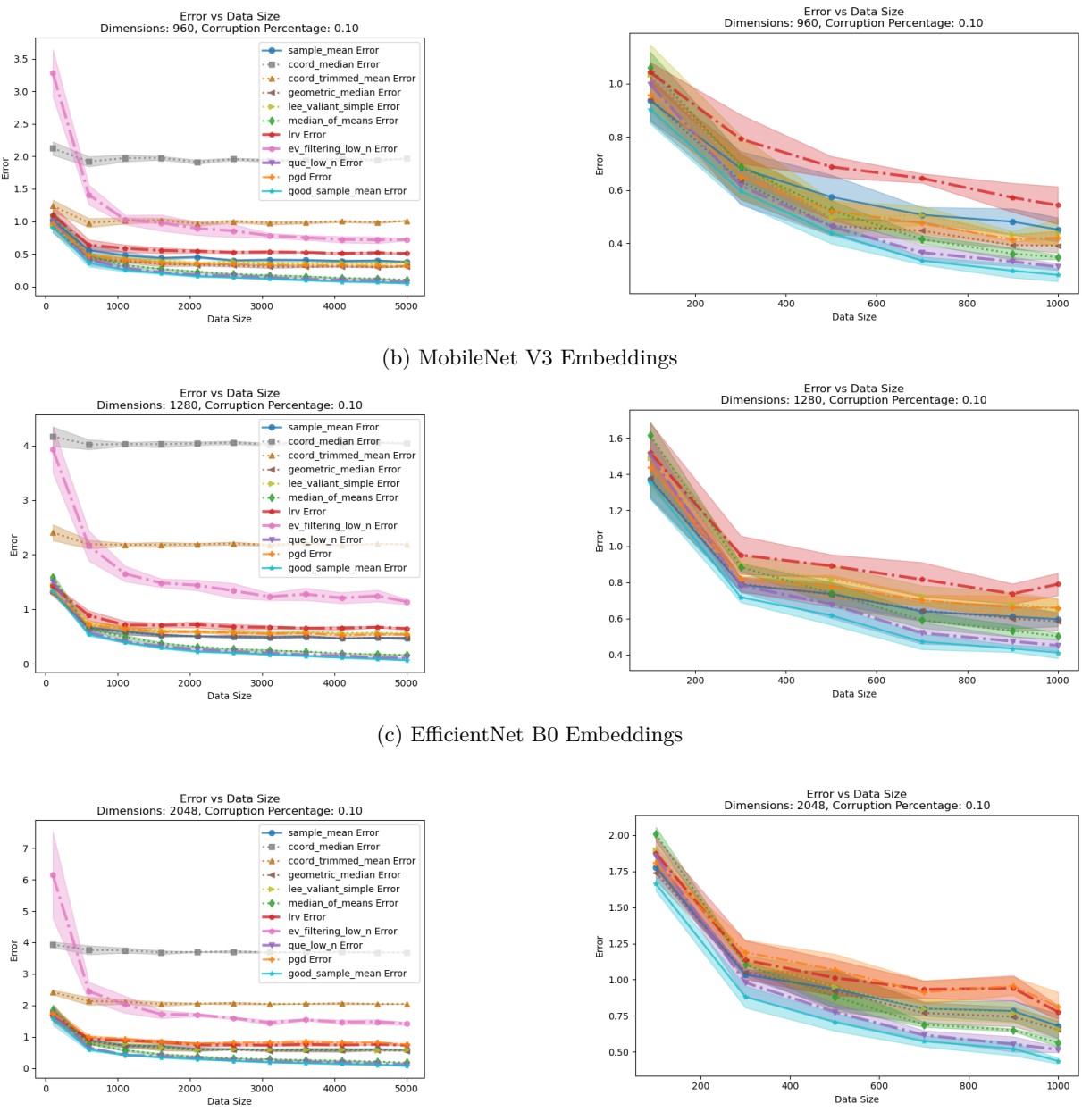

Figure 64: Error Vs Data Size on Corrupted Image Data

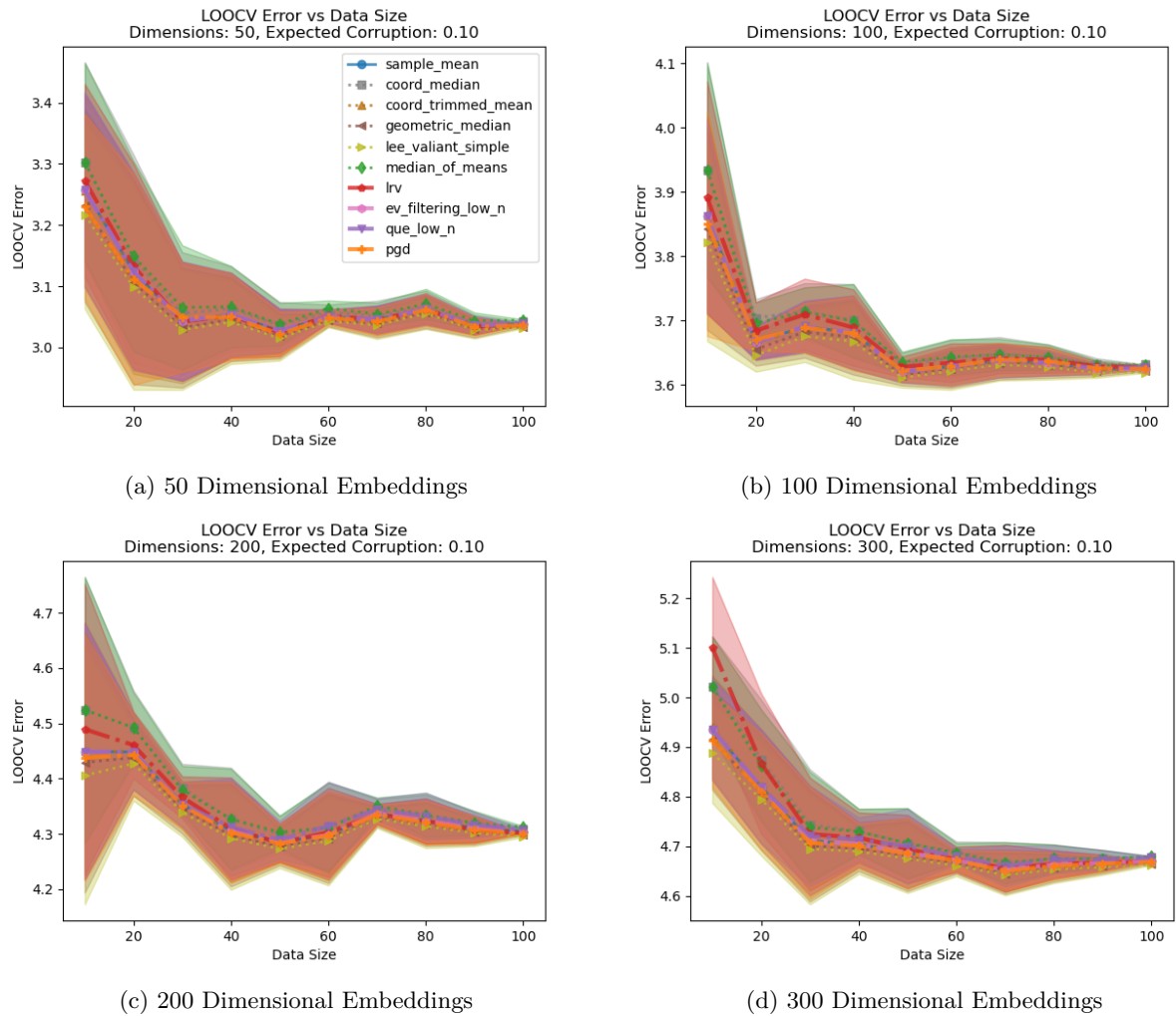

Figure 65: LOOCV Error on "Pleasant" GloVe Embeddings

## A.8 Word Embedding Experiments

We further evaluate algorithms on the problem of estimating the mean of non attention based embeddings of words. As in the LLM experiment, we first examine the problem of mean estimation over words belonging in the same category, reporting LOOCV error. We then examine a corrupted distribution where words belonging to one category are considered inliers and those belonging to another are considered outliers. We examine four different pretrained GloVe (Pennington et al., 2014) models from GluonNLP[6] generating 50, 100, 200, and 300 dimensional embeddings. We utilize datasets of 100 pleasant words and 100 unpleasant words from Aboagye et al. (2023). The very limited data size available under this setting provides a valuable real world test for robust estimators under low data size.

**Common Category Words**    Here we examine LOOCV error vs data size on embeddings of "pleasant" words. Experiments are run identically to the LLM experiment, employing the trace scaling heuristic on ev_filtering_low_n and QUE_low_n, the halting heuristic on QUE_low_n, and averaging results over 5 runs. Results are shown in Figure 65.

---

[6]https://github.com/dmlc/gluon-nlp

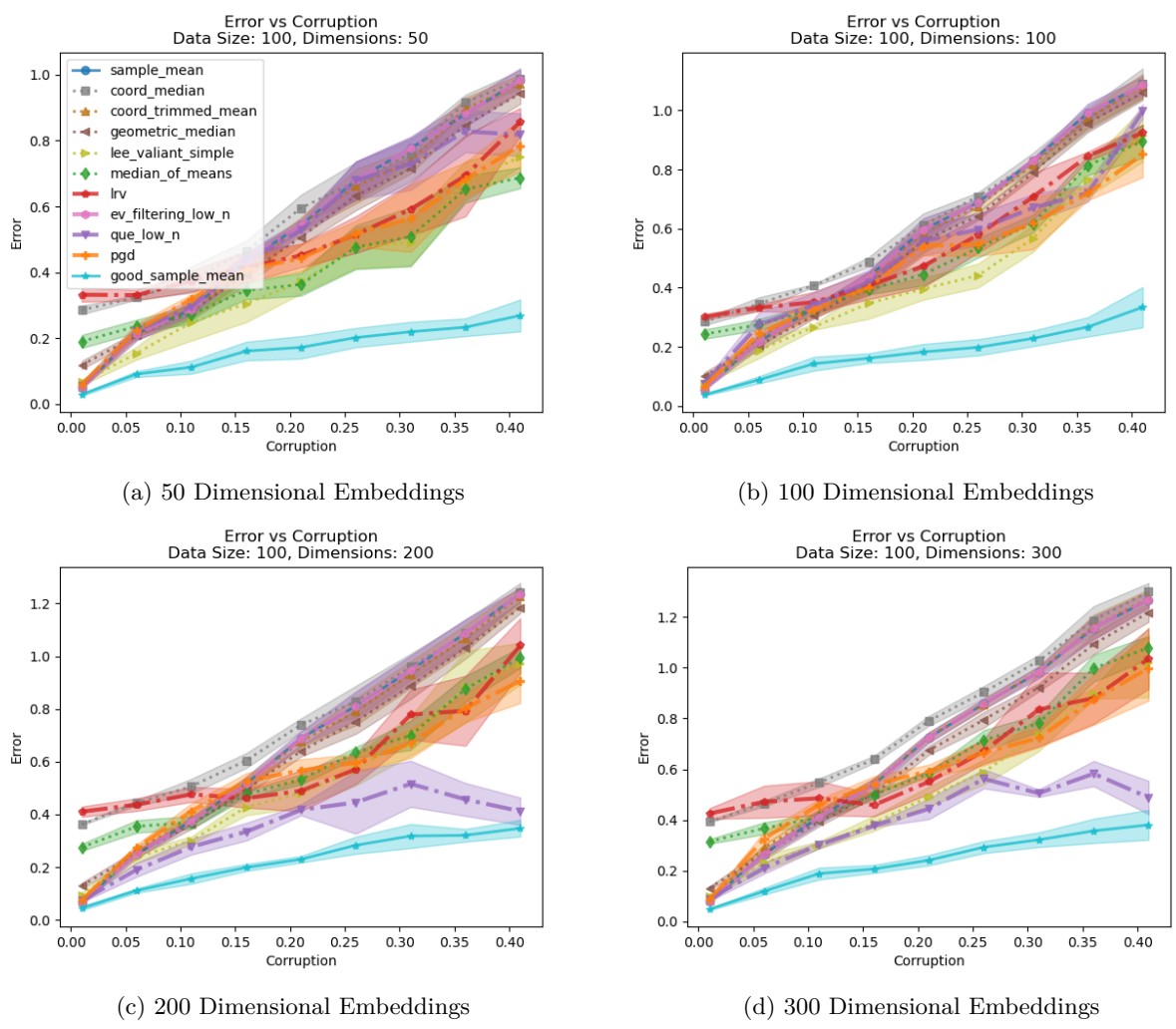

(a) 50 Dimensional Embeddings

(b) 100 Dimensional Embeddings

(c) 200 Dimensional Embeddings

(d) 300 Dimensional Embeddings

Figure 66: Error on "Pleasant" Embeddings Corrupted with "Unpleasant" Embeddings

As in the LLM experiment, we observe that no algorithm significantly outperforms sample_mean, despite the nontrivial LOOCV error in each setting. Moreover, we observe that median_of_means consistently achieves error slightly worse than sample_mean, which is not seen in the LLM experiments, suggesting the algorithm's sensitivity to distributional assumptions. However, unlike in the LLM experiment ev_filtering_low_n does not fail catastrophically here, instead nearly matching sample_mean. This is not unexpected given ev_filtering_low_n, and the trace estimate techniques sensitivity to distributional assumptions will sometime work – including this case. LRV also tends to perform worse than other algorithms, though this gap is not as large as in the LLM experiment.

**Corrupted Words** For the corrupted case, we draw data $X \sim (1-\eta)P + \eta Q$, where the inlier distribution, $P$, consists of embeddings of "pleasant" words, and the outlier distribution, $Q$, consists of embeddings of "unpleasant" words. This models a more extreme version of the case where ill-defined words may be placed in a category, inducing bias. This is a notable problem for word vectors, which do not take context into account (Hu et al., 2016). The experimental setup is identical to in the LLM experiments. Results are shown in Figure 66.

While these results are different from the LLM experiment, they demonstrate similar trends. In particular, QUE_low_n is again the strongest performer, noticeably outperforming all other estimators across the 200 and 300 dimensional cases, and never performing worse than sample_mean in the 50 and 100 dimensional

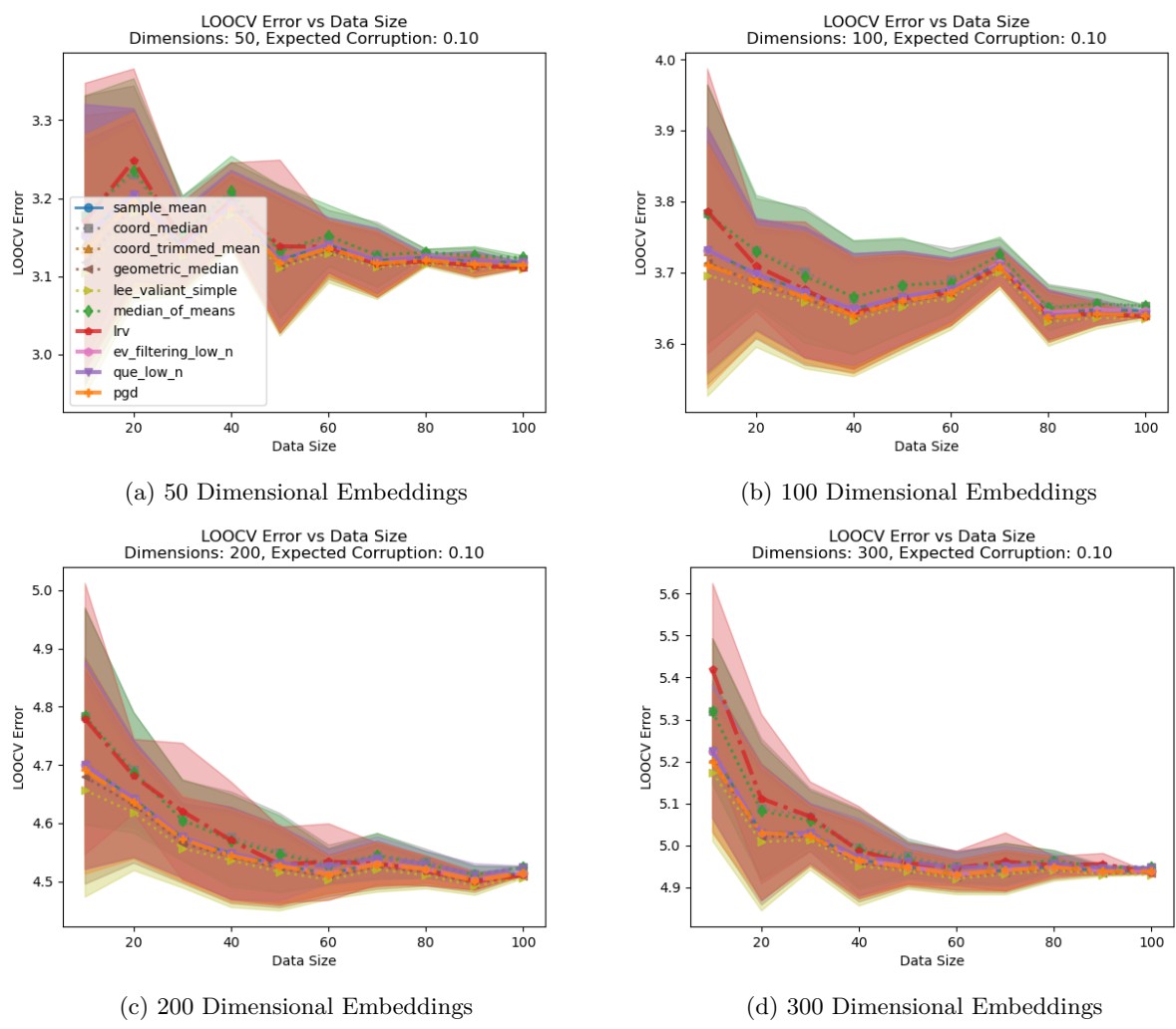

(a) 50 Dimensional Embeddings      (b) 100 Dimensional Embeddings

(c) 200 Dimensional Embeddings      (d) 300 Dimensional Embeddings

Figure 67: LOOCV Error on "Unpleasant" GloVe Embeddings

cases. Notably, this strong performance occurs even with $n$ much less than $d$. Unlike the LLM experiments, here QUE_low_n never approaches good_sample_mean, and is beat by other estimators in the 50 and 100 dimensional cases. lee_valiant_simple, which tended to perform similarly to QUE_low_n and nearly match good_sample_mean in the LLM experiments, does not perform as well in this case. It always beats sample_mean but does not come close to matching good_sample_mean and performs similarly to other estimators. Likewise, median_of_means does not perform as strongly here as in the LLM experiments and even performs worse than sample_mean over very low corruption. Supported by synthetic data results, this suggests the sensitivity of median_of_means and lee_valiant_simple to distributional assumptions. As in the LLM experiments, LRV tends to perform much worse than sample_mean under low corruption and outperform sample_mean slightly with higher corruption; PGD tends to outperform sample_mean slightly; and coord_median, coord_trimmed_mean, and geometric_median tend to perform similarly or slightly worse than sample_mean. As in the LOOCV experiments, ev_filtering_low_n simply matches sample_mean here.

**Additional Experiments** We perform additional experiments, swapping the roles of "pleasant" and "unpleasant" embeddings. We report LOOCV error vs data size on embeddings of "unpleasant" words in Figure 67. We report corrupted error vs data size on embeddings of "unpleasant" words corrupted with "pleasant" words in Figure 68. We observe the same trends as in the previous word embedding experiments.

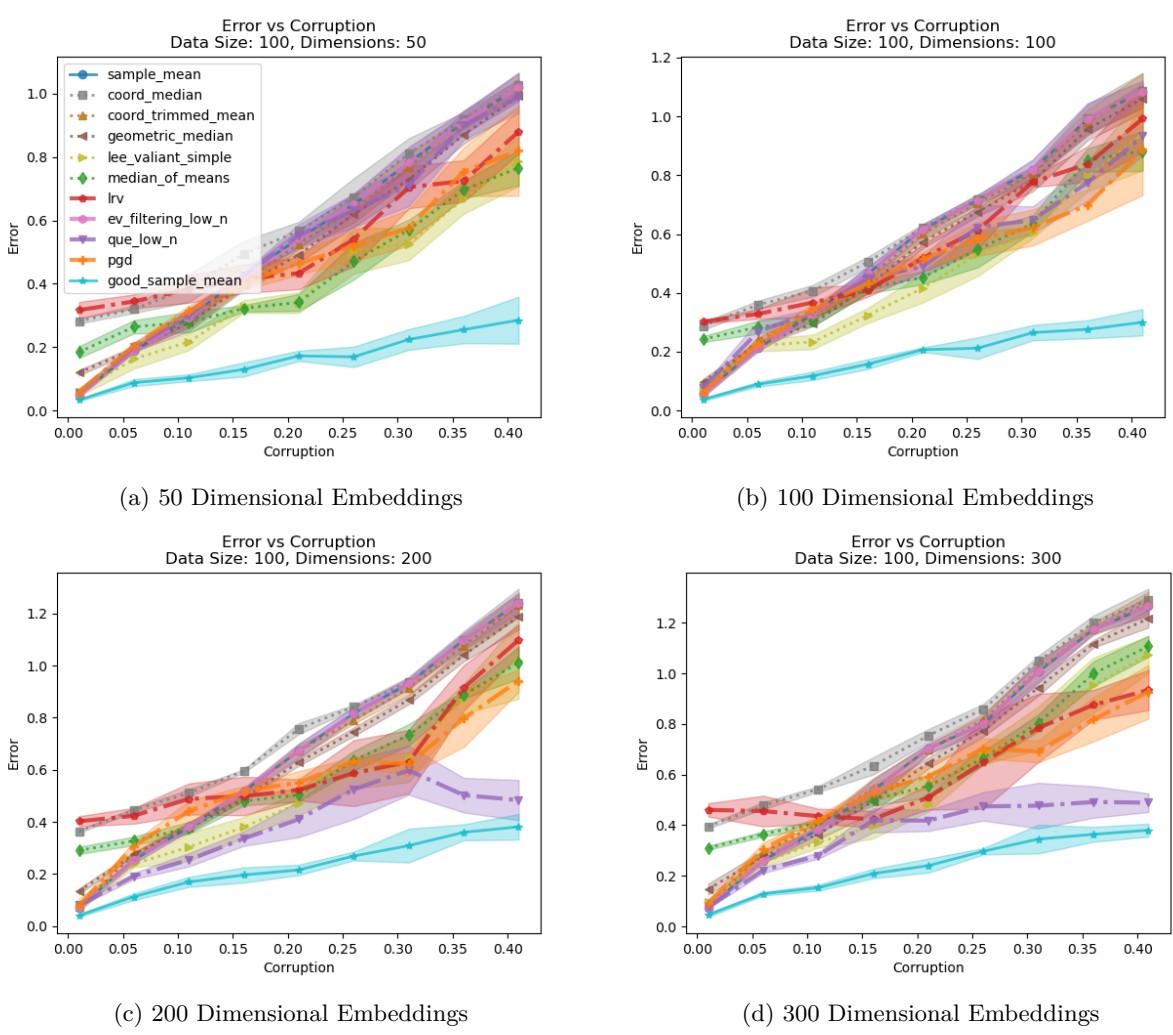

(a) 50 Dimensional Embeddings

(b) 100 Dimensional Embeddings

(c) 200 Dimensional Embeddings

(d) 300 Dimensional Embeddings

Figure 68: Error on "Unpleasant" Embeddings Corrupted with "Pleasant" Embeddings

## A.9 LLM Experiment Ablations

**Eigenvalue Pruning Method Comparison** We compare the performance of different pruning sub-routines for ev_filtering_low_n over a selection of LLM experiments: LOOCV and Corruption Error over MiniLM and BERT embeddings. We evaluate Gaussian pruning, used throughout this paper, along with randomized pruning and fixed pruning, described in Appendix A.5. We retain the same conditions as in the original experiments, first scaling data utilizing the sample trace. We also include sample_mean and QUE_low_n in our plots for the sake of comparison, noting that QUE_low_n and ev_filtering_low_n with fixed pruning only differ in their method of scoring outliers. These results are shown in Figure 69. We notice that both randomized and fixed pruning methods do indeed perform better than the Gaussian pruning method. In particular, fixed pruning has the best LOOCV error over MiniLM and matches the error of sample_mean over BERT, whereas Gaussian pruning fails dramatically. However, this performance does not translate into the corrupted case, where all three pruning routines lead to significant error compared to even sample_mean, except with large $\eta$ where it sample_mean's error approaches that of these methods. Additionally, as discussed in Appendix A.5, ev_filtering_low_n with fixed pruning is not robust to noise distributions that require several runs of the algorithm to prune i.e. cases where noise lays in multiple orthogonal clusters. Notably, QUE_low_n outperforms all variations of ev_filtering_low_n in the corrupted data case, reinforcing the observation that the outlier detection method of QUE_low_n is more robust to distributional assumptions than that of ev_filtering_low_n.

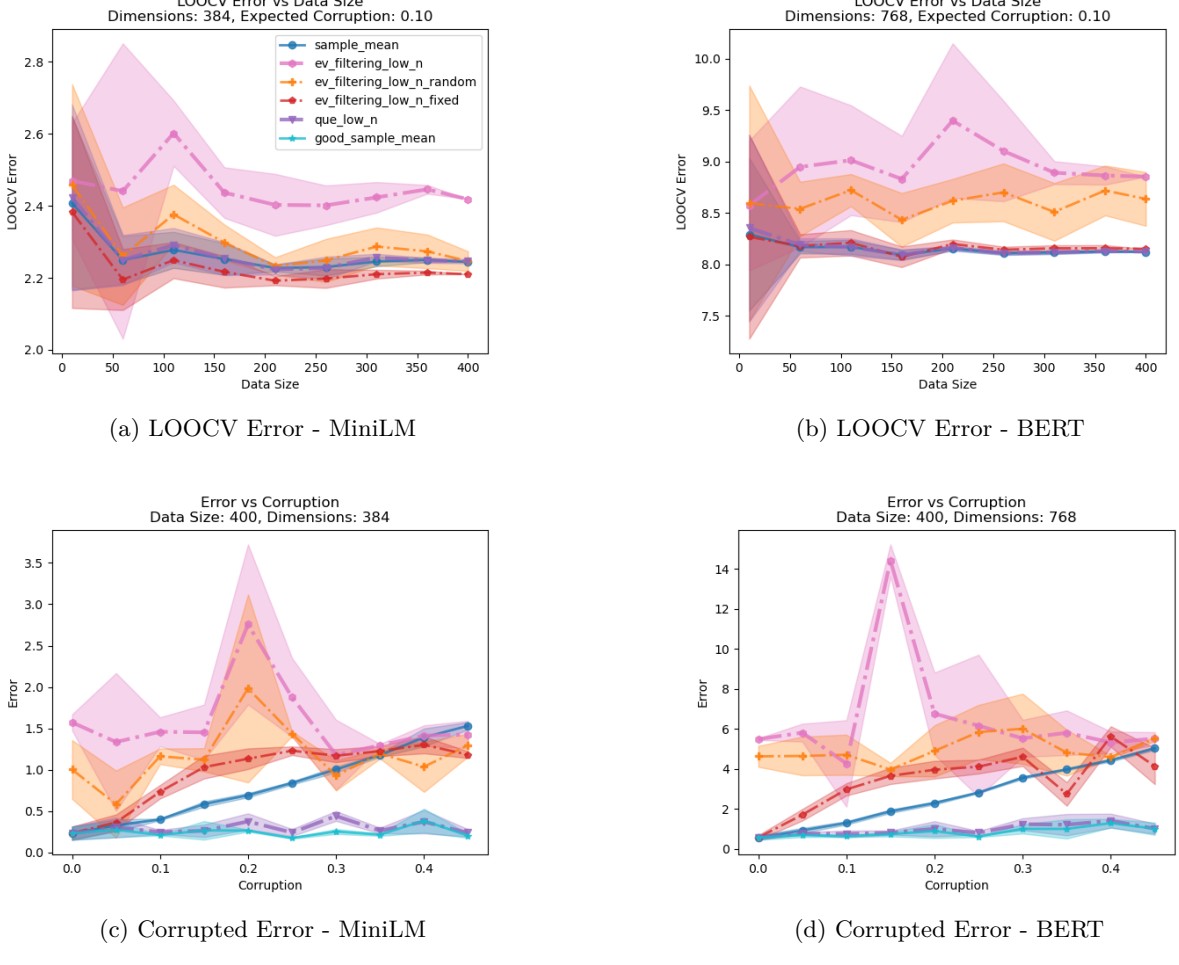

(a) LOOCV Error - MiniLM

(b) LOOCV Error - BERT

(c) Corrupted Error - MiniLM

(d) Corrupted Error - BERT

Figure 69: Eigenvalue Pruning - Pruning Method: LLM Comparison

**LRV Weighting Procedure**    Here we compare the two different weighting procedures for LRV described in Appendix A.5: Gaussian weighting, based on downweighting outliers, and general (non-Gaussian) weighting, based on completely pruning outliers. We evaluate these two methods over the same subselection of LLM experiments: LOOCV and Corruption Error over MiniLM and BERT embeddings. These results are shown in Figure 70. We notice that general weighting outperforms Gaussian weighting in LOOCV error, with this difference being especially noticeable across BERT embeddings. However, this performance increase is not seen in either corrupted case, where Gaussian weighting notably outperforms general weighting, except with small $\eta$. This suggests that, at least under low data size, LRV is not robust to general distributions, even using a general outlier weighting procedure.

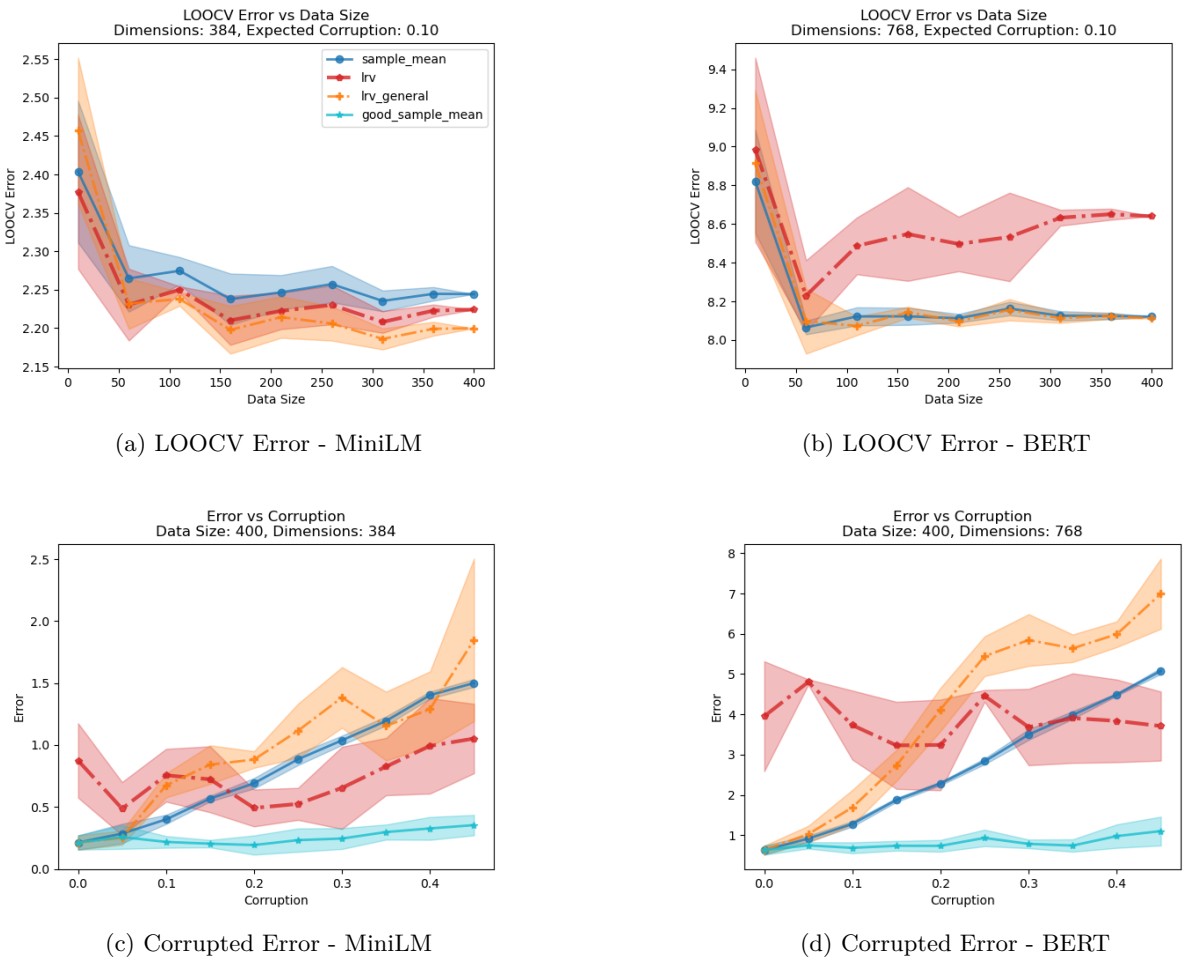

(a) LOOCV Error - MiniLM

(b) LOOCV Error - BERT

(c) Corrupted Error - MiniLM

(d) Corrupted Error - BERT

Figure 70: LRV - Gaussian Vs General Weighting: LLM Comparison

**Additional Experiments** We recreate the experiments in Section 5 over two different settings. First, we examine LOOCV Error over embeddings of the word field that correspond to the "field of study" definition rather than to the "field of land" definition. These results are shown in Figure 71. Second, we examine corrupted embeddings $X \sim (1 - \eta)P + \eta Q$, where inlier data, $P$, consists of embeddings of the word "field" corresponding to the "field of study" definition and outlier data, $Q$, consists of embeddings of the word "field" corresponding to the "field of land" definition; inverting the inlier and outlier data originally examined. These results are shown in Figure 72. While the LOOCV error plots are not identical to the original experiment, corresponding to the expected differences in structure between the distributions of $P$ and $Q$, we find the same overall trends across the 4 plots. We additionally observe the same overall trends for corrupted data compared to the original experiment. However, lee_valiant_simple, which was consistently the best algorithm alongside QUE_low_n for corrupted data originally, breaks down for MiniLM here; always performing notably worse than good_sample_mean. Supported by the general poor performance of lee_valiant_simple over synthetic data experiments, this reinforces the unpredictable sensitivity of lee_valiant_simple to distributional assumptions. QUE_low_n does not see any such degradation, performing near optimally across all cases, as it does in the original LLM experiment.

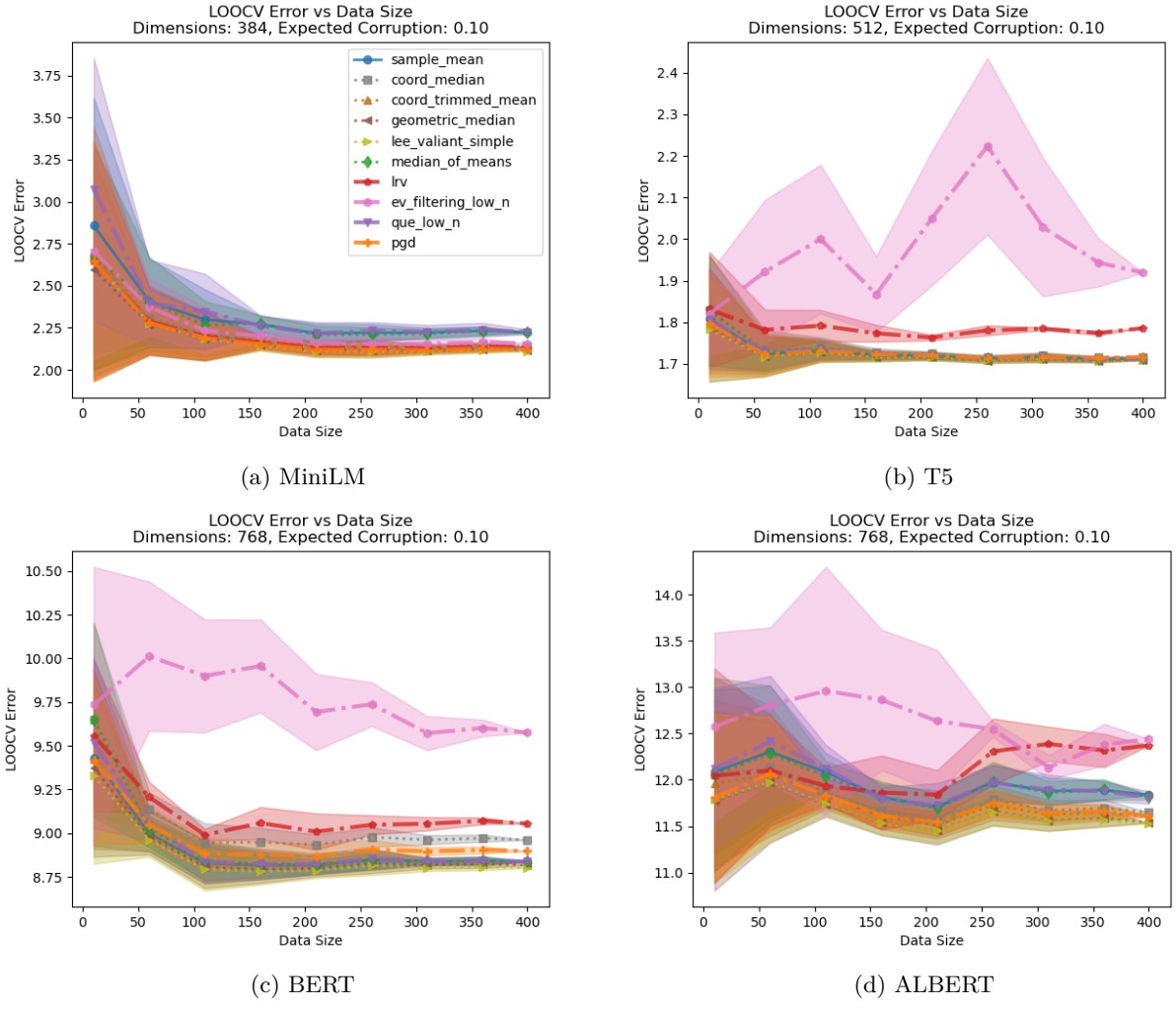

Figure 71: LOOCV Error on "Field Of Study" Embeddings

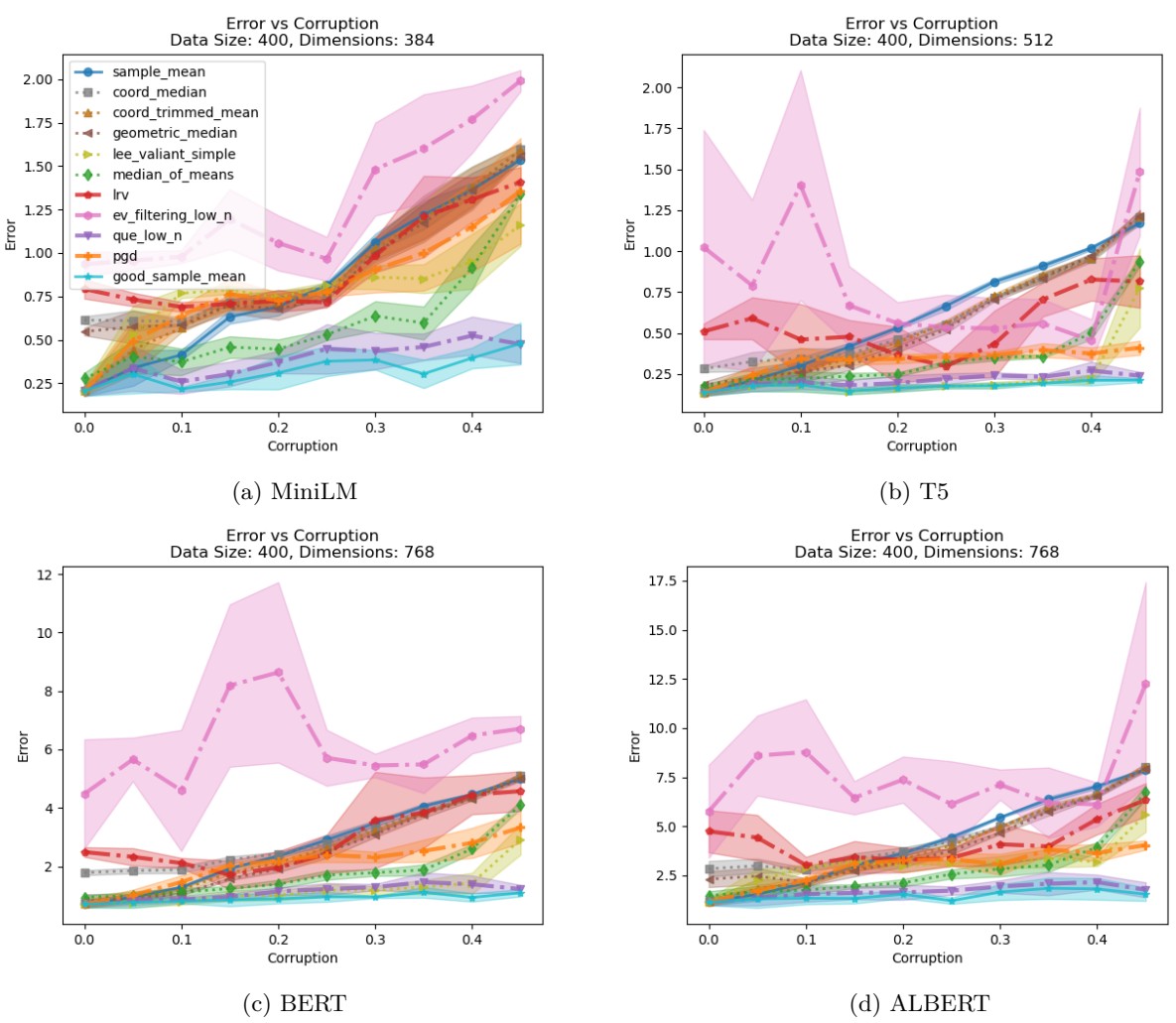

(a) MiniLM

(b) T5

(c) BERT

(d) ALBERT

Figure 72: Error on "Field of Study" Embeddings Corrupted with "Field of Land" Embeddings

### A.10  Dataset Generation

We generate a dataset of 400 sentences for each definition of the word *field* using ChatGPT-4o, accessed in June 2024. Attention based embeddings for the word *field* are extracted from these sentences for use in our LLM experiments. We used the following two prompts to obtain the sentences:

**Field of Study**

> *I am running an experiment where I examine embeddings of the word "field" with two different contexts. Please generate 400 unique sentences using the word "field" in context with the following definition: "a particular branch of study or sphere of activity or interest." Please return these sentences in the format of a JSON file.*

**Field of Land**

> *I am running an experiment where I examine embeddings of the word "field" with two different contexts. Please generate 400 unique sentences using the word "field" in context with the following definition: "an area of open land, especially one planted with crops or pasture, typically bounded by hedges or fences." Please return these sentences in the format of a JSON file.*

**Additional Prompts**  ChatGPT-4o did not produce the full 400 sentences in one go. To address this, we used the following additional prompts until we had generated the required number of sentences, and then manually combined the generated outputs. The prompt for "field of study" sentences is slightly different, as we originally observed that ChatGPT-4o would reuse the same field of study across numerous sentences.

For *Field of Study*:

> *Please generate 100 more sentences. Do not repeat similar sentences or use "field" to refer to the same field of study multiple times.*

For *Field of Land*:

> *Please generate 100 more sentences.*

**Tables Of Generated Sentences**  We include the following tables of generated sentences.

**Field Of Land Sentences**:

| Index | Sentence |
|---|---|
| 1 | The scarecrow stood tall in the middle of the field. |
| 2 | The deer were spotted grazing in the field at dawn. |
| 3 | The field was an ideal spot for stargazing. |
| 4 | He loved to watch the sunset over the field. |
| 5 | The field stretched out as far as the eye could see. |
| 6 | Sunflowers swayed in the field under the clear blue sky. |
| 7 | The field stretched out to the edge of the forest. |
| 8 | The field was alive with the sound of chirping crickets. |
| 9 | They played hide and seek in the field, darting among the tall grasses. |
| 10 | He built a small shed at the edge of the field for storage. |
| 11 | The hot air balloon landed gently in the field. |
| 12 | She enjoyed picnicking in the field of wildflowers near her home. |

*Continued on next page*

*Continued from previous page*

| Index | Sentence |
|---|---|
| 13 | She found an old, weathered barn at the edge of the field. |
| 14 | The field was fenced off to keep out wild animals. |
| 15 | They walked hand in hand through the field of wildflowers, lost in conversation. |
| 16 | He loved the quiet solitude of the open field. |
| 17 | The field was a perfect spot for birdwatching. |
| 18 | We had a picnic in the wide, open field. |
| 19 | The open field was covered in morning dew. |
| 20 | We watched the meteor shower from the field. |
| 21 | The open field was perfect for stargazing. |
| 22 | The field of strawberries was a patchwork of red and green. |
| 23 | The field was filled with the scent of blooming flowers. |
| 24 | Hikers followed the trail through the field of wild grasses, enjoying the solitude. |
| 25 | They played ultimate frisbee in the field. |
| 26 | The field of herbs was fragrant, each plant releasing its unique scent. |
| 27 | She found a hidden path that led to the field. |
| 28 | The field was a sea of green during the spring. |
| 29 | She found a hidden path that led to the field. |
| 30 | Wildflowers grew abundantly in the field. |
| 31 | A lone tree stood in the middle of the field, providing shade. |
| 32 | The field was a riot of color in the fall. |
| 33 | They harvested wheat from the vast field. |
| 34 | Deer grazed in the field at dusk, their silhouettes blending with the shadows. |
| 35 | The field was a vibrant green after the rain. |
| 36 | The field was a riot of color during the summer. |
| 37 | The field was a sea of gold during the harvest. |
| 38 | She enjoyed painting the landscape of the field. |
| 39 | He loved the feeling of the grass under his feet in the field. |
| 40 | We spotted deer grazing in the distant field. |
| 41 | The field was surrounded by rolling hills. |
| 42 | We walked through the field at sunrise. |
| 43 | The field was blanketed in snow during the winter. |
| 44 | The field was a playground for the neighborhood children. |
| 45 | The field was a sea of purple lavender in full bloom. |
| 46 | The field was blanketed in snow during the winter. |
| 47 | Cows grazed peacefully in the field enclosed by wooden fences. |
| 48 | The field, bordered by ancient oak trees, was a serene spot for a picnic. |
| 49 | She enjoyed walking through the field, picking flowers. |
| 50 | The field was blanketed with snow in winter. |
| 51 | He loved the peace and quiet of the open field. |
| 52 | The field was a patchwork of different crops. |
| 53 | The field was a sea of gold during the wheat harvest. |
| 54 | A scarecrow stood watch over the field. |
| 55 | He spent his afternoons walking through the field, lost in thought. |
| 56 | The field was divided into neat rows for planting. |
| 57 | The field was a perfect spot for a family picnic. |
| 58 | Birds chirped happily in the field, searching for insects among the plants. |
| 59 | The field was dotted with patches of wild grass. |
| 60 | She painted a landscape of the field in her art class. |
| 61 | The field was dotted with hay bales after a long day of harvesting. |

*Continued from previous page*

| Index | Sentence |
|---|---|
| 62 | He built a small fire pit in the middle of the field. |
| 63 | He loved the smell of fresh-cut grass in the field. |
| 64 | She found a quiet spot in the field to read her book. |
| 65 | The field was a burst of color in the autumn. |
| 66 | The field of rye swayed in the breeze, creating waves of green. |
| 67 | The field was alive with the sound of crickets. |
| 68 | The field was a popular spot for local festivals. |
| 69 | He loved the smell of fresh-cut grass in the field. |
| 70 | The field was a favorite spot for local photographers. |
| 71 | He loved to run through the field with his friends. |
| 72 | The field was alive with the sound of crickets. |
| 73 | We could see the farmhouse from across the field. |
| 74 | He loved the quiet solitude of the open field. |
| 75 | A scarecrow stood at the center of the field, arms outstretched. |
| 76 | We spotted a fox darting through the field. |
| 77 | The children flew paper airplanes in the field. |
| 78 | They set up a makeshift baseball diamond in the field. |
| 79 | A herd of sheep grazed peacefully in the field. |
| 80 | They played a game of tag in the spacious field. |
| 81 | He loved to explore the field with his dog. |
| 82 | A gentle breeze rustled the leaves of the crops in the field. |
| 83 | The field was covered in a blanket of fresh snow. |
| 84 | The horses galloped freely across the open field. |
| 85 | The dogs ran freely in the wide field. |
| 86 | The field of blueberries was a favorite spot for summer picking. |
| 87 | The field was a peaceful place to reflect and relax. |
| 88 | The scarecrow stood guard in the field, its tattered clothes fluttering in the breeze. |
| 89 | The field was a place of peace and serenity. |
| 90 | A gentle fog settled over the field in the morning. |
| 91 | The field was home to several species of birds. |
| 92 | The field was ideal for an outdoor concert. |
| 93 | She enjoyed walking through the field, listening to the birds sing. |
| 94 | The field was alive with the sound of crickets chirping at night. |
| 95 | The field was alive with the sound of crickets. |
| 96 | The field was a perfect spot for a family picnic. |
| 97 | The field of corn rustled in the wind, creating a soothing sound. |
| 98 | The field was a quiet refuge from the busy city. |
| 99 | He built a small bench at the edge of the field. |
| 100 | They played frisbee in the open field. |
| 101 | He loved the smell of fresh-cut grass in the field. |
| 102 | She enjoyed picnicking in the field with her friends. |
| 103 | The farmer worked tirelessly in the field to ensure a good harvest. |
| 104 | A small stream ran along the edge of the field, providing water for the livestock. |
| 105 | She enjoyed walking through the field, picking flowers. |
| 106 | The community garden was set up in the field. |
| 107 | A tractor moved slowly across the field, plowing the earth for new seeds. |
| 108 | They had an Easter egg hunt in the field. |
| 109 | We could see the farmhouse from across the field. |

*Continued from previous page*

| Index | Sentence |
|---|---|
| 110 | A gentle breeze blew across the field. |
| 111 | The field stretched out to the horizon, seemingly endless. |
| 112 | The field buzzed with the sound of bees collecting nectar. |
| 113 | The field was a riot of color in the fall. |
| 114 | The field was home to a variety of wildlife. |
| 115 | Birds nested in the hedges surrounding the field, singing melodious tunes. |
| 116 | The field was dotted with hay bales, ready for storage. |
| 117 | The field was a sea of gold during the wheat harvest. |
| 118 | The field was lush and green after the rain. |
| 119 | She loved to dance barefoot in the field. |
| 120 | The kids enjoyed a treasure hunt in the field. |
| 121 | The field of cherry blossoms was a sight to behold, petals drifting in the wind. |
| 122 | The farmer walked across the field, inspecting the growing wheat. |
| 123 | After the rain, the field was dotted with puddles reflecting the clouds. |
| 124 | Butterflies flitted about in the field, adding to its charm. |
| 125 | She could see the field from her kitchen window. |
| 126 | We found a quiet spot in the field to relax. |
| 127 | They set up a campfire in the field. |
| 128 | The farmer plowed the field in preparation for planting. |
| 129 | She enjoyed walking through the field, picking flowers. |
| 130 | The field's soil was rich and fertile, ideal for planting. |
| 131 | The field was a vibrant green after the rain. |
| 132 | The field was a peaceful place to escape to. |
| 133 | The field of sunflowers attracted bees with its abundance of pollen. |
| 134 | The field was blanketed in snow during the winter. |
| 135 | A small brook ran alongside the field, providing irrigation. |
| 136 | He watched the sunrise over the field from his porch. |
| 137 | She found a quiet corner of the field to meditate. |
| 138 | She found a hidden path that led to the field. |
| 139 | She picked wild strawberries in the field, their sweetness bursting in her mouth. |
| 140 | The large field was perfect for flying kites. |
| 141 | The field of cotton was ready for picking, the fluffy bolls bursting open. |
| 142 | The field of rapeseed glowed bright yellow against the blue sky. |
| 143 | He loved to run through the field with his friends. |
| 144 | The field was a canvas of colors in the spring. |
| 145 | The field was surrounded by a wooden fence. |
| 146 | Farmers plowed the field, preparing it for the next planting season. |
| 147 | He loved to watch the sunset over the field. |
| 148 | She could hear the distant sound of a tractor working in the field. |
| 149 | Children flew kites in the field, the colorful tails dancing in the wind. |
| 150 | The field was a place of beauty and tranquility. |
| 151 | They had a barbecue in the field last weekend. |
| 152 | The field of lettuce was irrigated regularly, ensuring crisp leaves. |
| 153 | She found a hidden trail that led to the field. |
| 154 | She enjoyed walking through the field, listening to the birds sing. |
| 155 | A narrow path cut through the field, leading to the old barn. |
| 156 | She found a perfect spot in the field for her garden. |
| 157 | The field was a sea of green grass in the spring. |
| 158 | Farm workers toiled from dawn to dusk, tending to the vast field. |

*Continued on next page*

*Continued from previous page*

| Index | Sentence |
|---|---|
| 159 | The field of oats rustled softly as the wind passed through. |
| 160 | She loved to dance barefoot in the field. |
| 161 | A gentle mist rose from the field in the early morning. |
| 162 | The field of soybeans stretched for miles, a sea of green leaves. |
| 163 | The field was a place of peace and serenity. |
| 164 | The farmer surveyed the field, planning his next move. |
| 165 | The field was a sea of gold during the wheat harvest. |
| 166 | The field was a patchwork of different crops, each thriving in the rich soil. |
| 167 | She spent her afternoons wandering through the field. |
| 168 | The wheat field swayed gently in the wind. |
| 169 | Sunflowers bloomed vibrantly in the field, their faces turning towards the sun. |
| 170 | The field was a perfect place for a family gathering. |
| 171 | The field was alive with the sound of crickets. |
| 172 | The field was lush with green crops swaying in the breeze. |
| 173 | In the distance, a barn overlooked the sprawling field. |
| 174 | The field was a sea of green during the growing season. |
| 175 | The field glistened with morning dew. |
| 176 | They picked strawberries in the field. |
| 177 | The field was a playground for the neighborhood children. |
| 178 | A herd of cows roamed freely in the field. |
| 179 | A flock of birds flew over the field. |
| 180 | The field was blanketed in wildflowers during the spring. |
| 181 | The field was full of life, even in the winter. |
| 182 | The field was a favorite spot for watching the stars at night. |
| 183 | Children played a game of soccer in the field behind the school. |
| 184 | The field was a perfect spot for birdwatching. |
| 185 | The field was a peaceful place to escape to. |
| 186 | The kids enjoyed running freely in the open field. |
| 187 | He set up a telescope in the field to watch the stars. |
| 188 | The children lay down in the field to watch the clouds. |
| 189 | The field was dotted with blooming flowers. |
| 190 | He found a perfect spot in the field for his garden. |
| 191 | The cows roamed freely in the open field. |
| 192 | The field of corn was ready for harvesting. |
| 193 | The field of daisies swayed gently in the wind, a sea of white petals. |
| 194 | The field provided a perfect backdrop for photos. |
| 195 | The field was a sea of gold during the harvest. |
| 196 | They held a yoga class in the field at dawn. |
| 197 | The field lay fallow, resting before the next planting season. |
| 198 | The sun set behind the distant field. |
| 199 | He could see the field from his bedroom window. |
| 200 | The field was a vibrant green after the rain. |
| 201 | She ran through the field of tall grass, her laughter ringing out. |
| 202 | They set up a tent in the field for the event. |
| 203 | The field was a favorite spot for local photographers. |
| 204 | She enjoyed collecting wildflowers from the field. |
| 205 | They flew kites in the large, empty field. |
| 206 | The field was a vibrant green after the rain. |
| 207 | The festival was held in the open field every summer. |

*Continued on next page*

*Continued from previous page*

| Index | Sentence |
|-------|----------|
| 208 | The field was a haven for wildflowers. |
| 209 | The field of radishes was ready for picking, their bright red roots peeking out. |
| 210 | The field was lush with green crops swaying in the breeze. |
| 211 | They saw a fox darting across the field. |
| 212 | She found a quiet corner of the field to meditate. |
| 213 | Rows of corn stretched across the field, reaching up towards the sky. |
| 214 | The field was a playground for the neighborhood children. |
| 215 | She took a stroll through the field, enjoying the fresh air. |
| 216 | He loved the smell of fresh-cut hay in the field. |
| 217 | The field was a patchwork of colors during the flower festival. |
| 218 | She spent her afternoons wandering through the field. |
| 219 | The field was covered in a blanket of wildflowers. |
| 220 | The picnic was set up in the field by the lake. |
| 221 | The field was a sea of green during the growing season. |
| 222 | A beautiful field of sunflowers stretched as far as the eye could see. |
| 223 | The scarecrow stood watch over the field, deterring hungry birds. |
| 224 | The horses galloped across the field, kicking up dust behind them. |
| 225 | The horses grazed peacefully in the field. |
| 226 | He loved the peace and quiet of the open field. |
| 227 | The field of tulips was a riot of color, reds, yellows, and pinks blending together. |
| 228 | The field was buzzing with activity during the harvest season. |
| 229 | He loved the quiet solitude of the open field. |
| 230 | She painted a beautiful landscape of the field. |
| 231 | He set up a telescope in the field to watch the stars. |
| 232 | They wandered through the field of pumpkins, searching for the perfect one. |
| 233 | In the winter, the field was covered in a thick layer of snow. |
| 234 | Cows grazed peacefully in the field enclosed by wooden fences. |
| 235 | The farmers planted potatoes in the field closest to the farmhouse. |
| 236 | They set up a tent in the field, ready for a weekend of camping. |
| 237 | The field was a haven for wildflowers. |
| 238 | At sunset, the field glowed with a golden hue, creating a picturesque scene. |
| 239 | The sheep roamed freely in the field, nibbling on fresh grass. |
| 240 | She enjoyed painting the landscape of the field. |
| 241 | He loved to explore the field with his dog. |
| 242 | The field was a patchwork of different crops. |
| 243 | They saw a rainbow stretching over the field. |
| 244 | He found solace in the quiet field, away from the hustle and bustle. |
| 245 | The field was a place of peace and serenity. |
| 246 | The field was an expanse of green grass. |
| 247 | The field was a haven for wildlife, including rabbits and deer. |
| 248 | He built a small bench at the edge of the field. |
| 249 | A gentle breeze rustled the leaves in the field. |
| 250 | She collected wildflowers from the field to make a bouquet. |
| 251 | The field, surrounded by rolling hills, was a picture of serenity. |
| 252 | She found an old, weathered barn at the edge of the field. |
| 253 | The festival was held in the large, open field. |
| 254 | She spent her afternoons wandering through the field. |
| 255 | The field was a playground for the neighborhood kids. |
| 256 | The field was a vibrant green after the rain. |

*Continued from previous page*

| Index | Sentence |
|-------|----------|
| 257 | The field of cabbages was neatly arranged in rows, their heads forming a patchwork. |
| 258 | She found a quiet spot in the field to read her book. |
| 259 | Farmers tended to the field of pumpkins, ensuring each one grew plump and round. |
| 260 | The field was a quiet refuge from the busy city. |
| 261 | They set up camp in the field, under a canopy of stars. |
| 262 | The field was a favorite spot for local artists. |
| 263 | The field was plowed into neat, straight rows. |
| 264 | The kids played catch in the large field. |
| 265 | The field was a favorite spot for local photographers. |
| 266 | The farmer rotated his crops to keep the field fertile. |
| 267 | We had a family reunion in the field. |
| 268 | He loved the smell of fresh-cut hay in the field. |
| 269 | The field of onions was harvested in the fall, bulbs dug up and stored for winter. |
| 270 | The field was blanketed in fog early in the morning. |
| 271 | The field provided ample space for the annual county fair. |
| 272 | She enjoyed collecting wildflowers from the field. |
| 273 | She loved to dance barefoot in the field. |
| 274 | The field was a riot of color in the summer. |
| 275 | The field was a perfect spot for birdwatching. |
| 276 | She ran across the field, her laughter echoing in the open space. |
| 277 | Butterflies fluttered over the wildflowers that dotted the field. |
| 278 | The field stretched out to the horizon. |
| 279 | They practiced their golf swings in the open field. |
| 280 | The field was perfect for a game of cricket. |
| 281 | In the summer, the field was a sea of golden barley ready for harvest. |
| 282 | The field was home to a family of rabbits. |
| 283 | The field was alive with the hum of insects. |
| 284 | The field was a vibrant green after the rain. |
| 285 | The field was dotted with patches of wild grass. |
| 286 | We enjoyed a picnic lunch in the sunny field. |
| 287 | We lay on the blanket in the middle of the field. |
| 288 | She found a quiet spot in the field to read her book. |
| 289 | Children played soccer in the open field near the village. |
| 290 | The field was covered in a blanket of snow during the winter months. |
| 291 | The field was a vibrant green after the spring rain. |
| 292 | He loved to explore the field with his dog. |
| 293 | Wildflowers dotted the field, adding splashes of color to the landscape. |
| 294 | They walked through the field of lavender, inhaling its sweet fragrance. |
| 295 | We had a bonfire in the middle of the field. |
| 296 | He found a quiet spot in the field to read his book. |
| 297 | The field was a sea of green during the growing season. |
| 298 | The field of barley was ready for harvest, the grains turning golden. |
| 299 | The field was surrounded by a white picket fence. |
| 300 | He built a small shed at the edge of the field. |
| 301 | The field of sugar cane stretched to the horizon, its stalks swaying gently. |
| 302 | The field was a burst of color in the autumn. |
| 303 | She enjoyed collecting wildflowers from the field. |

*Continued from previous page*

| Index | Sentence |
|-------|----------|
| 304 | He loved to watch the sunset over the field. |
| 305 | A beautiful field of sunflowers stretched as far as the eye could see. |
| 306 | She found an old, rusty plow at the edge of the field. |
| 307 | We watched the sunset from the edge of the field. |
| 308 | The dogs loved running around in the open field. |
| 309 | The field of lavender was a haven for bees, buzzing busily among the flowers. |
| 310 | The field was a riot of color in the fall. |
| 311 | Farmers worked diligently in the corn field. |
| 312 | The field was dotted with patches of wild grass. |
| 313 | Sheep roamed freely in the field, their woolly coats glistening in the sun. |
| 314 | The children played soccer in the field. |
| 315 | The field was a favorite spot for local artists. |
| 316 | He built a small bench at the edge of the field. |
| 317 | The field was a tapestry of colors in the spring, with various flowers in bloom. |
| 318 | We could see rabbits hopping in the field. |
| 319 | The field was a sea of gold during the harvest. |
| 320 | She sat under the oak tree in the field, enjoying the shade. |
| 321 | Cattle grazed peacefully in the field, surrounded by rolling hills. |
| 322 | The field looked magical under the light of the full moon. |
| 323 | The field was dotted with bales of hay. |
| 324 | The field was a riot of color during the summer. |
| 325 | He set up a picnic in the middle of the field. |
| 326 | The old oak tree stood alone in the field. |
| 327 | A scarecrow stood tall in the middle of the field, warding off birds. |
| 328 | The field was a favorite spot for local photographers. |
| 329 | She spent hours wandering through the field, collecting herbs. |
| 330 | She enjoyed picnicking in the field with her family. |
| 331 | The field was home to a variety of wildlife. |
| 332 | She lay down in the field, gazing up at the clear blue sky. |
| 333 | She found a hidden trail that led to the field. |
| 334 | The field was full of life, even in the winter. |
| 335 | The field was surrounded by a dense forest. |
| 336 | In the fall, the field turned a golden hue as the crops matured. |
| 337 | A light breeze swept through the open field. |
| 338 | He found a hidden trail that led to the field. |
| 339 | The path led us through a vast field. |
| 340 | He built a small shed at the edge of the field. |
| 341 | He found a perfect spot in the field for his garden. |
| 342 | The field was a haven for wildflowers. |
| 343 | The field was a riot of color during the summer. |
| 344 | The scent of freshly cut grass filled the air as the field was mowed. |
| 345 | They walked through the field of wheat, the stalks brushing against their legs. |
| 346 | Children ran around playing in the field all day. |
| 347 | The field was a quiet refuge from the busy city. |
| 348 | The field was a peaceful place to escape to. |
| 349 | He could see the field from his bedroom window. |
| 350 | The field was a favorite spot for local artists. |
| 351 | The field was full of life, even in the winter. |
| 352 | A fence made of wooden posts and wire encircled the field. |

*Continued on next page*

*Continued from previous page*

| Index | Sentence |
|---|---|
| 353 | The field was a burst of color in the autumn. |
| 354 | He loved to run through the field with his friends. |
| 355 | A tractor moved slowly across the field, tilling the soil. |
| 356 | She lay in the field, watching the clouds drift by. |
| 357 | The field was a sea of green during the spring. |
| 358 | He loved the peace and quiet of the open field. |
| 359 | The field was a patchwork of different crops. |
| 360 | The field was a favorite spot for flying drones. |
| 361 | Farmers harvested hay from the field, stacking it neatly in bales. |
| 362 | She found a quiet corner of the field to meditate. |
| 363 | The field was a sea of green during the spring. |
| 364 | Children played soccer in the field behind the school, their laughter echoing. |
| 365 | The tractor moved slowly across the plowed field. |
| 366 | The field was dotted with patches of clover. |
| 367 | He built a small shed at the edge of the field. |
| 368 | The field was a place of beauty and tranquility. |
| 369 | He set up a telescope in the field to watch the stars. |
| 370 | The field was used for growing sunflowers. |
| 371 | She enjoyed walking through the field, listening to the birds sing. |
| 372 | The field of wheat stretched across the horizon, golden under the afternoon sun. |
| 373 | They planted a variety of vegetables in the field. |
| 374 | She loved the scent of fresh earth in the field after it rained. |
| 375 | The field was a peaceful place to reflect and relax. |
| 376 | They picked wildflowers from the edge of the field. |
| 377 | A lone tree stood in the middle of the field. |
| 378 | He could see the field from his bedroom window. |
| 379 | The field of hemp grew tall and strong, its fibers used for various products. |
| 380 | He loved the smell of fresh-cut hay in the field. |
| 381 | The field was buzzing with bees collecting nectar. |
| 382 | A rainbow arched over the field after the rain. |
| 383 | They planted rows of vegetables in the fertile field. |
| 384 | The field was a peaceful place to reflect and relax. |
| 385 | The field of vineyards produced grapes for fine wines, rows of vines neatly trellised. |
| 386 | She found an old, weathered barn at the edge of the field. |
| 387 | They picnicked in the field of clover, enjoying sandwiches and lemonade. |
| 388 | She enjoyed painting the landscape of the field. |
| 389 | We took a walk through the field at sunset. |
| 390 | They built a bonfire in the field, its flames lighting up the night sky. |
| 391 | The farmer's dog ran joyfully through the field. |
| 392 | The field was the perfect spot for a family gathering. |
| 393 | She enjoyed picnicking in the field with her family. |
| 394 | The field was covered in a thick layer of frost. |
| 395 | The field of potatoes was ready for digging, the earth yielding its treasures. |
| 396 | He could see deer grazing in the field from his window. |
| 397 | The field was a place of beauty and tranquility. |
| 398 | The field was a perfect spot for a family picnic. |
| 399 | Rain nourished the field, ensuring a bountiful crop for the season. |

*Continued from previous page*

| Index | Sentence |
|---|---|
| 400 | The field was a haven for birdwatchers. |

Table 2: Field Of Land Sentences

**Field Of Study Sentences**:

| Index | Sentence |
|---|---|
| 1 | The field of bioethics addresses the ethical issues in biology and medicine. |
| 2 | He is an innovator in the field of software development. |
| 3 | The conference attracted top professionals specialized in an interesting field of study. |
| 4 | The field of cultural studies examines how culture shapes identity. |
| 5 | The field of immunology studies the immune system. |
| 6 | The field of anthropology studies human cultures and societies. |
| 7 | His expertise in the field of structural engineering is invaluable. |
| 8 | My favorite baseball movie is field of dreams. |
| 9 | The field of socio-cultural anthropology examines human societies and their customs. |
| 10 | The field of cultural anthropology studies human societies. |
| 11 | The field of operations research uses mathematical methods to make decisions. |
| 12 | The field of evolutionary psychology explores the evolutionary origins of human behavior. |
| 13 | He is an authority in the field of health policy and management. |
| 14 | He is a leading expert in the field of forensic anthropology. |
| 15 | Her studies in the field of dance theory are intriguing. |
| 16 | He has made strides in the field of molecular genetics. |
| 17 | His work in the field of plasma physics is highly regarded. |
| 18 | The field of computational sociology uses computational methods to study social phenomena. |
| 19 | His studies in the field of marine biology are fascinating. |
| 20 | The field of telecommunications is rapidly advancing. |
| 21 | The field of supply chain management is vital for global commerce. |
| 22 | He decided to pursue a career in the field of computer science. |
| 23 | The field of biotechnology holds great promise for the future. |
| 24 | She has a background in the field of political science. |
| 25 | He is a leading researcher in the field of evolutionary genetics. |
| 26 | The field of digital forensics investigates cybercrimes. |
| 27 | The field of transpersonal psychology explores spiritual and transcendent aspects of the human experience. |
| 28 | He is a pioneer in the field of machine learning. |
| 29 | Her research in the field of music cognition explores how the brain processes music. |
| 30 | The field of sociology looks at how societies function. |
| 31 | Her research in the field of computational chemistry is groundbreaking. |
| 32 | Her expertise in the field of infectious diseases informs public health policies. |
| 33 | Her expertise in the field of environmental sociology addresses human interactions with the environment. |
| 34 | The field of optics studies light and its interactions. |

*Continued on next page*

*Continued from previous page*

| Index | Sentence |
| --- | --- |
| 35 | The field of biogeography studies the distribution of species across geographical areas. |
| 36 | He received an award for his contributions to the field of engineering. |
| 37 | Her passion for the field of public health is evident. |
| 38 | The field of marketing explores consumer behavior and advertising strategies. |
| 39 | She is passionate about her work in the field of social work. |
| 40 | The field of actuarial science assesses financial risks. |
| 41 | The field of hydrology studies the distribution and movement of water on Earth. |
| 42 | She has received accolades for her work in the field of artificial intelligence. |
| 43 | The field of health informatics improves patient care through data. |
| 44 | The field of neuroinformatics combines neuroscience and data analysis. |
| 45 | She is highly respected in the field of architectural history. |
| 46 | He is a pioneer in the field of digital humanities. |
| 47 | The field of sports medicine focuses on athletes' health and performance. |
| 48 | The field of chemical engineering involves the creation of new materials. |
| 49 | He is a leading voice in the field of peace and conflict studies. |
| 50 | He is a prominent figure in the field of environmental sociology. |
| 51 | The field of sports medicine helps athletes recover from injuries. |
| 52 | The field of industrial design merges function with aesthetics. |
| 53 | The field of geriatric medicine focuses on elderly care. |
| 54 | The field of behavioral economics blends psychology and economics. |
| 55 | Her research in the field of climatology addresses global warming. |
| 56 | His research in the field of evolutionary biology is groundbreaking. |
| 57 | Her research in the field of computer graphics enhances visual simulation techniques. |
| 58 | The field of health informatics uses technology to improve healthcare delivery. |
| 59 | He has a deep interest in the field of computational neuroscience. |
| 60 | He is a renowned figure in the field of machine learning. |
| 61 | The field of medical anthropology explores the intersection of culture and health. |
| 62 | The field of acoustics studies sound and its properties. |
| 63 | Her research in the field of gerontology focuses on aging. |
| 64 | The field of molecular gastronomy explores the science behind cooking. |
| 65 | She is advancing knowledge in the field of developmental psychology. |
| 66 | The field of physical therapy helps people recover mobility. |
| 67 | The field of social epidemiology examines health disparities. |
| 68 | The field of artificial intelligence presents many ethical questions. |
| 69 | She is a trailblazer in the field of behavioral neuroscience. |
| 70 | Her expertise in the field of neuroimaging enhances brain research. |
| 71 | She is an innovator in the field of fashion design. |
| 72 | His work in the field of artificial intelligence has been widely recognized. |
| 73 | He has made significant strides in the field of computational neuroscience. |
| 74 | She is making waves in the field of renewable energy. |
| 75 | The field of computational neuroscience models neural systems and behavior. |
| 76 | His studies in the field of immunology are groundbreaking. |
| 77 | He is a leading scholar in the field of information science. |
| 78 | He is a leading figure in the field of aerospace engineering. |
| 79 | The field of digital sociology explores the impact of digital technologies on society. |

*Continued from previous page*

| Index | Sentence |
|-------|----------|
| 80 | Her expertise in the field of gerontology addresses aging. |
| 81 | The field of linguistics helps us understand language structure. |
| 82 | The field of library science organizes and manages information resources. |
| 83 | Her expertise in the field of data science is highly sought after. |
| 84 | The field of literary criticism involves analyzing and interpreting texts. |
| 85 | He is a pioneer in the field of disaster risk reduction. |
| 86 | She is an influential figure in the field of gender studies. |
| 87 | The field of cognitive anthropology studies cultural variations in cognition. |
| 88 | The field of agronomy deals with crop production and soil management. |
| 89 | He is a leading figure in the field of quantum information science. |
| 90 | He has published extensively in the field of theoretical physics. |
| 91 | The field of health informatics combines healthcare and IT. |
| 92 | Her research in the field of psychopharmacology examines the effects of drugs on behavior. |
| 93 | He has authored several books in the field of history. |
| 94 | The field of international law governs legal relations between states. |
| 95 | The field of ethology examines animal behavior in natural environments. |
| 96 | The field of social psychology studies how individuals influence each other. |
| 97 | The field of political economy studies the relationship between politics and economics. |
| 98 | The field of anthropology examines human societies and cultures. |
| 99 | Her expertise in the field of marine archaeology uncovers submerged history. |
| 100 | Her work in the field of human-computer interaction designs user-friendly interfaces. |
| 101 | He is well-known in the field of quantum physics. |
| 102 | He has dedicated his career to the field of atmospheric sciences. |
| 103 | She has a distinguished career in the field of comparative literature. |
| 104 | The field of cultural studies examines cultural phenomena. |
| 105 | Her work in the field of artificial intelligence is innovative. |
| 106 | He is a notable expert in the field of space exploration. |
| 107 | Her research in the field of cognitive anthropology explores cultural cognition. |
| 108 | Her work in the field of cognitive psychology studies mental processes. |
| 109 | He is a well-known expert in the field of civil engineering. |
| 110 | The field of behavioral economics explores how psychology impacts economic decisions. |
| 111 | The field of consumer psychology studies consumer behavior and decision-making. |
| 112 | The field of artificial intelligence is growing rapidly. |
| 113 | The field of aerospace science investigates flight and space. |
| 114 | She has published numerous papers in the field of environmental science. |
| 115 | She has a keen interest in the field of film studies. |
| 116 | His innovations in the field of electrical engineering are impressive. |
| 117 | Her research in the field of mobile computing enhances smartphone technology. |
| 118 | The field of computational economics applies computational methods to economic analysis. |
| 119 | The field of marine chemistry studies the chemical composition of oceans. |
| 120 | His work in the field of marine biology is groundbreaking. |
| 121 | Her innovations in the field of textile science are notable. |
| 122 | The field of consumer psychology explores why people buy things. |

*Continued on next page*

*Continued from previous page*

| Index | Sentence |
|---|---|
| 123 | The field of evolutionary psychology explores human behavior. |
| 124 | The field of computational biology uses data to understand biology. |
| 125 | He has extensive experience in the field of quantum computing. |
| 126 | The field of neuroscience delves into the workings of the brain. |
| 127 | The field of veterinary science focuses on animal health. |
| 128 | She has a strong foundation in the field of theoretical physics. |
| 129 | The field of music therapy uses music to improve mental health. |
| 130 | She has a profound impact on the field of forensic science. |
| 131 | His work in the field of linguistics has redefined language theories. |
| 132 | Her interest in the field of space exploration began in childhood. |
| 133 | The field of industrial design creates functional and aesthetic products. |
| 134 | His studies in the field of artificial intelligence are influential. |
| 135 | The field of educational psychology enhances teaching methods. |
| 136 | He is a notable figure in the field of mechanical engineering. |
| 137 | The field of geophysics examines the physical properties of the Earth. |
| 138 | Her work in the field of urban planning promotes sustainable urban development. |
| 139 | The field of medical imaging develops techniques to visualize internal organs. |
| 140 | She has been working in the field of bioengineering for over a decade. |
| 141 | He has a background in the field of developmental psychology. |
| 142 | The field of musicology delves into the study of music. |
| 143 | He is a renowned scholar in the field of classical studies. |
| 144 | The field of computational archaeology uses computer models to study archaeological data. |
| 145 | He is an authority in the field of cybersecurity. |
| 146 | Her contributions to the field of immunology have been invaluable. |
| 147 | The field of mathematical biology applies mathematical models to biological processes. |
| 148 | Her research in the field of neuroscience is highly respected. |
| 149 | The field of sports psychology helps athletes improve performance. |
| 150 | The field of environmental engineering seeks sustainable solutions. |
| 151 | The field of social neuroscience investigates the neural basis of social behavior. |
| 152 | The field of social geography studies the spatial distribution of social phenomena. |
| 153 | The field of landscape architecture focuses on designing outdoor spaces. |
| 154 | The field of psycholinguistics investigates the relationship between language and the mind. |
| 155 | He is a trailblazer in the field of genetic counseling. |
| 156 | The field of materials science investigates the properties of materials. |
| 157 | The field of computational linguistics develops algorithms for natural language processing. |
| 158 | He has a background in the field of political sociology. |
| 159 | The field of semiotics analyzes signs and symbols in communication. |
| 160 | The field of planetary science explores the formation and evolution of planets. |
| 161 | Her research in the field of robotics engineering advances automation technology. |
| 162 | He is a specialist in the field of aerospace engineering. |
| 163 | Her research in the field of cultural psychology investigates cultural influences on cognition. |

*Continued on next page*

*Continued from previous page*

| Index | Sentence |
|---|---|
| 164 | She has published numerous papers in the field of quantum mechanics. |
| 165 | He is an expert in the field of agribusiness management. |
| 166 | The field of robotics is seeing remarkable advancements. |
| 167 | The field of entomology studies insects and their behaviors. |
| 168 | The field of literary criticism analyzes literary works. |
| 169 | Her expertise in the field of transportation engineering is crucial for infrastructure projects. |
| 170 | The field of pharmacology investigates how drugs affect the body. |
| 171 | She received an award for her work in the field of environmental science. |
| 172 | She is working on a project in the field of urban planning. |
| 173 | Her research in the field of endocrinology has been transformative. |
| 174 | She is conducting groundbreaking work in the field of bioinformatics. |
| 175 | The field of veterinary medicine cares for animal health. |
| 176 | Her work in the field of evolutionary psychology examines psychological traits. |
| 177 | She has made a name for herself in the field of textile engineering. |
| 178 | The field of forensic science applies scientific methods to criminal investigations. |
| 179 | The field of quantum optics studies the behavior of light and matter at the quantum level. |
| 180 | He is renowned in the field of human-computer interaction. |
| 181 | The field of biotechnology holds great promise for the future. |
| 182 | The field of paleoclimatology reconstructs past climate conditions. |
| 183 | His contributions to the field of economics have been groundbreaking. |
| 184 | He has a deep interest in the field of robotics. |
| 185 | The field of musicology analyzes music history and theory. |
| 186 | Her expertise in the field of forensic anthropology aids in criminal investigations. |
| 187 | The field of computational genetics analyzes genetic data using computational methods. |
| 188 | Her expertise in the field of legal studies is unmatched. |
| 189 | He is a specialist in the field of cardiology. |
| 190 | She has made significant strides in the field of biotechnology. |
| 191 | Her contributions to the field of environmental law are significant. |
| 192 | He is a key player in the field of financial technology. |
| 193 | Her research in the field of computational neuroscience is influential. |
| 194 | She is interested in the field of cognitive science. |
| 195 | He has a background in the field of telecommunications. |
| 196 | He is a respected authority in the field of clinical research. |
| 197 | Her discoveries in the field of biochemistry have been revolutionary. |
| 198 | Her research in the field of artificial life simulates biological processes in computer models. |
| 199 | She has a strong background in the field of political science. |
| 200 | The field of bibliometrics analyzes academic publication patterns. |
| 201 | Her contributions to the field of educational technology are noteworthy. |
| 202 | The field of meteorology studies weather patterns and forecasting. |
| 203 | Her work in the field of financial engineering optimizes investment strategies. |
| 204 | The field of bioethics navigates moral issues in medicine. |
| 205 | The field of environmental economics studies the economic impact of environmental policies. |
| 206 | She has a strong foundation in the field of human resources. |

*Continued on next page*

*Continued from previous page*

| Index | Sentence |
|---|---|
| 207 | He has a background in the field of peace studies and conflict resolution. |
| 208 | The field of organizational behavior studies how individuals and groups behave within organizations. |
| 209 | He has made significant contributions to the field of renewable energy. |
| 210 | The field of forestry studies the management of forests and natural resources. |
| 211 | The field of agricultural science seeks to improve food production. |
| 212 | The field of photonics involves the study of light generation and manipulation. |
| 213 | He is a recognized expert in the field of computational physics. |
| 214 | He is an innovator in the field of genetic research. |
| 215 | The field of computational biology uses computational methods to analyze biological data. |
| 216 | Her research in the field of human-computer interaction improves user experience. |
| 217 | The field of molecular biology examines the building blocks of life. |
| 218 | His research in the field of telecommunications has advanced the industry. |
| 219 | He is researching climate change within the field of environmental studies. |
| 220 | The field of historical linguistics studies language change over time. |
| 221 | The field of aerospace medicine focuses on the health of pilots and astronauts. |
| 222 | The field of educational sociology examines educational institutions and processes. |
| 223 | She has dedicated her career to the field of education. |
| 224 | Her expertise in the field of public health is widely recognized. |
| 225 | She is advancing the field of clinical psychology. |
| 226 | The field of visual arts encompasses various creative disciplines. |
| 227 | She is a renowned expert in the field of pediatric medicine. |
| 228 | He has a strong background in the field of systems engineering. |
| 229 | The field of astrophysics seeks to understand the universe. |
| 230 | Her work in the field of bioacoustics explores animal communication through sound. |
| 231 | He is a renowned expert in the field of computational linguistics. |
| 232 | He has a profound impact on the field of digital marketing. |
| 233 | The field of marine science explores oceanic systems. |
| 234 | The field of materials science involves studying the properties of materials. |
| 235 | The field of bioinformatics combines biology and computer science. |
| 236 | The field of environmental chemistry studies chemical processes in the environment. |
| 237 | The field of cognitive development explores how thinking processes evolve over time. |
| 238 | The field of toxicology studies the effects of chemicals on living animals. |
| 239 | Her research in the field of climate science is groundbreaking. |
| 240 | Her contributions to the field of computational fluid dynamics are substantial. |
| 241 | The field of fluid dynamics studies the behavior of liquids and gases. |
| 242 | The field of petrochemical engineering deals with petroleum products. |
| 243 | The field of ergonomics designs equipment for efficiency. |
| 244 | She is a key figure in the field of digital humanities. |
| 245 | He has a background in the field of educational leadership. |
| 246 | He is a pioneer in the field of nanotechnology. |
| 247 | The field of genetics explores the inheritance of traits. |
| 248 | He has a deep interest in the field of computational linguistics. |

*Continued on next page*

*Continued from previous page*

| Index | Sentence |
|---|---|
| 249 | The field of political sociology studies political institutions and behavior. |
| 250 | The field of paleontology uncovers the history of life on Earth. |
| 251 | The field of astrophysics reveals the wonders of the cosmos. |
| 252 | The field of cognitive science examines how we think and learn. |
| 253 | He has a prolific career in the field of synthetic biology. |
| 254 | The field of quantum computing is still in its infancy. |
| 255 | Her career in the field of art history has been illustrious. |
| 256 | He is making significant contributions to the field of microbiology. |
| 257 | He has a deep understanding of the field of financial mathematics. |
| 258 | The field of criminology examines the causes of crime. |
| 259 | The field of computational linguistics combines language and computing. |
| 260 | He is a leader in the field of pharmacology. |
| 261 | The field of actuarial science helps manage financial risks. |
| 262 | The field of environmental economics addresses the impact of economic activity on natural resources. |
| 263 | Her work in the field of cognitive neuroscience investigates brain function. |
| 264 | Her research in the field of behavioral ecology examines animal behavior. |
| 265 | Her research in the field of nutrition has led to healthier eating guidelines. |
| 266 | The field of developmental linguistics studies language acquisition in children. |
| 267 | The field of gerontology explores aging and its effects on individuals and societies. |
| 268 | The field of dialectology studies regional differences in language. |
| 269 | She has dedicated her life to the field of humanitarian aid. |
| 270 | The field of information technology is constantly changing. |
| 271 | He is an expert in the field of acoustical engineering. |
| 272 | The field of developmental psychology studies human growth and development. |
| 273 | She is advancing the field of artificial intelligence. |
| 274 | The field of climate science studies weather and climate change. |
| 275 | He is a thought leader in the field of sustainable development. |
| 276 | The field of evolutionary ecology examines the adaptation of organisms to their environments. |
| 277 | He has made notable contributions to the field of computer engineering. |
| 278 | She is a distinguished researcher in the field of plant biology. |
| 279 | Her research in the field of conservation biology protects biodiversity. |
| 280 | The field of computational chemistry models chemical structures and reactions. |
| 281 | His work in the field of artificial neural networks is groundbreaking. |
| 282 | Her work in the field of atmospheric science predicts weather patterns. |
| 283 | The field of synthetic chemistry creates new compounds. |
| 284 | The field of archaeology uncovers the secrets of ancient civilizations. |
| 285 | The field of chemistry explores the properties of matter. |
| 286 | Advances in the field of medicine have improved patient outcomes significantly. |
| 287 | He has a deep understanding of the field of artificial intelligence. |
| 288 | Her contributions to the field of visual perception are significant. |
| 289 | He has made significant contributions to the field of behavioral economics. |
| 290 | Her expertise in the field of environmental sociology addresses human-environment interactions. |
| 291 | He is a respected voice in the field of climatology. |
| 292 | The field of visual arts encompasses painting, sculpture, and more. |
| 293 | He has made significant advances in the field of cognitive robotics. |

*Continued on next page*

*Continued from previous page*

| Index | Sentence |
|---|---|
| 294 | The field of cyber security is critical in today's digital world. |
| 295 | The field of geology studies the Earth's physical structure. |
| 296 | The field of educational psychology applies psychology to educational settings. |
| 297 | The field of behavioral genetics investigates the genetic basis of behavior. |
| 298 | He is a pioneer in the field of computational photography. |
| 299 | He has a background in the field of social psychology. |
| 300 | She has a strong interest in the field of sociology. |
| 301 | Her work in the field of psychology has been groundbreaking. |
| 302 | He has a deep understanding of the field of environmental microbiology. |
| 303 | The field of cognitive neuroscience studies the biological basis of cognition. |
| 304 | She is a prominent researcher in the field of computer vision. |
| 305 | He is an expert in the field of social network analysis. |
| 306 | The field of computational linguistics develops algorithms for natural language processing. |
| 307 | The field of ergonomics designs equipment to improve human use. |
| 308 | He is an expert in the field of urban sociology. |
| 309 | She chose to specialize in the field of bioinformatics. |
| 310 | The field of educational psychology applies psychological principles to education. |
| 311 | The field of artificial intelligence is evolving rapidly. |
| 312 | The field of developmental biology examines the growth of organisms. |
| 313 | The field of occupational therapy helps people perform daily activities. |
| 314 | The field of neuropsychology studies the brain-behavior relationship. |
| 315 | He is a leader in the field of sustainable agriculture. |
| 316 | Her expertise in the field of digital anthropology explores online cultures. |
| 317 | He is a leader in the field of renewable energy. |
| 318 | The field of economics encompasses a wide range of topics. |
| 319 | He has a notable career in the field of emergency management. |
| 320 | The field of population genetics investigates genetic variation within populations. |
| 321 | The field of cultural heritage management preserves historical artifacts. |
| 322 | The field of computational genomics analyzes genetic data using computational methods. |
| 323 | The field of cultural sociology examines cultural patterns and practices. |
| 324 | He is conducting research in the field of renewable energy. |
| 325 | The field of biomedical informatics combines healthcare and data science. |
| 326 | The field of astrophysics explores the mysteries of the universe. |
| 327 | The field of cryptography focuses on securing communication. |
| 328 | The field of computational physics uses numerical methods to study physical phenomena. |
| 329 | She has a degree in the field of marine ecology. |
| 330 | Her work in the field of museum studies enhances cultural preservation. |
| 331 | The field of environmental toxicology studies the effects of pollutants. |
| 332 | The field of media studies examines how media affects society. |
| 333 | He is a prominent figure in the field of data analytics. |
| 334 | He has a background in the field of industrial psychology. |
| 335 | Her research in the field of child psychology is pioneering. |
| 336 | His contributions to the field of artificial intelligence are notable. |
| 337 | The field of biomedical engineering innovates healthcare technologies. |

*Continued on next page*

*Continued from previous page*

| Index | Sentence |
|---|---|
| 338 | He is a leading expert in the field of developmental economics. |
| 339 | He is a recognized authority in the field of bioinformatics. |
| 340 | The field of neuroeconomics combines neuroscience, psychology, and economics. |
| 341 | He is a pioneer in the field of artificial life research. |
| 342 | She is an authority in the field of biomedical engineering. |
| 343 | The field of archaeology uncovers the mysteries of ancient civilizations. |
| 344 | The field of computational social science uses data to study social phenomena. |
| 345 | She has published extensively in the field of medieval literature. |
| 346 | Her work in the field of digital humanities bridges technology and humanities research. |
| 347 | She is an authority in the field of network security. |
| 348 | He has a deep understanding of the field of nanomaterials. |
| 349 | He is an authority in the field of digital anthropology. |
| 350 | The field of game design creates interactive entertainment experiences. |
| 351 | Her research in the field of evolutionary linguistics explores language evolution. |
| 352 | He is a leading researcher in the field of human rights law. |
| 353 | The field of forensic science is crucial for solving crimes. |
| 354 | He is a leading expert in the field of urban ecology. |
| 355 | The field of marine biology studies ocean ecosystems.Her innovative ideas have reshaped the field of urban design. |
| 356 | The field of biomedical sciences advances medical knowledge. |
| 357 | His career in the field of nanotechnology is flourishing. |
| 358 | She is a leading expert in the field of genetics. |
| 359 | The field of data science is transforming industries worldwide. |
| 360 | The field of genetic engineering is a hot topic in scientific circles. |
| 361 | He is well-versed in the field of cultural anthropology. |
| 362 | The field of ethnomusicology studies music within cultural contexts. |
| 363 | The field of psychometrics measures psychological traits and abilities. |
| 364 | Her work in the field of game theory has practical applications in economics. |
| 365 | She is a thought leader in the field of educational technology. |
| 366 | The field of psychology studies the human mind and behavior. |
| 367 | The field of linguistics offers many fascinating areas of study. |
| 368 | The field of public policy shapes governance and society. |
| 369 | Her research in the field of linguistics has garnered international acclaim. |
| 370 | The field of international relations examines global politics. |
| 371 | Her contributions to the field of artificial intelligence are substantial. |
| 372 | He is a key player in the field of international relations. |
| 373 | The field of ecological economics integrates ecology and economics for sustainable development. |
| 374 | The field of sociology examines social behavior and institutions. |
| 375 | He is considered a pioneer in the field of nanotechnology. |
| 376 | He is an authority in the field of risk management. |
| 377 | He is a recognized authority in the field of rehabilitation engineering. |
| 378 | The field of cognitive science integrates psychology, neuroscience, and linguistics. |
| 379 | His expertise in the field of supply chain management is invaluable. |
| 380 | The field of computer vision develops algorithms for interpreting visual data. |
| 381 | She is a thought leader in the field of environmental law. |
| 382 | The field of cybersecurity is essential for protecting information systems. |

*Continued on next page*

*Continued from previous page*

| Index | Sentence |
|---|---|
| 383 | He is a respected scholar in the field of urban planning. |
| 384 | The field of biophysics combines biology and physics principles. |
| 385 | The field of climate modeling predicts future climate changes. |
| 386 | Her work in the field of neuroethics addresses the moral implications of neuroscience. |
| 387 | Her insights in the field of strategic management are highly valued. |
| 388 | She is exploring new techniques in the field of digital art. |
| 389 | The field of educational psychology helps improve teaching methods. |
| 390 | Her work in the field of social work helps vulnerable populations. |
| 391 | The field of artificial intelligence is continuously evolving. |
| 392 | He is highly respected in the field of electrical engineering. |
| 393 | He has a deep understanding of the field of molecular biology. |
| 394 | Her expertise in the field of computational chemistry aids drug discovery. |
| 395 | The field of urban sociology explores the dynamics of cities. |
| 396 | Her studies in the field of strategic management help businesses thrive. |
| 397 | He has made significant contributions to the field of game development. |
| 398 | Her work in the field of computational neuroscience models neural processes. |
| 399 | The field of medicine requires years of rigorous training. |
| 400 | The field of educational technology enhances teaching and learning through technology. |

Table 3: Field Of Study Sentences

