# OpenReview forum: "Robust High-Dimensional Mean Estimation With Low Data Size, an Empirical Study"
_TMLR — Accepted by TMLR_

### Review · Reviewer_DdfH · 2024-11-15

**Summary Of Contributions:**

This paper analysis different mean estimation approaches under contamination in the case of relatively low sample sizes and in particular when the number of dimensions exceeds the number of observations. They analyse several scenarios and different contamination regimes always assuming a samples from a Gaussian distribution that are contaminated by varying distributions Q. They summarize and include various algorithms and also add their own versions of some of those algorithms in their experiments. Finally they end with interesting real-world experiments using neural nets.

**Audience:**

Yes

**Broader Impact Concerns:**

No broader Impact concerns

**Claims And Evidence:**

Yes

**Requested Changes:**

The paper is definitely interesting and discusses an important topic. It's biggest problem from my side is the readability. This is in part due to the language used; sometimes
sentences do not seem to make sense grammatically, as well as a lack of explanations/definitions. I give some examples below, but another careful reading should be done to identify these sentences. Due to the nature of the paper which summarizes many different concepts, readability and clarity of the language used is crucial here. Letting it proofread by a third person might be helpful.


General:
------------
+ In general, I am satisfied with the range of included methods. For general robust estimation, it is also possible to use MMD minimization, as, e.g., in this paper https://arxiv.org/pdf/1912.05737. They discuss the mean estimation of a Gaussian distribution in Section 4. The authors do not need to follow this suggestion, but if it is possible to include it, I would be extremely curious how this MMD estimation would perform in this benchmark.
+ You should also explain the  \Omega, O notation, maybe even in the "key notation" table. In general make sure that you define the symbols used for a smoother reading experience.
+ Brackets around the citations seem to be missing. For instance at the end of page 2, I would write "the data to be adversarially corrupted from X (Huber (1946);...).

Introduction:
------------
+ "data lies in a very high-dimensional setting"  --> maybe change to something like "data lies in/is part of high-dimensional spaces"
+ "However, in the corrupted setting, we find that some of the theoretical improvements, after some adjustments, are the best performers signficantly besting the sample mean." --> Maybe improve this sentence, it was very confusing for me upon first reading.
+ "We also consider some cases of unknown covariance, but as straight-forward adaptations of these approaches towards estimating covariance further stresses the need for data size n as a function of d, we mostly leave this scenario for future work." This sentence is very confusing to me, it might make sense to split it up.

Background
----------------
"For many years in high dimensions, practitioners were faced with either potentially large error (e.g., on order of η √ d) in using the sample mean or other generalizations of the median" --> other? Again seems to be a bit weird.


Section 5:
------------
I am not familiar with embeddings, and so Section 5 was somewhat hard to read. As the first part of the paper is interesting to statisticians like me that are likely not very familiar with this topic, it could make sense to give more details here. These details should concern in what space the means are taken and what this mean should exactly represent: What is an in-context embedding? What is an embedding dimension? Does one simply map a word to Rq where q is the embedding dimension?

Section 6:
------------
+ I would not use "catastrophic failure"; it seems a bit unscientific but also does not really convey what the problem is exactly (it could be that the method breaks down or just performs badly). Clearer would be something like "low performance" or even "abysmal performance", if the emphasis should be on that.
+ "QUE_low_n begins to perform better with much less data size than ev_filtering_low_n, but this is unsurprising given the rapid convergence of the inlier top eigenvalue to 1."
I am confused, should it not be QUE and ev_filtering here instead of QUE_low_n and ev_filtering_low_n?

**Strengths And Weaknesses:**

Strengths
---------------
+ The paper performs extensive experiments over a wide range of settings.
+ It studies many different methods and proposes their own sensible versions, which seem to work very well
+ The real data experiments deviate from the simple simulation settings and are quite interesting
+ Authors provide code and a Python library

Weaknesses
---------------
+ The paper is relatively long and hard to read in places.
+ Notation is sometimes not explained
+ The applications, though interesting, do not necessarily help with clarity

---

> ### Author Response · Authors · 2024-12-15
>
> Thank you for your careful reading and thoughtful review of our submission!
>
> We have posted a new draft, and all updated text is shown in a light purple.
> Thanks to your suggestions we rewrote,  reorganized, or expanded several parts: most notably in the Introduction, the Background section, and the Section on word embedding models.
>
> Other changes were more localized.  We respond to a few other of your comments next:
>
> ### On using MMD for parameter estimation
>
> We are familiar with MMD, but are unsure how to use it for parameter estimation -- only how it models data non-parametrically.  We spent some time with the linked paper which implied something was possible for the special case of Gaussian MMD and Gaussian distributions with known covariance.   But it was unclear to us what the monotonic function $F$ was (which seems to link the MMD distributional distance to $\|\theta- \theta'\|$, the distance between mean parameters), and how to recover the mean parameters from the RKHS.  So we did not make this comparison.
>
> ### Explaining big-Oh notation
>
> We reworked the Section 2 : Background, and added a footnote that explains the $O()$, $\Omega()$, and $\Theta()$ notation used by the CS-algorithms community.
>
> ### citation format
>
> We made a pass on use of \cite{ vs. \citep{
>
>
> ### Exposition of embeddings
>
> We have significantly expanded the first paragraph of Section 5 to present better exposition on embeddings in ML. We detail types of embeddings, their properties, and include supporting references.
>
>
>
> ### QUE and EV in Section 6.
>
> We had a typo which led to your confusion, referring to **ev_filter_low_n** in one spot that should have been **ev_filter**.  Fixed.

---

### Review · Reviewer_khgm · 2024-11-29

**Summary Of Contributions:**

This paper studies robust mean estimators in high-dimension and low data size settings. The authors survey numerous robust mean estimators and summarized the methods. For some estimators, the authors correct parameters and steps based on their theoretical results so that the estimators perform also well in relatively low data size settings. They extensively compare those robust mean estimators on Gaussian synthetic data under uncorrupted and corrupted settings, as well as on real-world dataset other than Gaussian using LLMs.

**Audience:**

Yes

**Claims And Evidence:**

Yes

**Requested Changes:**

Please refer to Weaknesses and Questions above.

**Strengths And Weaknesses:**

I think the paper is easy to read and well-written. The proposed eigenvalue bound is simple but useful in their modification in existing methods for the considered problem setting.
Numerous experiments are conducted, and they shows extensive comparisons between the methods. Overall, I feel this paper is survey-like paper about empirical study of the robust mean estimate in high-dimension and low data size, which is of interest in ML community.

Weaknesses.
1. Although experiments on the estimators are sufficiently conducted (which I enjoyed), I think overall contribution regarding theoretical view is insufficient. For instance, could the authors provide reasoning or intuition behind why QUE generally performs better than others in their experiment settings?
2. It is difficult to distinguish the curves in Figure. A better visualization would be helpful as I think experiments are main contribution. Is it possible to provide enlarged version of some portion of figures? or in the top left figures, are curves at n=20 necessary? it is also shown in the top right figure.
3. For synthetic experiments, despite lots of corruption scenarios are considered, it only provides Gaussian for base data distribution. Including experiments with other data distributions like Poisson, Multinomial, or mixture of Gaussians would strengthen the empirical analysis.

Questions.

1. Isn't the sample mean a maximum likelihood estimator for the mean of Gaussian? Perhaps, this is the reason why the sample mean yields best performance on uncorrupted experiments? If so, it would be good to relate this with the empirical findings.
2. Isn't **good_sample_mean_error** evaluated on uncorrupted data? If it is, in the bottom left Figure 3, how does LRV outperform the good_sample_mean_error in high-dim? I could not understand how we get a better result with corrupted data than uncorrupted data.
3. In Section 5.1, "LOOCV under the sample mean represents a valuable baseline for comparison as it demonstrates the minimum error to be expected across this data set.", why would the sample mean perform with minimum error?

---

> ### Author Response · Authors · 2024-12-15
>
> Thank you for your careful reading and thoughtful review of our submission!
>
> We have posted a new draft, and all updated text is shown in a light purple.
>
>
> ### Why QUE performs best
>
> We added a short discussion on why QUE seems to perform the best in the $n \leq d$ setting.
>
> ### Synthetic non-Gaussian experiment
>
> We conducted 4 additional experiments on a few settings of non-Gaussian inlier data:  a multivariate-t, Laplace, Poisson, and a mixture of 3 Gaussians distributions. While some settings (Laplace and Poisson) behave very similar to Gaussians, the multivariate-t and Mixure of Gaussians showed interesting deviations from the Gaussian results.  We were planning to just throw this in the appendix, but we found it interesting enough to modify the main message (adding a disclaimer for deviations from Gaussians inliers), and so included it as a new Section 6 in the main text.
> Thank you for requesting these additional experiments.
>
>
> ### Better visualization?
>
> We agree that figures are somewhat cluttered.  We actually removed several algorithms and algorithmic variants from the paper since we felt they did not contribute meaningfully to the story, and cluttered things even more.
>
> At the same time, we felt it important to include shading of 1-standar deviation ranges to illustrate the natural variation in algorithmic performance.  This means when algorithms have very similar performance, they are hard to distinguish -- to some extent this is unavoidable; and we prefer this to being able to visually highlight insignificant differences.  We tried to couple this with text descriptions that makes clear where an algorithm falls when it is hard to see.  \
> **If there are instances we missed, please let us know.**
> or if you have other specific suggestions that will materially improve the paper, we will see what we can do.
>
> Regarding the removal of $n=20$ case, we felt this was important to include since the focus of the paper was on the $n < d$ case, which this is a prime example of.  In fact, struggling with a case with $d=300$ and $n=20$ was the impetus for this project.
>
> Finally, (as discussed in reply to reviewer UoCd), we have a consistent format for all figures, and replacing them all (270 of them) would be a major undertaking.  So we are a bit hesitant to unless very important.
>
>
> ## Responses to Questions:
>
> 1. We added text (on page 8) that it is not surprising that the sample mean performs best on un-corupted data.
>
> 2. Yes **good_sample_mean** is the sample mean is on the uncorrupted data.  It is a *bit* unusual that **LRV** outperforms it in some cases, but its roughly with the natural variation in performance.  We suspect it's a combination of random variation, and perhaps **LRV** somehow does better on the strange corrupted data from the DKK construction.
>
> 3. Most algorithms perform about the same here, so it is not surprising that **sample_mean** is among them (excluding ones without early halting in Figure 8).  That is especially true in the experiments where we measure LOOCV where there is not explicit corruption of the data.  Under cross-validation, if there is a natural unimodal structure, it makes sense to use that held-out data will tend to be close to the sample mean of kept-in data.
> We updated the text a bit to mention that LOOCV is adjusted to filter out outliers, so corresponds to what the robust algorithms are trying to optimize for.   Excluding these furthest-from-mean points do not have bias in direction, so should also should not effect the relative performance of the sample mean.

---

### Review · Reviewer_UoCd · 2024-12-03

**Summary Of Contributions:**

The paper focuses on the challenge of robust mean estimation in high-dimensional settings with limited data, where traditional algorithms often fail to achieve near-optimal performance. The authors experimentally evaluate various robust mean estimation techniques and find that, with certain modifications to outlier pruning, these methods can achieve similar performance to the sample mean of uncorrupted inlier data, even with very low data sizes. The authors test these methods on synthetic Gaussian data, and real-world data, including embeddings from large language models. The paper also contributes a Python library for robust mean estimation, promoting reproducibility and encouraging further research.

**Audience:**

Yes

**Claims And Evidence:**

Yes

**Requested Changes:**

- Improve Writing and Structure (Critical):
  * Rewrite sections for clarity and conciseness, especially the Background section, condensing the first three paragraphs and expanding on the sixth paragraph.
  * Better highlight the new algorithm variants, including detailed motivation for their introduction.
Figures:

- Use vector-based graphics for better readability.

- Fix formatting issues, such as redundant labels in figures.

- Expand Figure 2 to study mismatches between expected and actual corruption levels.

- Discuss and analyze instabilities in plots, such as the explosion of lee_variant in Figure 5.

- Discuss Related Work:
  Explicitly explain the link between differential privacy and robust statistics, incorporating the references:
Differential Privacy and Robust Statistics (Dwork et al., 2009)
From Robustness to Privacy and Back (Asi et al., 2023).

- Testing Framework:
Conduct experiments on synthetic data with non-Gaussian distributions to evaluate algorithm robustness under varying distributional assumptions.

- Fix Terminology and Typos

- Revise Title:
Update the title to better reflect the paper’s empirical evaluation focus, such as Empirical Evaluation of Robust High-Dimensional Mean Estimation Techniques.

**Strengths And Weaknesses:**

Strengths :

- Comprehensive Empirical Study: The paper provides a detailed empirical evaluation of existing robust mean estimation methods, emphasizing practical performance in the high-dimensional, low-data regime.
- Introduction of Algorithm Variants: The proposed modifications to existing algorithms enhance their practical applicability in scenarios where assumptions like Gaussianity may not hold.
- Real-World Applicability: The paper includes experiments on meaningful real-world datasets, making the contributions more relatable and actionable for practitioners.
- Reproducibility: The accompanying Python library is clean and scalable, providing an excellent resource for further research.

Weaknesses :

- Writing and Clarity:
  * The paper lacks a strong and clear message, with convoluted writing in several sections. For instance, the contributions related to new algorithm variants are insufficiently detailed and not well-highlighted in the structure. For an example of a benchmark article that is ambitious in its scope while still managing to maintain clarity, the authors may refer to Descending through a Crowded Valley — Benchmarking Deep Learning Optimizers (Schmidt et al., 2021).
  * The Background section is verbose, with the first three paragraphs easily condensed into one succinct paragraph. The sixth paragraph is unclear and needs elaboration.

- Figures:
  * The figures are exported as pixel-based graphics, making them difficult to read when zoomed in. Switching to vector-based graphics is recommended.
  * Redundant labelling (e.g., y-axis and legends both reading "error") reduces readability.
  * In Figure 2, the authors should explore cases where the actual corruption level deviates from the expected corruption level.
  * Instability in some plots, such as the explosion of lee_variant in Figure 5 under high corruption, needs to be analyzed and discussed.

- Link to Related Work:
  * The connection between differential privacy and robust statistics is mentioned but not explicitly explained. An important heuristic is that robust estimators are more amenable to privatization via noise addition, and that private mechanisms (i.e. for which it is hard to test membership) are naturally robust. Suggested references include:
Differential Privacy and Robust Statistics (Dwork et al., 2009)
From Robustness to Privacy and Back (Asi et al., 2023).

- Testing Framework: The focus on real-world data is valuable, but additional experiments on synthetic data deviating from Gaussianity are necessary to fully characterize the algorithms' robustness, and to understant the failing situations of each algorithm.

Terminology and Typos:

- Page 2: The term "isometric covariance" should likely be replaced with "isotropic covariance" for clarity and consistency with standard terminology. To be clear, the only isometric (i.e. orthogonal) covariance matrix is the identity matrix, so what is written is not wrong, yet the formulation is confusing.
- The statement "this can be shown to concentrate tightly" should specify the concentration sense (e.g., with high probability).
- The explanation of the trace of the covariance matrix ("where Tr(Σ) is the trace of the covariance matrix") appears in a confusing order and should be clarified.
- The reference to Lee & Valiant (2022) for the scaling of Gaussian variance with the trace of the covariance matrix is unnecessary; this is a basic property of Gaussian distributions.

- Title: The title does not reflect the empirical nature of the work. A clearer alternative could be: Empirical Evaluation of Robust High-Dimensional Mean Estimation Techniques.

---

> ### Author Response · Authors · 2024-12-15
>
> Thank you for your careful reading and thoughtful review of our submission!
>
> We have posted a new draft, and all updated text is shown in a light purple.  This includes addressing your requested changes (e.g., terminology and typos).  We next highlight a few things, as well as were we deviated from your requests.
>
> ### Strong message and clarity
>
> We reorganized the introduction to have an enumerated list for our two main contributions (1) the large experimental study and (2) the emphasis and changes needed for the $d > n$ setting.
>
> We streamlined and tightened the first paragraph in the background.
>
> We made a few key changes to the longer background paragraph on more tangentially related work.  This includes elaborating and adding more references to the link to differential privacy.  We are reticent to elaborate too much further, since then maybe we should expand the discussion on several other areas mentioned in this paragraph, and we do not think this is necessary for the main point of the paper.
>
> ### Mismatch in Corruption Levels
>
> We already devoted a subsection in the appendix (Appendix A.6) to the effect of the relation between the expected (upper bound of) the level of corruption $\tau$ and the actual amount of corruption $\eta$.  As long as $\tau > \eta$ as intended, not much interesting happens, so we felt this was better left to the appendix to avoid distracting from the main points of the paper.  But we have now included a reference to this experiment near Figure 2.
>
> ### Redoing All Figures
>
> While our python library allows us to easily reproduce any experiment or figure, unfortunately, we did not save the raw data of the algorithmic runs used to generate the figures.  So regenerating all figures would take a substantial amount of time.  There are three listed reasons for this request:
>   - A: they are pngs not vector graphics
>   - B: they include an extra "Error" in all legends
>   - C: a request to remove the $n=20$ experiment
>
> We agree with points A and B.  However, regarding A, we believe we include figures with enough resolution that all important differences are either clear, or differences are not significant and would be hard to read anyways even if in vector graphics.  We believe we made sure to describe all of these hard-to-see similarities and distinctions in the text.  We also think point B, the extra "Error" in the legend, is correct, but not significant.
>
> Regarding point C (from reviewer khgm), we would like to keep the $n=20$ experiment in the plots, since a central point of the paper is the $n < d$ case which this represents.
>
> Ultimately, we will replace all figures  addressing points A and B **if needed** for acceptance, *but would prefer to leave them as is*, since we think it does not materially improve the paper.  Moreover, it would require a significant time investment (there are 286 figures in the paper).  If this is needed, we will request extra time to complete this revision.
>
> That said, if there a few figure that you think this would make a material difference, we will gladly update them.  The 16 new figures we added in the update have followed suggestions A and B, and we will be sure to with any further updates.
>
> ### Changing the Title
>
> We have updated the title to:\
> **Robust High-Dimensional Mean Estimation with Low Data Size, an Empirical Study**
>
> We want to keep the emphasis on the $d > n$ setting in the title, as this was our primary objective for pursuing this project.
>
>
> ### Tests on Synthetic non-Gaussian Data
>
> We conducted 4 additional experiments on a few settings of non-Gaussian inlier data:  a multivariate-t, Laplace, Poisson, and a mixture of 3 Gaussians distributions. While some settings (Laplace and Poisson) behave very similar to Gaussians, the multivariate-t and Mixure of Gaussians showed interesting deviations from the Gaussian results.  We were planning to just throw this in the appendix, but we found it interesting enough to modify the main message (adding a disclaimer for deviations from Gaussians inliers), and so included it as a new Section 6 in the main text.
> Thank you for requesting these additional experiments.

---

### Decision · Action_Editor_2bLh · 2025-01-09

**Recommendation:** Accept as is

**Comment:**

All of the reviewers were in favor of accepting this paper, and I agree with them.

**Audience:**

Mean estimation in high dimensions is a basic yet critical statistical inference task, and as evidenced by the recent papers cited in this work, is still an area of active research. This study is well-written and presents clear results which I think will be of reference to members of the TMLR community.

**Claims And Evidence:**

This paper has a clear problem setting of interest, namely mean estimation when the data size is relatively small given the data dimensionality. The authors consider numerous well-established benchmark methods, including one with their own cutomizations which they show leads to desirable estimation accuracy. The problem of interest, experimental setup, and results are presented clearly.